# Optimal Rates for Vector-Valued Spectral Regularization Learning Algorithms

**Dimitri Meunier**[*]
Gatsby Computational Neuroscience Unit
University College London
dimitri.meunier.21@ucl.ac.uk

**Zikai Shen**[*]
Department of Statistical Science
University College London
zikai.shen.22@ucl.ac.uk

**Mattes Mollenhauer**
Merantix Momentum
mattes.mollenhauer@merantix-momentum.com

**Arthur Gretton**
Gatsby Computational Neuroscience Unit
University College London
arthur.gretton@gmail.com

**Zhu Li**
Department of Mathematics
Imperial College London
zli12@ic.ac.uk

## Abstract

We study theoretical properties of a broad class of regularized algorithms with vector-valued output. These *spectral algorithms* include kernel ridge regression, kernel principal component regression and various implementations of gradient descent. Our contributions are twofold. First, we rigorously confirm the so-called *saturation effect* for ridge regression with vector-valued output by deriving a novel lower bound on learning rates; this bound is shown to be suboptimal when the smoothness of the regression function exceeds a certain level. Second, we present an upper bound on the finite sample risk for general vector-valued spectral algorithms, applicable to both well-specified and misspecified scenarios (where the true regression function lies outside of the hypothesis space), and show that this bound is minimax optimal in various regimes. All of our results explicitly allow the case of infinite-dimensional output variables, proving consistency of recent practical applications.

## 1 Introduction

We investigate a fundamental topic in modern machine learning—the behavior and efficiency of learning algorithms for regression in high-dimensional and potentially infinite-dimensional output spaces $\mathcal{Y}$. Given two random variables $X$ and $Y$, we seek to empirically minimize the squared expected risk

$$\mathcal{E}(F) \coloneqq \mathbb{E}\left[\|Y - F(X)\|_{\mathcal{Y}}^2\right] \tag{1}$$

over functions $F$ in a *reproducing kernel Hilbert space* consisting of vector-valued functions from a topological space $\mathcal{X}$ to a Hilbert space $\mathcal{Y}$. The study of this setting as an ill-posed statistical inverse problem is well established: see e.g. 46, 6, 53, 3, 5, 17. In this work, we study the setting when $\mathcal{Y}$ is high- or infinite-dimensional, since it has been less well covered by the literature, yet has many applications in multitask regression [7, 2] and infinite-dimensional learning problems, including

---

[*]Equal Contribution.

38th Conference on Neural Information Processing Systems (NeurIPS 2024).

the conditional mean embedding [20, 21, 41], structured prediction [11, 12], causal inference [43], regression with instrumental and proximal variables [42, 35], the estimation of linear operators and dynamical systems [47, 37, 26, 38, 25], and functional regression [24]. Interestingly, the aforementioned infinite-dimensional applications typically use the classical *ridge regression* algorithm. Our goal is to motivate the use of alternative learning algorithms in these settings, while providing strong theoretical guarantees.

Classically, the ill-posed problem (1) is solved via regularization strategies, which are often implemented in terms of so-called *spectral filter functions* in the context of inverse problems in Hilbert spaces [16]. When applied to the learning problem given by (1), these filter functions correspond to learning algorithms including ridge regression, a variety of different implementations of *gradient descent*, *principal component regression*, and other related methods (we refer the reader to 19 and 2 for overviews of the real-valued and vector-valued output variable case, respectively). Algorithms based on spectral filter functions when $\mathcal{Y} = \mathbb{R}$ are studied extensively, see e.g. [5, 34]. To the best of our knowledge, the detailed behavior of this general class of algorithms has remained unknown when $\mathcal{Y}$ is a general Hilbert space, with the exception of a few results for special cases in the setting of ridge regression [6, 31].

**Overview of our contributions.** In this manuscript, we aim to theoretically understand vector-valued spectral learning algorithms. The contribution of our work is twofold: (i) we rigorously confirm the *saturation effect* of ridge regression for general Hilbert spaces $\mathcal{Y}$ (see paragraph below) in the context of lower bounds on rates for the learning problem (1) and (ii) we cover a gap in the existing literature by providing *upper rates for general spectral algorithms* in high- and infinite-dimensional spaces. Our results explicitly allow the *misspecified learning case* in which the true regression function is not contained in the hypothesis space. We base our analysis on the concept of *vector-valued interpolation spaces* introduced by [30, 31]. The interpolation space norms measure the smoothness of the true regression function, replacing typical source conditions found in the literature which only cover the well-specified case. *To the best of our knowledge, these are the first bounds covering this general setting for vector-valued spectral algorithms.*

**Saturation effect of ridge regression.** The widely-used ridge regression algorithm is known to exhibit the so-called saturation effect: it fails to exploit additional smoothness in the target function beyond a certain threshold. This effect has been thoroughly investigated in the context of Tikhonov regularization in inverse problems [16, Chapter 5], but is generally reflected only in upper rates in the learning literature, see e.g. [34, 5]. Interestingly, existing lower bounds [6, 5, 31] are usually formulated in a more general setting and do not reflect this saturation effect, leaving a gap between upper and lower rates. We leverage the bias-variance decomposition paradigm to lower bound the learning risk of kernel ridge regression with vector-valued output, in order to close this gap.

**Learning rates of vector-valued spectral algorithms.** Motivated by the fact that the saturation effect is technically unavoidable with vector-valued ridge regression, we proceed to study the generalization error of popular alternative learning algorithms. In particular, we provide upper rates in the vector-valued setting consistent with the known behavior of spectral algorithms in the real-valued learning setting, based on their so-called *qualification property* [5, 34]. In particular, we confirm that a saturation effect can be bypassed in high and infinite dimensions by algorithms such as principal component regression and gradient descent, allowing for a better sample complexity for high-smoothness problems. Furthermore, we study the misspecified setting and show that upper rates for spectral algorithms match the state-of-the-art upper rates for misspecified vector-valued ridge regression recently obtained by [31]. Those rates are optimal for a wide variety of settings. Moreover, we argue that applications of vector-valued spectral algorithms are easy to implement by making use of an extended *represener theorem* based on [2], allowing for the numerical evaluation based on empirical data—even in the infinite-dimensional case.

**Related Work.** The saturation effect of regularization techniques in deterministic inverse problems is well-known. For example, [40, 36, 22] study the saturation effect for Tikhonov regularization and general spectral algorithms. In the kernel statistical learning framework, the general phenomenon of saturation is discussed by e.g. [3, 19]. Recent work by [29] investigates saturation effect in the learning context by providing a lower bound on the learning rate. To the best of our knowledge, however, all studies in the statistical learning context focus on the case when $Y$ is real-valued. General upper bounds of kernel ridge regression with real-valued or finite-dimensional $Y$ have been extensively studied in the literature (see e.g., [6, 50, 8, 17]), where minimax optimal learning

rates are derived. Recent work [30, 31] studies the infinite-dimensional output space setting with Tikhonov regularization and obtains analogous minimax optimal learning rates. [23] later study a setting where both the input and output space is the infinite dimensional Sobolev RKHS and establish the minimax optimal rate. For kernel learning with spectral algorithms, existing work (see e.g., [3, 5, 32, 34, 54, 28]) focuses on real-valued output space setting and obtains optimal upper learning rates depending on the qualification number of the spectral algorithms, where only [54, 28] consider the misspecified learning scenario where the target regression function does not lie in the hypothesis space. For vector-valued output spaces, [27] considers learning with vector-valued random features. However, general investigations of spectral algorithms for vector-valued output spaces are absent in the literature.

**Structure of this paper.** This paper is structured as follows. In Section 2, we introduce mathematical preliminaries related to reproducing kernel Hilbert spaces, vector-valued regression as well as the concept of vector-valued interpolation spaces. Section 3 contains a brief review the so-called saturation effect and a corresponding novel lower bound for vector-valued kernel ridge regression. In Section 4, we investigate general spectral learning algorithms in the context of vector-valued interpolations spaces and provide our main result: upper learning rates for this setting.

## 2 Background and Preliminaries

Throughout the paper, we consider a random variable $X$ (the covariate) defined on a second countable locally compact Hausdorff space[2] $\mathcal{X}$ endowed with its Borel $\sigma$-field $\mathcal{F}_{\mathcal{X}}$, and the random variable $Y$ (the output) defined on a potentially infinite dimensional separable real Hilbert space $(\mathcal{Y}, \langle \cdot, \cdot \rangle_{\mathcal{Y}})$ endowed with its Borel $\sigma$-field $\mathcal{F}_{\mathcal{Y}}$. We let $(\Omega, \mathcal{F}, \mathbb{P})$ be the underlying probability space with expectation operator $\mathbb{E}$. Let $P$ be the push-forward of $\mathbb{P}$ under $(X, Y)$ and $\pi$ and $\nu$ be the marginal distributions on $\mathcal{X}$ and $\mathcal{Y}$, respectively; i.e., $X \sim \pi$ and $Y \sim \nu$. We use the Markov kernel $p : \mathcal{X} \times \mathcal{F}_{\mathcal{Y}} \to \mathbb{R}_+$ to express the distribution of $Y$ conditioned on $X$ as

$$\mathbb{P}[Y \in A | X = x] = \int_A p(x, dy),$$

for all $x \in \mathcal{X}$ and events $A \in \mathcal{F}_{\mathcal{Y}}$, see e.g. [15]. We introduce some notation related to linear operators on Hilbert spaces and vector-valued integration; formal definitions can be found in Appendix A for completeness, or we refer the reader to [52, 14]. The spaces of Bochner square-integrable functions with respect to $\pi$ and taking values in $\mathcal{Y}$ are written as $L_2(\mathcal{X}, \mathcal{F}_{\mathcal{X}}, \pi; \mathcal{Y})$, abbreviated as $L_2(\pi; \mathcal{Y})$. We obtain the classical Lebesgue spaces as $L_2(\pi) \coloneqq L_2(\pi; \mathbb{R})$. Throughout the paper, we write $[F]$ or more explicitly $[F]_\pi$ for the $\pi$-equivalence class of (potentially pointwise defined) measurable functions from $\mathcal{X}$ to $\mathcal{Y}$, which we naturally interpret as elements in $L_2(\pi; \mathcal{Y})$ whenever they are square-integrable. Let $H$ be a separable real Hilbert space with inner product $\langle \cdot, \cdot \rangle_H$. We write $\mathcal{L}(H, H')$ as the Banach space of bounded linear operators from $H$ to another Hilbert space $H'$, equipped with the operator norm $\| \cdot \|_{H \to H'}$. When $H = H'$, we simply write $\mathcal{L}(H)$ instead. We write $S_2(H, H')$ as the Hilbert space of Hilbert-Schmidt operators from $H$ to $H'$ and $S_1(H, H')$ as the Banach space of trace class operators (see Appendix A for a complete definition). For two Hilbert spaces $H, H'$, we say that $H$ is (continuously) embedded in $H'$ and denote it as $H \hookrightarrow H'$ if $H$ can be interpreted as a vector subspace of $H'$ and the inclusion operator $i : H \to H'$ performing the change of norms with $ix = x$ for $x \in H$ is continuous; and we say that $H$ is isometrically isomorphic to $H'$ and denote it as $H \simeq H'$ if there is a linear isomorphism between $H$ and $H'$ which is an isometry.

**Tensor Product of Hilbert Spaces:** Denote $H \otimes H'$ the tensor product of Hilbert spaces $H$, $H'$. The element $x \otimes x' \in H \otimes H'$ is treated as the linear rank-one operator $x \otimes x' : H' \to H$ defined by $y' \to \langle y', x' \rangle_{H'} x$ for $y' \in H'$. Based on this identification, the tensor product space $H \otimes H'$ is isometrically isomorphic to the space of Hilbert-Schmidt operators from $H'$ to $H$, i.e., $H \otimes H' \simeq S_2(H', H)$. We will hereafter not make the distinction between these two spaces, and treat them as being identical.

**Remark 1** (1, Theorem 12.6.1). *Consider the Bochner space $L_2(\pi; H)$ where $H$ is a separable Hilbert space. One can show that $L_2(\pi; H)$ is isometrically identified with the tensor product space $H \otimes L_2(\pi)$, and we denote as $\Psi$ the isometric isomorphism between the two spaces. See Appendix A for more details on tensor product spaces and the explicit definition of $\Psi$.*

---

[2]Under additional technical assumptions, the results in this paper can also be formulated when $\mathcal{X}$ is a more general topological space. However, some properties of kernels defined on $\mathcal{X}$ such as the so-called $c_0$-*universality* [10] simplify the exposition when $\mathcal{X}$ is a second countable locally compact Hausdorff space.

**Scalar-valued Reproducing Kernel Hilbert Space (RKHS).** We let $k : \mathcal{X} \times \mathcal{X} \to \mathbb{R}$ be a symmetric and positive definite kernel function and $\mathcal{H}$ be a vector space of functions from $\mathcal{X}$ to $\mathbb{R}$, endowed with a Hilbert space structure via an inner product $\langle \cdot, \cdot \rangle_{\mathcal{H}}$. We say that $k$ is a reproducing kernel of $\mathcal{H}$ if and only if for all $x \in \mathcal{X}$ we have $k(\cdot, x) \in \mathcal{H}$ and for all $x \in \mathcal{X}$ and $f \in \mathcal{H}$, we have $f(x) = \langle f, k(x, \cdot) \rangle_{\mathcal{H}}$. A space $\mathcal{H}$ which possesses a reproducing kernel is called a reproducing kernel Hilbert space (RKHS; see e.g. 4). We denote the canonical feature map of $\mathcal{H}$ as $\phi(x) = k(\cdot, x)$.

We require some technical assumptions on the previously defined RKHS and kernel, which we assume to be satisfied throughout the text:

1. $\mathcal{H}$ is separable: this is satisfied if $k$ is continuous, given that $\mathcal{X}$ is separable[3];
2. $k(\cdot, x)$ is measurable for $\pi$-almost all $x \in \mathcal{X}$;
3. $k(x, x) \leq \kappa^2$ for $\pi$-almost all $x \in \mathcal{X}$.

The above assumptions are not restrictive in practice, as well-known kernels such as the Gaussian, Laplace, and Matérn kernels satisfy them on $\mathcal{X} \subseteq \mathbb{R}^d$ [48]. We now introduce some facts about the interplay between $\mathcal{H}$ and $L_2(\pi)$, which has been extensively studied by [44, 45], [13] and [51]. We first define the (not necessarily injective) embedding $I_\pi : \mathcal{H} \to L_2(\pi)$, mapping a function $f \in \mathcal{H}$ to its $\pi$-equivalence class $[f]$. The embedding is a well-defined compact operator as long as its Hilbert-Schmidt norm is finite. In fact, this requirement is satisfied since its Hilbert-Schmidt norm can be computed as [51, Lemma 2.2 & 2.3] $\|I_\pi\|_{S_2(\mathcal{H}, L_2(\pi))} = \|k\|_{L_2(\pi)} \leq \kappa$. The adjoint operator $S_\pi := I_\pi^* : L_2(\pi) \to \mathcal{H}$ is an integral operator with respect to the kernel $k$, i.e. for $f \in L_2(\pi)$ and $x \in \mathcal{X}$ we have [49, Theorem 4.27]

$$(S_\pi f)(x) = \int_{\mathcal{X}} k(x, x') f(x') \, \mathrm{d}\pi(x').$$

Next, we define the self-adjoint, positive semi-definite and trace class integral operators

$$L_X := I_\pi S_\pi : L_2(\pi) \to L_2(\pi) \quad \text{and} \quad C_X := S_\pi I_\pi : \mathcal{H} \to \mathcal{H}.$$

**Vector-valued Reproducing Kernel Hilbert Space (vRKHS).** Let $K : \mathcal{X} \times \mathcal{X} \to \mathcal{L}(\mathcal{Y})$ be an operator valued positive-semidefinite (psd) kernel. Fix $K$, $x \in \mathcal{X}$, and $h \in \mathcal{Y}$, then $(K_x h)(\cdot) := K(\cdot, x)h$ defines a function from $\mathcal{X}$ to $\mathcal{Y}$. The completion of

$$\mathcal{G}_{\mathrm{pre}} := \mathrm{span}\{K_x h \mid x \in \mathcal{X}, h \in \mathcal{Y}\}$$

with inner product on $\mathcal{G}_{\mathrm{pre}}$ defined on the elementary elements as $\langle K_x h, K_{x'} h' \rangle_{\mathcal{G}} := \langle h, K(x, x') h' \rangle_{\mathcal{Y}}$, defines a vRKHS denoted as $\mathcal{G}$. For a more complete overview of the vector-valued reproducing kernel Hilbert space, we refer the reader to [9], [10] and [31, Section 2]. In the following, we will denote $\mathcal{G}$ as the vRKHS induced by the kernel $K : \mathcal{X} \times \mathcal{X} \to \mathcal{L}(\mathcal{Y})$ with

$$K(x, x') := k(x, x') \mathrm{Id}_{\mathcal{Y}}, \quad x, x' \in \mathcal{X}. \tag{2}$$

We emphasize that this family of kernels is the de-facto standard for high- and infinite-dimensional applications [20, 21, 41, 11, 12, 42, 35, 43, 37, 26, 38, 25, 24] due to the crucial *representer theorem* which gives a closed form solution for the ridge regression problem based on the data. We generalize this representer theorem to cover the general spectral algorithm case in Proposition 1.

**Remark 2** (General multiplicative kernel). *Without loss of generality, we provide our results for the vRKHS $\mathcal{G}$ induced by the operator-valued kernel given by $K(x, x') = k(x, x') \mathrm{Id}_{\mathcal{Y}}$. However, with suitably adjusted constants in the assumptions, our results transfer directly to the more general vRKHS $\widetilde{\mathcal{G}}$ induced by the more general operator-valued kernel*

$$\widetilde{K}(x, x') := k(x, x')T$$

*where $T : \mathcal{Y} \to \mathcal{Y}$ is any positive-semidefinite self-adjoint operator. The precise characterization of the adjusted constants is given by [31, Section 4.1].*

An important property of $\mathcal{G}$ is that it is isometrically isomorphic to the space of Hilbert-Schmidt operators between $\mathcal{H}$ and $\mathcal{Y}$ [31, Corollary 1]. Similarly to the scalar case we can map every element in $\mathcal{G}$ into its $\pi$−equivalence class in $L_2(\pi; \mathcal{Y})$ and we use the shorthand notation $[F] = [F]_\pi$ (see Definition 6 in Appendix A for more details).

---

[3]This follows from [49, Lemma 4.33]. Note that the Lemma requires separability of $\mathcal{X}$, which is satisfied since we assume that $\mathcal{X}$ is a second countable locally compact Hausdorff space.

**Theorem 1** (vRKHS isomorphism). *For every function $F \in \mathcal{G}$ there exists a unique operator $C \in S_2(\mathcal{H}, \mathcal{Y})$ such that $F(\cdot) = C\phi(\cdot) \in \mathcal{Y}$ with $\|C\|_{S_2(\mathcal{H}, \mathcal{Y})} = \|F\|_{\mathcal{G}}$ and vice versa. Hence $\mathcal{G} \simeq S_2(\mathcal{H}, \mathcal{Y})$ and we denote the isometric isomorphism between $S_2(\mathcal{H}, \mathcal{Y})$ and $\mathcal{G}$ as $\bar{\Psi}$. It follows that $\mathcal{G}$ can be written as $\mathcal{G} = \{F : \mathcal{X} \to \mathcal{Y} \mid F = C\phi(\cdot), C \in S_2(\mathcal{H}, \mathcal{Y})\}$.*

## 2.1 Vector-valued Regression

We briefly recall the basic setup of regularized least-squares regression with Hilbert space-valued random variables. The squared expected risk for vector-valued regression is

$$\mathcal{E}(F) := \mathbb{E}\big[\|Y - F(X)\|_{\mathcal{Y}}^2\big] = \int_{\mathcal{X} \times \mathcal{Y}} \|y - F(x)\|_{\mathcal{Y}}^2 p(x, dy) \pi(dx), \tag{3}$$

for measurable functions $F : \mathcal{X} \to \mathcal{Y}$. The analytical minimizer of the risk over measurable functions is the *regression function* or the *conditional mean function* $F_* \in L_2(\pi; \mathcal{Y})$ given by

$$F_*(x) := \mathbb{E}[Y \mid X = x] = \int_{\mathcal{Y}} y\, p(x, dy), \quad x \in \mathcal{X}.$$

Throughout the paper, we assume that $\mathbb{E}[\|Y\|_{\mathcal{Y}}^2] < +\infty$, i.e., the random variable $Y$ is square-integrable. Note that this implies $F_* \in L_2(\pi; \mathcal{Y})$. Our focus in this work is to approximate $F_*$ with kernel-based regularized least-squares algorithms, where we pay special attention to the case when $\mathcal{Y}$ is of high or infinite dimension. We pick $\mathcal{G}$ as a hypothesis space of functions in which to estimate $F_*$. Note that by Theorem 1, minimizing the functional $\mathcal{E}$ on $\mathcal{G}$ is equivalent to minimizing the following functional on $S_2(\mathcal{H}, \mathcal{Y})$,

$$\bar{\mathcal{E}}(C) := \mathbb{E}\big[\|Y - C\phi(X)\|_{\mathcal{Y}}^2\big]. \tag{4}$$

It is shown in [38, Proposition 3.5 and Section 3.4] that the optimality condition can be written as

$$C_{YX} = C_* C_X, \qquad C_* \in S_2(\mathcal{H}, \mathcal{Y}), \tag{5}$$

where $C_{YX} := \mathbb{E}[Y \otimes \phi(X)]$ is the cross-covariance operator. As discussed in full detail by [38], the problem (5) can be formulated as a potentially ill-posed inverse problem on the space of Hilbert–Schmidt operators. As such, a regularization is required; we introduce regularized solutions of this problem in Section 4 through the classical concept of spectral filter functions.

## 2.2 Vector-valued Interpolation Space and Source Condition

We now introduce the background required in order to characterize the smoothness of the target function $F_*$, both in the well-specified setting ($F_* \in \mathcal{G}$) and in the misspecified setting ($F_* \notin \mathcal{G}$). We review the results of [51] and [17] in constructing scalar-valued interpolation spaces, and [30] in defining vector-valued interpolation spaces.

**Real-valued Interpolation Space:** By the spectral theorem for self-adjoint compact operators, there exists an at most countable index set $I$, a non-increasing sequence $(\mu_i)_{i \in I} > 0$, and a family $(e_i)_{i \in I} \in \mathcal{H}$, such that $([e_i])_{i \in I}$ [4] is an orthonormal basis (ONB) of $\overline{\mathrm{ran}\, I_\pi} \subseteq L_2(\pi)$ and $(\mu_i^{1/2} e_i)_{i \in I}$ is an ONB of $(\ker I_\pi)^\perp \subseteq \mathcal{H}$, and we have

$$L_X = \sum_{i \in I} \mu_i \langle \cdot, [e_i] \rangle_{L_2(\pi)} [e_i], \qquad C_X = \sum_{i \in I} \mu_i \langle \cdot, \mu_i^{\frac{1}{2}} e_i \rangle_{\mathcal{H}} \mu_i^{\frac{1}{2}} e_i \tag{6}$$

For $\alpha \geq 0$, the $\alpha$-interpolation space [51] is defined by

$$[\mathcal{H}]^\alpha := \left\{ \sum_{i \in I} a_i \mu_i^{\alpha/2} [e_i] : (a_i)_{i \in I} \in \ell_2(I) \right\} \subseteq L_2(\pi),$$

equipped with the inner product

$$\left\langle \sum_{i \in I} a_i (\mu_i^{\alpha/2} [e_i]), \sum_{i \in I} b_i (\mu_i^{\alpha/2} [e_i]) \right\rangle_{[\mathcal{H}]^\alpha} = \sum_{i \in I} a_i b_i,$$

---

[4] We recall that the bracket $[\cdot]$ denotes the embedding that maps $f$ to its equivalence class $I_\pi(f) \in L_2(\pi)$.

for $(a_i)_{i \in I}, (b_i)_{i \in I} \in \ell_2(I)$. The $\alpha$-interpolation space defines a Hilbert space. Moreover, $\left( \mu_i^{\alpha/2} [e_i] \right)_{i \in I}$ forms an ONB of $[\mathcal{H}]^\alpha$ and consequently $[\mathcal{H}]^\alpha$ is a separable Hilbert space. In the following, we use the abbreviation $\| \cdot \|_\alpha := \| \cdot \|_{[\mathcal{H}]^\alpha}$.

**Vector-valued Interpolation Space:** Introduced in [30], vector-valued interpolation spaces generalize the notion of scalar-valued interpolation spaces to vRKHS with a kernel of the form (2).

**Definition 1** (Vector-valued interpolation space). *Let $k$ be a real-valued kernel with associated RKHS $\mathcal{H}$ and let $[\mathcal{H}]^\alpha$ be the real-valued interpolation space associated to $\mathcal{H}$ with some $\alpha \geq 0$. The vector-valued interpolation space $[\mathcal{G}]^\alpha$ is defined as (refer to Remark 1 for the definition of $\Psi$)*

$$[\mathcal{G}]^\alpha := \Psi\left( S_2([\mathcal{H}]^\alpha, \mathcal{Y}) \right) = \{ F \mid F = \Psi(C), \; C \in S_2([\mathcal{H}]^\alpha, \mathcal{Y}) \}.$$

*The space $[\mathcal{G}]^\alpha$ is a Hilbert space equipped with the inner product*

$$\langle F, G \rangle_\alpha := \langle C, L \rangle_{S_2([\mathcal{H}]^\alpha, \mathcal{Y})} \qquad (F, G \in [\mathcal{G}]^\alpha),$$

*where $C = \Psi^{-1}(F)$, $L = \Psi^{-1}(G)$. For $\alpha = 0$, we retrieve $\|F\|_0 = \|F\|_{L_2(\pi; \mathcal{Y})} = \|C\|_{S_2(L_2(\pi), \mathcal{Y})}$.*

**Remark 3** (Interpolation space inclusions). *Note that we have $F_* \in L_2(\pi; \mathcal{Y})$ since $Y \in L_2(\mathbb{P}; \mathcal{Y})$ by assumption. Furthermore, for $0 < \beta < \alpha$, [17, Eq. (7)] imply the inclusions*

$$[\mathcal{G}]^\alpha \hookrightarrow [\mathcal{G}]^\beta \hookrightarrow [\mathcal{G}]^0 \subseteq L_2(\pi; \mathcal{Y}).$$

*Under assumptions 1 to 3 and with $\mathcal{X}$ being a second-countable locally compact Hausdorff space, $[\mathcal{G}]^0 = L_2(\pi; \mathcal{Y})$ if and only if $\mathcal{H}$ is dense in the space of continuous functions vanishing at infinity, equipped with the uniform norm [31, Remark 4].*

**Remark 4** (Well-specified versus misspecified setting). *We say that we are in the well-specified setting if $F_* \in [\mathcal{G}]^1$. In this case, there exists $\bar{F} \in \mathcal{G}$ such that $F_* = \bar{F}$ $\pi$−almost surely and $\|F_*\|_1 = \|\bar{F}\|_{\mathcal{G}}$, i.e. $F_*$ admits a representer in $\mathcal{G}$ (see Remark 5 in Appendix A). When $F_* \in [\mathcal{G}]^\beta$ for $\beta < 1$, $F_*$ may not admit such a representation and we are in the misspecified setting, as $[\mathcal{G}]^1 \subseteq [\mathcal{G}]^\beta$.*

Definition 1 and Remarks 3 and 4 motivate the use of following assumption on the smoothness of the target function: there exists $\beta > 0$ and a constant $B \geq 0$ such that $F_* \in [\mathcal{G}]^\beta$ and

$$\|F_*\|_\beta \leq B. \tag{SRC}$$

We let $C_* := \Psi^{-1}(F_*) \in S_2([\mathcal{H}]^\beta, \mathcal{Y})$. (SRC) directly generalizes the notion of a so-called Hölder-type source condition in the learning literature [6, 17, 32, 34] and allows to characterize the misspecified learning scenario.

## 2.3 Further Assumptions

In addition to (SRC), we require standard assumptions to obtain the precise learning rates for kernel learning algorithms. We list them below. For constants $D_2 > 0$ and $p \in (0, 1]$ and for all $i \in I$,

$$\mu_i \leq D_2 i^{-1/p}. \tag{EVD}$$

For constants $D_1, D_2 > 0$ and $p \in (0, 1)$ and for all $i \in I$,

$$D_1 i^{-\frac{i}{p}} \leq \mu_i \leq D_2 i^{-1/p}. \tag{EVD+}$$

(EVD) and (EVD+) are standard assumptions on the *eigenvalue decay* of the integral operator: they describe the interplay between the marginal distribution $\pi$ and the RKHS $\mathcal{H}$ (see more details in 6, 17). (EVD+) is needed in order to establish lower bounds on the excess risk. Note that we have excluded the value $p = 1$ from (EVD+); indeed, $p = 1$ is incompatible with the assumption of a bounded kernel, a fact missed by previous works and of independent interest (see Appendix, Remark 7).

For $\alpha \in [p, 1]$, the inclusion $I_\pi^{\alpha, \infty} : [\mathcal{H}]^\alpha \hookrightarrow L_\infty(\pi)$ is continuous, and $\exists A > 0$ such that

$$\|I_\pi^{\alpha, \infty}\|_{[\mathcal{H}]^\alpha \to L_\infty(\pi)} \leq A. \tag{EMB}$$

Property (EMB) is referred to as the *embedding property* in [17]. It can be shown that it holds if and only if there exists a constant $A \geq 0$ with $\sum_{i \in I} \mu_i^\alpha e_i^2(x) \leq A^2$ for $\pi$-almost all $x \in \mathcal{X}$ [17, Theorem 9]. Since we assume $k$ to be bounded, the embedding property always hold true when $\alpha = 1$.

Furthermore, (EMB) implies a polynomial eigenvalue decay of order $1/\alpha$, which is why we take $\alpha \geq p$. (EMB) is not needed when we deal with the well-specified setting, but is crucial to bound the excess risk in the misspecified setting.

Finally, we assume that there are constants $\sigma, R > 0$ such that

$$\int_{\mathcal{Y}} \|y - F_*(x)\|_{\mathcal{Y}}^q p(x, dy) \leq \frac{1}{2} q! \sigma^2 R^{q-2}, \tag{MOM}$$

is satisfied for $\pi$-almost all $x \in \mathcal{X}$ and all $q \geq 2$. The (MOM) condition on the Markov kernel $p(x, dy)$ is a *Bernstein moment condition* used to control the noise of the observations (see 6, 17 for more details). If $Y$ is almost surely bounded, for example $\|Y\|_{\mathcal{Y}} \leq Y_\infty$ almost surely, then (MOM) is satisfied with $\sigma = R = 2Y_\infty$. It is possible to prove that the Bernstein condition is equivalent to sub-exponentiality, see [38, Remark 4.9].

## 3   Saturation Effect of Kernel Ridge Regression

The most established way of learning $F_*$ is by kernel ridge regression (KRR), which can be formulated as the following optimization problem: given a dataset $D = \{(x_i, y_i)\}_{i=1}^n$ independently and identically sampled from the joint distribution of $X$ and $Y$,

$$\hat{F}_\lambda := \arg\min_{F \in \mathcal{G}} \frac{1}{n} \sum_{i=1}^n \|y_i - F(x_i)\|_{\mathcal{Y}}^2 + \lambda \|F\|_{\mathcal{G}}^2, \tag{7}$$

where $\lambda > 0$ is the regularization parameter. The generalization error of vector-valued KRR is expressed as $\hat{F}_\lambda - F_*$, and controlled in different norms: see [31] for an extensive study. We recall here a simplified special case of the key results obtained in this work. In the next Theorem, $\lesssim, \gtrsim$ are inequality up to positive multiplicative constants that are independent of $n$.

**Theorem 2** (Upper and lower bounds for KRR in the well-specified regime). *Let $\hat{F}_\lambda$ be the KRR estimator from* (7). *Furthermore, let the conditions* (EVD+)*,* (SRC) *and* (MOM) *be satisfied for some $0 < p \leq 1$ and $\beta \geq 1$. Then, with high probability we have*

$$\left\| [\hat{F}_{\lambda_n}] - F_* \right\|_{L_2(\pi; \mathcal{Y})}^2 \lesssim n^{-\frac{\min\{\beta, 2\}}{\min\{\beta, 2\} + p}} \quad \text{for a choice } \lambda_n = \Theta\left(n^{-\frac{1}{\beta + p}}\right),$$

*and furthermore for all learning methods (i.e., measurable maps) of the form $D \to \hat{F}_D$,*

$$\left\| [\hat{F}_D] - F_* \right\|_{L_2(\pi; \mathcal{Y})}^2 \gtrsim n^{-\frac{\beta}{\beta + p}}.$$

Theorem 2 shows the minimax optimal learning rate for vector-valued KRR for $\beta \in [1, 2]$. However, when $\beta > 2$, the obtained upper bound saturates at $n^{-\frac{2}{2+p}}$, creating a gap with the lower bound. This phenomenon is referred to as the saturation effect of Tikhonov regularization, and has been well investigated in deterministic inverse problems [40]. In the case where $\mathcal{Y}$ is real-valued, [29] prove that the saturation effect cannot be avoided with Tikhonov regularization. Below, we give a similar but generalized bound on lower rates for the case that $\mathcal{Y}$ is a Hilbert space. For this result only, we assume that $\mathcal{X}$ is a compact subset of $\mathbb{R}^d$. We give the proof in Appendix B.

**Theorem 3** (Saturation of KRR). *Let $\mathcal{X}$ be a compact subset of $\mathbb{R}^d$. Let $\lambda = \lambda(n)$ be an arbitrary choice of regularization parameter satisfying $\lambda(n) \to 0$ as $n \to +\infty$ and let $\hat{F}_\lambda$ be the KRR estimator from* (7). *We assume that the noise is non-zero and bounded below, i.e. there exists $\sigma > 0$, such that*

$$\int_{\mathcal{Y}} \|y - F_*(x)\|_{\mathcal{Y}}^2 p(x, dy) \geq \sigma^2,$$

*is satisfied for $\pi$-almost all $x \in \mathcal{X}$. We assume in addition and for this result only that $k$ is Hölder continuous (see Definition 11 in the appendix), i.e., $k \in C^\theta(\mathcal{X} \times \mathcal{X})$ for $\theta \in (0, 1]$. Suppose that Assumptions* (EVD+) *and* (SRC) *hold with $p \in (0, 1)$ and $\beta \geq 2$. For $\tau \geq 0$, for sufficiently large $n > 0$, where the hidden index bound depends on $\tau$, with probability greater than $1 - e^{-\tau}$, there exists some constant $c_\tau > 0$ such that*

$$\mathbb{E}\left[ \left\| [\hat{F}_\lambda] - F_* \right\|_{L_2(\pi; \mathcal{Y})}^2 \Big| x_1, \ldots, x_n \right] \geq c_\tau n^{-\frac{2}{2+p}}.$$

The assumption that $k$ is Hölder continuous is crucial in lower bounding the variance with a covering number argument. Kernels satisfying this assumption include Gaussian kernels, Laplace kernels and Matérn kernels. Theorem 3 clearly demonstrates that the learning rate from vector-valued KRR cannot reach the information theoretic lower rate given in Theorem 2.

As discussed above, [29] propose a similar lower bound in the real-valued case, and we now highlight two fundamental differences with [29] in the proof. First, while both works adopt the same bias-variance decomposition, we need to lower bound the bias and the variance term with infinite-dimensional output in our setting. Second, we adopt a different and simpler approach in proving the lower bound, since there are a number of issues with the proof of [29], both in the treatment of the bias and of the variance. For a detailed comparison with the earlier work, and an explanation of the differences in our approach, please refer to Remark 6 in the Appendix.

## 4 Consistency and optimal rates for general spectral algorithms

**Regularized population solution**: Our goal is to regularize (5) in such a way that we get a unique and well-defined solution that provides a good approximation to $F_*$. We first recall the concept of a filter function (i.e., a function on an interval which is applied on self-adjoint operators to each individual eigenvalue via the spectral calculus, see 16), that will allow to define a regularization strategy. One may think of the following definition as a class of functions approximating the inversion map $x \mapsto 1/x$ while still being defined for $x = 0$ in a reasonable way. We use the definition given by [34], but equivalent definitions can be found throughout the literature.

**Definition 2** (Filter function). *Let $\Lambda \subseteq \mathbb{R}^+$. A family of functions $g_\lambda : [0, \infty) \to [0, \infty)$ indexed by $\lambda \in \Lambda$ is called a filter with qualification $\rho \geq 0$ if it satisfies the following two conditions:*

*1. There exists a positive constant $E$ such that, for all $\lambda \in \Lambda$*

$$\sup_{\alpha \in [0,1]} \sup_{x \in [0,\kappa^2]} \lambda^{1-\alpha} x^\alpha g_\lambda(x) \leq E \tag{8}$$

*2. There exists a positive constant $\omega_\rho < \infty$ such that*

$$\sup_{\alpha \in [0,\rho]} \sup_{\lambda \in \Lambda} \sup_{x \in [0,\kappa^2]} |r_\lambda(x)| x^\alpha \lambda^{-\alpha} \leq \omega_\rho, \qquad with \qquad r_\lambda(x) := 1 - g_\lambda(x)x. \tag{9}$$

Below, we give some standard examples which are discussed by e.g. [19, 5] in the context of kernel regression with scalar output variables, and in [2] for the vector-valued case. A variety of additional algorithms can be expressed in terms of a filter function.

1. *Ridge regression.* From the Tikhonov filter function $g_\lambda(x) = (x + \lambda)^{-1}$, we obtain the known ridge regression algorithm. In this case, we have $E = \rho = \omega_\rho = 1$.

2. *Gradient Descent.* From the Landweber iteration filter function given by

$$g_k(x) := \tau \sum_{i=0}^{k-1} (1 - \tau x)^i \text{ for } k := 1/\lambda, k \in \mathbb{N}$$

we obtain the gradient descent scheme with constant step size $\tau > 0$, which corresponds to the population gradient iteration given by $F_{k+1} := F_k - \tau \nabla \mathcal{E}(F_k)$ for $k \in \mathbb{N}$. In this case, we have $E = 1$ and arbitrary qualification with $\omega_\rho = 1$ whenever $0 < \rho \leq 1$ and $\omega_\rho = \rho^\rho$ otherwise. Gradient schemes with more complex update rules can be expressed in terms of filter functions as well [39, 32, 34].

3. *Kernel principal component regression.* The truncation filter function $g_\lambda(x) = x^{-1}\mathbb{1}[x \geq \lambda]$ yields kernel principal component regression, corresponding to a hard thresholding of eigenvalues at a truncation level $\lambda$. In this case we have $E = \omega_\rho = 1$ for arbitrary qualification $\rho$.

**Population solution**: Given a filter function $g_\lambda$, we call $g_\lambda(C_X)$[5] the regularized inverse of $C_X$. We may think of the regularized inverse as approximating the *pseudoinverse* of $C_X$ (see e.g. [16]) when $\lambda \to 0$. We define the regularized population solution to (4) as

$$C_\lambda := C_{YX} g_\lambda(C_X) \in S_2(\mathcal{H}, \mathcal{Y}), \qquad F_\lambda(\cdot) := C_\lambda \phi(\cdot) \in \mathcal{G}. \tag{10}$$

---

[5] $g_\lambda(C_X)$ is defined with the rules of spectral calculus, see Definition 9 in the Appendix.

The solution arising from standard regularization strategies leads to well-known statistical methodologies. We refer to [16] for the background on filter functions in classical regularization theory.

**Empirical solution**: Given the dataset $D = \{(x_i, y_i)\}_{i=1}^n$, the empirical analogue of (10) is

$$\hat{C}_\lambda := \hat{C}_{YX} g_\lambda(\hat{C}_X), \qquad \hat{F}_\lambda(\cdot) := \hat{C}_\lambda \phi(\cdot) \in \mathcal{G}, \qquad (11)$$

where $\hat{C}_{YX}, \hat{C}_X$ are empirical covariance operators define as

$$\hat{C}_X := \frac{1}{n} \sum_{i=1}^n \phi(x_i) \otimes \phi(x_i) \qquad \hat{C}_{YX} := \frac{1}{n} \sum_{i=1}^n y_i \otimes \phi(x_i).$$

Note that (11) is the regularized solution of the empirical inverse problem

$$\hat{C}_{YX} = \hat{C} \hat{C}_X, \qquad \hat{C} \in S_2(\mathcal{H}, \mathcal{Y}),$$

which arises as the optimality condition for minimizers on $\mathcal{G}$ of the empirical analogue of (3), given by $\mathcal{E}_n(F) := \frac{1}{n} \sum_{i=1}^n \|y_i - F(x_i)\|_{\mathcal{Y}}^2$; see Proposition 2 in the Appendix for a proof. For the vector-valued kernel given in (2), it is well-known that $\hat{F}_\lambda$ can be computed in closed-form for the ridge regression estimator—even in infinite dimensions [47]. For general filter functions, an extended representer theorem is given by [2] in the context of finite-dimensional multitask learning: this approach works in infinite dimensions as well. We give the closed form solution based on [2] below (we include the proof in Appendix D.1).

**Proposition 1** (Representer theorem for general spectral filter). *Let* $(\mathbf{K})_{ij} = k(x_i, x_j)$, $1 \le i, j \le n$ *denote the Gram matrix associated to the scalar-valued kernel* $k$. *We have*

$$\hat{F}_\lambda(x) = \sum_{i=1}^n y_i \alpha_i(x), \qquad \alpha(x) = \frac{1}{n} g_\lambda \left(\frac{\mathbf{K}}{n}\right) \mathbf{k}_x \in \mathbb{R}^n, \qquad (\mathbf{k}_x)_i = k(x, x_i), \quad 1 \le i \le n. \quad (12)$$

**Example 1** (Conditional integration). *Consider now a random variable* $Z$ *taking values in a topological space* $\mathcal{Z}$ *on which we define a second RKHS* $\mathcal{H}' \subseteq \mathbb{R}^{\mathcal{Z}}$ *with kernel* $\ell : \mathcal{Z} \times \mathcal{Z} \to \mathbb{R}$ *and canonical feature map* $\psi : \mathcal{Z} \to \mathcal{H}', z \mapsto \ell(z, \cdot)$. *The conditional mean embedding [47, 20] is defined as*

$$F_*(x) := \mathbb{E}[\psi(Z) \mid X = x], \qquad x \in \mathcal{X}.$$

*We immediately see the link with vector-valued regression with* $Y = \psi(Z)$ *and* $\mathcal{Y} = \mathcal{H}'$. *The conditional mean embedding allows us to compute the conditional expectation of any element of* $\mathcal{H}'$. *Indeed, using the reproducing property, for* $f \in \mathcal{H}'$, *we have for all* $x \in \mathcal{X}$,

$$\mathbb{E}[f(Z) \mid X = x] = \langle f, \mathbb{E}[\psi(Z) \mid X = x] \rangle_{\mathcal{H}'}.$$

*Given a dataset* $\{(x_i, z_i)\}_{i=1}^n$[6] *and an estimate of the conditional mean embedding* $F_*$ *with a spectral algorithm* $\hat{F}_\lambda$ *as in Eq. (11), and substituting the formula in Eq. (12), we obtain* $\mathbb{E}[f(Z) \mid X = x] \approx \langle f, \hat{F}_\lambda(x) \rangle_{\mathcal{H}'} = \sum_{i=1}^n \langle f, \psi(z_i) \rangle_{\mathcal{H}'} \alpha_i(x) = \mathbf{f}_z^\top \alpha(x)$, *where* $(\mathbf{f}_z)_i = f(z_i)$, $1 \le i \le n$.

**Learning rates:** We now give our main result, the learning rates for the difference between $[\hat{F}_\lambda]$ and $F_*$ in the interpolation norm, where $F_\lambda$ and $\hat{F}_\lambda$ are given by (10) and (11) based on a general spectral filter satisfying Definition 2. The proof is deferred to Section C in the Appendix.

**Theorem 4** (Upper learning rates). *Let* $\hat{F}_\lambda$ *be an estimator based on a general spectral filter with qualification* $\rho \ge 0$. *Furthermore, let the conditions* (EVD), (EMB), (MOM) *be satisfied with* $0 < p \le \alpha \le 1$. *With* $0 \le \gamma \le 1$, *if* (SRC) *is satisfied with* $\gamma < \beta \le 2\rho$, *we have*

1. *in the case* $\beta + p \le \alpha$, *let* $\lambda_n = \Theta\left(\left(n/\log^\theta(n)\right)^{-\frac{1}{\alpha}}\right)$ *for some* $\theta > 1$, *for all* $\tau > \log(6)$ *and sufficiently large* $n \ge 1$, *there is a constant* $J > 0$ *independent of* $n$ *and* $\tau$ *such that*

$$\left\|[\hat{F}_{\lambda_n}] - F_*\right\|_\gamma^2 \le \tau^2 J \left(\frac{n}{\log^\theta n}\right)^{-\frac{\beta-\gamma}{\alpha}}$$

*is satisfied with* $P^n$-*probability not less than* $1 - 6e^{-\tau}$.

---

[6]Note that this induces a dataset $D = \{(x_i, \psi(z_i))\}_{i=1}^n$ where we identify $y_i = \psi(z_i)$.

2. *in the case* $\beta + p > \alpha$, *let* $\lambda_n = \Theta\left(n^{-\frac{1}{\beta+p}}\right)$, *for all* $\tau > \log(6)$ *and sufficiently large* $n \geq 1$, *there is a constant* $J > 0$ *independent of* $n$ *and* $\tau$ *such that*

$$\left\|[\hat{F}_{\lambda_n}] - F_*\right\|_{\gamma}^2 \leq \tau^2 J n^{-\frac{\beta-\gamma}{\beta+p}}$$

*is satisfied with* $P^n$-*probability not less than* $1 - 6e^{-\tau}$.

Theorem 4 provides the upper rate for vector-valued spectral algorithms. In particular, in combination with the lower bound in Theorem 2, we see that vector-valued spectral algorithms with qualification $\rho$ achieve an optimal learning rate when the smoothness $\beta$ of the regression function is in the range $(\alpha-p, 2\rho]$. For algorithms with infinite $\rho$ such as gradient descent and principal component regression, we confirm that they can exploit smoothness of the target function just as in the real-valued setting [3, 5, 30], while not suffering from saturation. For Tikhonov regularization, where $\rho = 1$, the rates recover the state-of-the-art results from [31]. Finally, we point out that obtaining minimax optimal learning rates for $\beta < \alpha - p$ still remains challenging even in the real-valued output scenario. Note however that for a large variety of RKHS, $\alpha$ is arbitrarily close to $p$ and we obtain optimal rates for the whole range $(0, 2\rho]$: we refer to [31, 54] for a detailed discussion.

We provide a proof sketch for Theorem 4. The key technical challenge in extending the results of [31] to spectral filter functions lies in the analysis of the estimation error. The estimation error in $\gamma$−norm is bounded as $\left\|[\hat{C}_\lambda - C_\lambda]\right\|_{S_2([\mathcal{H}]^\gamma, \mathcal{Y})} \leq 3\lambda^{-\frac{\gamma}{2}} \left\|(\hat{C}_\lambda - C_\lambda)\hat{C}_{X,\lambda}^{\frac{1}{2}}\right\|_{S_2(\mathcal{H}, \mathcal{Y})}$ (see Eq. (37) in Appendix C.3). We rely on the fact that $\mathrm{Id}_{\mathcal{H}} = \hat{C}_X g_\lambda(\hat{C}_X) + r_\lambda(\hat{C}_X)$ (see Definition 2), to obtain the decomposition $\hat{C}_\lambda - C_\lambda = (\hat{C}_{YX} - C_\lambda\hat{C}_X) g_\lambda(\hat{C}_X) - C_\lambda r_\lambda(\hat{C}_X)$, which yields two terms to be controlled,

$$\left\|(\hat{C}_\lambda - C_\lambda)\hat{C}_{X,\lambda}^{\frac{1}{2}}\right\|_{S_2(\mathcal{H}, \mathcal{Y})} \leq \underbrace{\left\|(\hat{C}_{YX} - C_\lambda\hat{C}_X)g_\lambda(\hat{C}_X)\hat{C}_{X,\lambda}^{\frac{1}{2}}\right\|_{S_2(\mathcal{H}, \mathcal{Y})}}_{(I)} + \underbrace{\left\|C_\lambda r_\lambda(\hat{C}_X)\hat{C}_{X,\lambda}^{\frac{1}{2}}\right\|_{S_2(\mathcal{H}, \mathcal{Y})}}_{(II)}$$

To control term (I), we use the definition of the filter function $g_\lambda$ (Definition 2) to obtain that $\left\|\hat{C}_{X,\lambda}g_\lambda(\hat{C}_X)\right\|_{\mathcal{H}\to\mathcal{H}} \lesssim 1$. Thus it suffices to control the term $\left\|(\hat{C}_{YX} - C_\lambda\hat{C}_X)C_{X,\lambda}^{-\frac{1}{2}}\right\|_{S_2(\mathcal{H}, \mathcal{Y})} = \left\|\frac{1}{n}\sum_{i=1}^n \xi(x_i, y_i)\right\|_{S_2(\mathcal{H}, \mathcal{Y})}$, where $\xi(x, y) = (y - C_\lambda\phi(x)) \otimes C_{X,\lambda}^{-\frac{1}{2}}\phi(x)$. We proceed by bounding $\mathbb{E}[\|\xi(X, X)\|_{S_2(\mathcal{H}, \mathcal{Y})}^m]$ for $m \geq 1$, and then use Bernstein's inequality to derive the upper bound on $\left\|(\hat{C}_{YX} - C_\lambda\hat{C}_X)C_{X,\lambda}^{-\frac{1}{2}}\right\|_{S_2(\mathcal{H}, \mathcal{Y})}$. To control term (II), Lemma 9 in Appendix C.1 shows that $(II) \lesssim \left\|\hat{C}_{X,\lambda}^{\frac{1}{2}}r_\lambda(\hat{C}_X)g_\lambda(C_X)C_X^{\frac{\beta+1}{2}}\right\|_{\mathcal{H}\to\mathcal{H}}$. The term on the right side is bounded in prior work on scalar-valued spectral method, and we refer the reader to [54, Theorem 16]. The results of Theorem 4 are then obtained by choosing regularization parameter $\lambda = \lambda(n)$ to optimally trade off approximation and estimation errors.

## 5 Conclusion

In this work, we have rigorously explored the theoretical properties of vector-valued spectral learning algorithms, focusing on their performance in infinite-dimensional output spaces. We first proved the saturation effect observed in vector-valued kernel ridge regression, highlighting its limitations in exploiting additional smoothness in regression functions. We then presented upper bounds on the finite sample risk for a general class of spectral learning algorithms, demonstrating their minimax optimality across various scenarios, including misspecified learning settings.

Our results open avenues for further research, particularly in developing more efficient implementations for practical use in high-dimensional machine learning problems such as causal inference and functional data analysis.

**Acknowledgement:** Dimitri Meunier, Arthur Gretton and Zhu Li were supported by the Gatsby Charitable Foundation.

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

# Appendices

The appendix is organized as follows. In Section A, we give additional mathematical background and notations. In Section B, we give the proof of Theorem 3 and provide a technical comparison of our proof with [29]. In Section C, we prove Theorem 4. Finally, in Section D, we provide auxiliary results used in the main proofs.

## A  Additional Background

### A.1  Hilbert spaces and linear operators

**Definition 3** (Bochner $L_q$–spaces, [14]). *Let $H$ be a separable Hilbert space and $\pi$ a probability measure on $\mathcal{X}$. For $1 \leq q \leq \infty$, $L_q(\mathcal{X}, \mathcal{F}_\mathcal{X}, \pi; H)$, abbreviated $L_q(\pi; H)$, is the space of strongly $\mathcal{F}_\mathcal{X} - \mathcal{F}_H$ measurable and Bochner $q$-integrable functions from $\mathcal{X}$ to $H$, with the norms*

$$\|f\|_{L_q(\pi;H)}^q = \int_\mathcal{X} \|f\|_H^q \, \mathrm{d}\pi, \quad 1 \leq q < \infty, \qquad \|f\|_{L_\infty(\pi;H)} = \inf\left\{C \geq 0 : \pi\{\|f\|_H > C\} = 0\right\}.$$

**Definition 4** ($p$-Schatten class, e.g. [52]). *Let $H, H'$ be separable Hilbert spaces. For $1 \leq q \leq \infty$, $S_p(H, H')$, abbreviated $S_p(H)$ if $H = H'$, is the Banach space of all compact operators $C$ from $H$ to $H'$ such that $\|C\|_{S_p(H,H')} := \|(\sigma_i(C))_{i \in I}\|_{\ell_p}$ is finite. Here $\|(\sigma_i(C))_{i \in I}\|_{\ell_p}$ is the $\ell_p$–sequence space norm of the sequence of the strictly positive singular values of $C$ indexed by the at most countable set $I$. For $p = 2$, we retrieve the space of Hilbert-Schmidt operators, for $p = 1$ we retrieve the space of Trace Class operators, and for $p = +\infty$, $\|\cdot\|_{S_\infty(H,H')}$ corresponds to the operator norm $\|\cdot\|_{H \to H'}$.*

**Definition 5** (Tensor Product of Hilbert Spaces, [1]). *Let $H, H'$ be Hilbert spaces. The Hilbert space $H \otimes H'$ is the completion of the algebraic tensor product with respect to the norm induced by the inner product $\langle x_1 \otimes x_1', x_2 \otimes x_2' \rangle_{H \otimes H'} = \langle x_1, x_2 \rangle_H \langle x_1', x_2' \rangle_{H'}$ for $x_1, x_2 \in H$ and $x_1', x_2' \in H'$ defined on the elementary tensors of $H \otimes H'$. This definition extends to $\mathrm{span}\{x \otimes x' | x \in H, x' \in H'\}$ and finally to its completion. The space $H \otimes H'$ is separable whenever both $H$ and $H'$ are separable. If $\{e_i\}_{i \in I}$ and $\{e_j'\}_{j \in J}$ are orthonormal basis in $H$ and $H'$, $\{e_i \otimes e_j'\}_{i \in I, j \in J}$ is an orthonormal basis in $H \otimes H'$.*

**Theorem 5** (Isometric Isomorphism between $L_2(\pi; \mathcal{Y})$ and $S_2(L_2(\pi), \mathcal{Y})$, Theorem 12.6.1 [1]). *Let $H$ be a separable Hilbert space. The Bochner space $L_2(\pi; H)$ is isometrically isomorphic to $S_2(L_2(\pi), \mathcal{Y})$ and the isometric isomorphism is realized by the map $\Psi : S_2(L_2(\pi), \mathcal{Y}) \to L_2(\pi; H)$ acting on elementary tensors as $\Psi(f \otimes y) = (\omega \to f(\omega)y)$.*

### A.2  RKHS embbedings into $L_2$ and Well-speciﬁedness

Recall that $I_\pi : \mathcal{H} \to L_2(\pi)$ is the embedding that maps every function in $\mathcal{H}$ into its $\pi$-equivalence class in $L_2(\pi)$ and that we used the shorthand notation $[f] = I_\pi(f)$ for all $f \in \mathcal{H}$. We define similarly $\mathcal{I}_\pi : \mathcal{G} \to L_2(\pi; \mathcal{Y})$ as the embedding that maps every function in $\mathcal{G}$ into its $\pi$-equivalence class in $L_2(\pi; \mathcal{Y})$.

**Definition 6** (Embedding $\mathcal{G}$ into $L_2(\pi; \mathcal{Y})$). Let $\mathcal{I}_\pi := I_\mathcal{Y} \otimes I_\pi$ be the tensor product of the operator $\mathrm{Id}_\mathcal{Y}$ with the operator $I_\pi$ (see [1, Definition 12.4.1.] for the definition of tensor product of operators). $\mathcal{I}_\pi$ maps every function in $\mathcal{G}$ into its $\pi$-equivalence class in $L_2(\pi; \mathcal{Y})$. We then use the shorthand notation $[F] = \mathcal{I}_\pi(F)$ for all $F \in \mathcal{G}$.

**Remark 5.** *Let $\{d_j\}_{j \in J}$ be an orthonormal basis of $\mathcal{Y}$ and recall that $\{\sqrt{\mu_i}[e_i]\}_{i \in I}$ forms an orthonormal basis of $[\mathcal{H}]^1$. Let $F \in [\mathcal{G}]^1$. Then $F$ can be represented as the element $C := \sum_{i \in I, j \in J} a_{ij} d_j \otimes \sqrt{\mu_i}[e_i]$ in $S_2([\mathcal{H}]^1, \mathcal{Y})$ by definition of $[\mathcal{G}]^1$ with $\|C\|_1^2 = \sum_{i,j} a_{ij}^2$. Hence defining $\bar{C} := \sum_{i \in I, j \in J} a_{ij} d_j \otimes \sqrt{\mu_i} e_i$ we have $C = \bar{C}$ $\pi$−a.e. and*

$$\|\bar{C}\|_\mathcal{G}^2 = \sum_{i \in I, j \in J} a_{i,j}^2 = \|C\|_1^2 < +\infty.$$

*Taking the elements identifying $\bar{C}$ in $\mathcal{G}$ gives a representer $\bar{F}$ of $F$ in $\mathcal{G}$.*

### A.3 Additional Notations

In the following, we fix $\{d_j\}_{j \in J}$ an orthonormal basis of $\mathcal{Y}$, where $J$ is at most countable. Recall that $\left\{ \mu_i^{1/2} e_i \right\}_{i \in I}$ is an ONB of $(\ker I_\pi)^\perp$ in $\mathcal{H}$, and $\{[e_i]\}_{i \in I}$ is an ONB of $\overline{\mathrm{ran}\, I_\pi}$ in $L_2(\pi)$. Let $\{\tilde{e}_i\}_{i \in I'}$ be an ONB of $\ker I_\pi$ (with $I \cap I' = \varnothing$), then $\left\{ \mu_i^{1/2} e_i \right\}_{i \in I} \cup \{\tilde{e}_i\}_{i \in I'}$ forms an ONB of $\mathcal{H}$, and $\left\{ d_j \otimes \mu_i^{1/2} e_i \right\}_{i \in I, j \in J} \cup \{d_j \otimes \tilde{e}_i\}_{i \in I', j \in J}$ forms an ONB of $\mathcal{Y} \otimes \mathcal{H} \simeq \mathcal{G}$.

For any Hilbert space $H$, linear operator $T : H \to H$ and scalar $\lambda > 0$, we define $T_\lambda := T + \lambda I_H$.

## B Saturation Effect with Tikhonov Regularization - Proof of Theorem 3

In the following proofs a quantity $h_n \geq 0$ depending on $n \geq 1$, but independent of $\tau$ the confidence level, is equal to $o(1)$ if $h_n \to 0$ when $n \to +\infty$.

We will make extensive use of the following notation in the subsequent analysis.

**Definition 7** (Empirical $L_2(\pi)-$norm)**.** *Denoted by $\langle \cdot, \cdot \rangle_{2,n}$, the empirical $L_2(\pi)-$norm associated to points $\{x_i\}_{i=1}^n$ independently and identically sampled from the distribution of $X$, is defined as, for any $f, g \in \mathcal{H}$,*

$$\langle f, g \rangle_{2,n} := \left\langle \hat{C}_X, f \otimes g \right\rangle_{S_2(\mathcal{H})} = \left\langle \hat{C}_X f, g \right\rangle_{\mathcal{H}} = \left\langle \hat{C}_X^{\frac{1}{2}} f, \hat{C}_X^{\frac{1}{2}} g \right\rangle_{\mathcal{H}} = \frac{1}{n} \sum_{i=1}^n f(x_i) g(x_i).$$

*This induces an inner product on $\mathcal{H}$, with associated norm,*

$$\|f\|_{2,n}^2 = \langle f, f \rangle_{2,n} = \frac{1}{n} \sum_{i=1}^n f(x_i)^2.$$

**Definition 8.** *Fix $x \in \mathcal{X}$ and $\lambda > 0$. The regularized canonical feature map is defined as*

$$f_{x,\lambda}(\cdot) = C_{X,\lambda}^{-1} k(x, \cdot) : \mathcal{X} \to \mathcal{H}.$$

Recall from Eq. (11) that the ridge estimator $\hat{F}_\lambda$ defined in Eq. (7) can be expressed as

$$\hat{C}_\lambda = \hat{C}_{YX} g_\lambda(\hat{C}_X), \qquad \hat{F}_\lambda(\cdot) = \hat{C}_\lambda \phi(\cdot) \in \mathcal{G},$$

where in Theorem 3 we focus on Tikhonov regularization where $g_\lambda(x) = (x + \lambda)^{-1}$. In that setting we have

$$r_\lambda(x) := 1 - \frac{x}{x + \lambda} = -\frac{\lambda}{x + \lambda}. \tag{13}$$

*Proof of Theorem 3.* Since $\beta \geq 2$, $F_* \in [\mathcal{G}]^\beta \subseteq [\mathcal{G}]^1$, therefore $F_*$ has a representer $\bar{F}$ in $\mathcal{G}$ such that $F_* = \bar{F}$ $\pi$-a.e. (see Remark 5), and by Theorem 1, $\bar{F}(\cdot) = \bar{C}\phi(\cdot)$, with $\bar{C} \in S_2(\mathcal{H}, \mathcal{Y})$. Define the errors $\epsilon_i := y_i - \bar{C}\phi(x_i)$, $i = 1, \ldots, n$, that are i.i.d samples with the same distribution as $\epsilon := Y - \bar{C}\phi(X)$. By assumption $\mathbb{E}\left[ \|\epsilon\|_{\mathcal{Y}}^2 \mid X \right] \geq \sigma^2$ and by definition $\mathbb{E}[\epsilon \mid X] = 0$. By Eq. (13), we have

$$r_\lambda\left(\hat{C}_X\right) := I - \hat{C}_X \hat{C}_{X,\lambda}^{-1} = -\lambda \hat{C}_{X,\lambda}^{-1}.$$

The following *bias-variance decomposition* is the essence of the proof. In the following derivation we abbreviate $S_2(L_2(\pi), \mathcal{Y})$ to $S_2\,L_2(\pi; \mathcal{Y})$ to $L_2$ and $x_1, \ldots, x_n$ to $\underline{x}_n$ to save space.

$$\mathbb{E}\left[\left\|[\hat{F}_\lambda] - F_*\right\|^2_{L_2} \mid \underline{x}_n\right] = \mathbb{E}\left[\left\|[\hat{C}_{YX}\hat{C}^{-1}_{X,\lambda} - \bar{C}]\right\|^2_{S_2} \mid \underline{x}_n\right]$$

$$= \mathbb{E}\left[\left\|\left[\left(\frac{1}{n}\sum_{i=1}^{n} y_i \otimes \phi(x_i)\right)\hat{C}^{-1}_{X,\lambda} - \bar{C}\right]\right\|^2_{S_2} \mid \underline{x}_n\right]$$

$$= \mathbb{E}\left[\left\|\left[\frac{1}{n}\sum_{i=1}^{n}\left(\bar{C}\phi(x_i) + \epsilon_i\right) \otimes \phi(x_i)\hat{C}^{-1}_{X,\lambda} - \bar{C}\right]\right\|^2_{S_2} \mid \underline{x}_n\right]$$

$$= \mathbb{E}\left[\left\|\left[-\bar{C}r_\lambda\left(\hat{C}_X\right) + \frac{1}{n}\sum_{i=1}^{n}\epsilon_i \otimes \left(\hat{C}^{-1}_{X,\lambda}\phi(x_i)\right)\right]\right\|^2_{S_2} \mid \underline{x}_n\right]$$

$$= \left\|\left[\bar{C}r_\lambda\left(\hat{C}_X\right)\right]\right\|^2_{S_2} + \frac{1}{n^2}\sum_{i=1}^{n}\mathbb{E}\left[\|\epsilon_i\|^2_{\mathcal{Y}} \mid x_i\right]\left\|\left[\hat{C}^{-1}_{X,\lambda}\phi(x_i)\right]\right\|^2_{L_2(\pi)}$$

$$\geq \lambda^2\left\|\left[\bar{C}\hat{C}^{-1}_{X,\lambda}\right]\right\|^2_{S_2} + \frac{\sigma^2}{n^2}\sum_{i=1}^{n}\left\|\left[\hat{C}^{-1}_{X,\lambda}\phi(x_i)\right]\right\|^2_{L_2(\pi)}.$$

The second term is a lower bound on the *variance* while the first term is a lower bound on the *bias*.

**Bounding the Bias term.** The idea is to first show that the population analogue of $\left\|\left[\bar{C}\hat{C}^{-1}_{X,\lambda}\right]\right\|^2_{S_2(L_2(\pi), \mathcal{Y})}$ can be bounded below by a non-zero constant. We can then bound the difference between the empirical and population version of $\left\|\left[\bar{C}\hat{C}^{-1}_{X,\lambda}\right]\right\|^2_{S_2(L_2(\pi), \mathcal{Y})}$ using a concentration inequality. By Lemma 1, for $\lambda > 0$, there is a constant $c > 0$ (see Lemma 1 for the exact value of $c$) such that

$$\left\|\left[\bar{C}C^{-1}_{X,\lambda}\right]\right\|^2_{S_2(L_2(\pi), \mathcal{Y})} \geq c > 0.$$

Furthermore by Lemma 2, there is a constant $c_0 > 0$ (see Lemma 2 for the exact value of $c_0$) such that for any $\tau \geq \log(4)$, with probability at least $1 - 4e^{-\tau}$, for $n \geq (c_0\tau)^{(4+2p)}$ and $1 \geq \lambda \geq n^{-\frac{1}{2+p}}$, we have

$$\left|\left\|\left[\bar{C}C^{-1}_{X,\lambda}\right]\right\|^2_{S_2(L_2(\pi), \mathcal{Y})} - \left\|\left[\bar{C}\hat{C}^{-1}_{X,\lambda}\right]\right\|^2_{S_2(L_2(\pi), \mathcal{Y})}\right| = \tau^2 o(1).$$

Therefore, under the same high probability,

$$\left\|\left[\bar{C}\hat{C}^{-1}_{X,\lambda}\right]\right\|^2_{S_2(L_2(\pi), \mathcal{Y})} \geq c - \tau^2 o(1).$$

It leads to our final bound on the bias term, for a constant $\rho_2 \geq 0$ and for sufficiently large $n \geq 1$, where the hidden index bound depends on $\tau$, we have

$$\lambda^2\left\|\left[\bar{C}\hat{C}^{-1}_{X,\lambda}\right]\right\|^2_{S_2(L_2(\pi), \mathcal{Y})} \geq \rho_1\lambda^2. \tag{14}$$

**Bounding the Variance Term.** Using the norm from Definition 7, we have the following chain of identities.

$$\frac{\sigma^2}{n^2}\sum_{i=1}^{n}\left\|\left[\hat{C}^{-1}_{X,\lambda}\phi(x_i)\right]\right\|^2_{L_2(\pi)} = \frac{\sigma^2}{n^2}\sum_{i=1}^{n}\int_{\mathcal{X}}\left\langle\phi(X), \hat{C}^{-1}_{X,\lambda}\phi(x_i)\right\rangle^2_{\mathcal{H}} d\pi(x)$$

$$= \frac{\sigma^2}{n}\int_{\mathcal{X}}\|\hat{C}^{-1}_{X,\lambda}\phi(X)\|^2_{2,n} d\pi(x). \tag{15}$$

Therefore it suffices to consider $\int_{\mathcal{X}}\|\hat{C}^{-1}_{X,\lambda}k(x, \cdot)\|^2_{2,n}d\pi(x)$.

Combining the result of Lemma 4 and Lemma 5 we obtain that for $1 \geq \lambda \geq n^{-\frac{1}{2+p}}$ with probability at least $1 - 6e^{-\tau}$, for $n \geq (c_0\tau)^{4+2p}$, the following bounds hold simultaneously for all $x \in \mathcal{X}$:

$$\left\|\hat{C}^{\frac{1}{2}}_X\left(\hat{C}^{-1}_{X,\lambda} - C^{-1}_{X,\lambda}\right)k(x, \cdot)\right\|_{\mathcal{H}} \leq \tau o(1)$$

$$\|[C^{-1}_{X,\lambda}k(x, \cdot)]\|^2_{2,n} - \frac{1}{2}\|[C^{-1}_{X,\lambda}k(x, \cdot)]\|^2_{L_2(\pi)} \geq -\tau o(1)$$

$$\|[C^{-1}_{X,\lambda}k(x, \cdot)]\|^2_{2,n} - \frac{3}{2}\|[C^{-1}_{X,\lambda}k(x, \cdot)]\|^2_{L_2(\pi)} \leq \tau o(1).$$

Fix $x \in \mathcal{X}$. Using the algebraic identity $a^2 - b^2 = (a-b)(2b + (a-b))$, and recalling that by Definition 7,

$$\|f\|_{2,n}^2 = \left\|\hat{C}_X^{\frac{1}{2}} f\right\|_{\mathcal{H}}^2,$$

we deduce

$$\left| \left\|\hat{C}_X^{\frac{1}{2}} \hat{C}_{X,\lambda}^{-1} k(x,\cdot)\right\|_{\mathcal{H}}^2 - \left\|\hat{C}_X^{\frac{1}{2}} C_{X,\lambda}^{-1} k(x,\cdot)\right\|_{\mathcal{H}}^2 \right|$$

$$\leq \left\|\hat{C}_X^{\frac{1}{2}} \left(\hat{C}_{X,\lambda}^{-1} - C_{X,\lambda}^{-1}\right) k(x,\cdot)\right\|_{\mathcal{H}} \cdot \left(\left\|\hat{C}_X^{\frac{1}{2}} \left(\hat{C}_{X,\lambda}^{-1} - C_{X,\lambda}^{-1}\right) k(x,\cdot)\right\|_{\mathcal{H}} + 2\left\|C_{X,\lambda}^{-1} k(x,\cdot)\right\|_{2,n}\right)$$

$$\leq \tau o(1) \left(\tau o(1) + 2\left\|C_{X,\lambda}^{-1} k(x,\cdot)\right\|_{2,n}\right).$$

Using Definition 7 again, this reads

$$\left\|\hat{C}_{X,\lambda}^{-1} k(x,\cdot)\right\|_{2,n}^2 \geq \left\|C_{X,\lambda}^{-1} k(x,\cdot)\right\|_{2,n}^2 - \tau o(1) \left(\tau o(1) + 2\left\|C_{X,\lambda}^{-1} k(x,\cdot)\right\|_{2,n}\right).$$

We have

$$\|C_{X,\lambda}^{-1} k(x,\cdot)]\|_{2,n}^2 \leq \frac{3}{2} \|[C_{X,\lambda}^{-1} k(x,\cdot)]\|_{L_2(\pi)}^2 + \tau o(1) \leq \left(\sqrt{1.5} \|[C_{X,\lambda}^{-1} k(x,\cdot)]\|_{L_2(\pi)} + \sqrt{\tau} o(1)\right)^2.$$

Hence,

$$\left\|\hat{C}_{X,\lambda}^{-1} k(x,\cdot)\right\|_{2,n}^2 \geq \left\|C_{X,\lambda}^{-1} k(x,\cdot)\right\|_{2,n}^2 - \tau o(1) \left(\|[C_{X,\lambda}^{-1} k(x,\cdot)]\|_{L_2(\pi)} + \sqrt{\tau} o(1) + \tau o(1)\right)$$

$$\geq \frac{1}{2} \|[C_{X,\lambda}^{-1} k(x,\cdot)]\|_{L_2(\pi)}^2 - \tau o(1)$$

$$- \tau o(1) \left(\|[C_{X,\lambda}^{-1} k(x,\cdot)]\|_{L_2(\pi)} + \sqrt{\tau} o(1) + \tau o(1)\right)$$

$$\geq \frac{1}{2} \|[C_{X,\lambda}^{-1} k(x,\cdot)]\|_{L_2(\pi)}^2 - \tau^2 o(1) - \tau o(1) \|[C_{X,\lambda}^{-1} k(x,\cdot)]\|_{L_2(\pi)}.$$

By Lemma 17,

$$\int_{\mathcal{X}} \left\|[C_{X,\lambda}^{-1} k(x,\cdot)]\right\|_{L_2(\pi)}^2 d\pi(x) = \mathcal{N}_2(\lambda).$$

Furthermore, by Jensen's inequality,

$$\int_{\mathcal{X}} \|[C_{X,\lambda}^{-1} k(x,\cdot)]\|_{L_2(\pi)}^2 d\pi(x) \geq \left(\int_{\mathcal{X}} \|[C_{X,\lambda}^{-1} k(x,\cdot)]\|_{L_2(\pi)} d\pi(x)\right)^2.$$

Recall from Lemma 16 that

$$c_{1,2} \lambda^{-p} \leq \mathcal{N}_2(\lambda) \leq c_{2,2} \lambda^{-p}.$$

Therefore we have

$$\int_{\mathcal{X}} \|[C_{X,\lambda}^{-1} k(x,\cdot)]\|_{L_2(\pi)} d\pi(x) \leq \sqrt{c_{2,2}} \lambda^{-\frac{p}{2}}$$

$$\int_{\mathcal{X}} \|[C_{X,\lambda}^{-1} k(x,\cdot)]\|_{L_2(\pi)}^2 d\pi(x) \geq c_{1,2} \lambda^{-p}.$$

Hence

$$\int_{\mathcal{X}} \|\hat{C}_{X,\lambda}^{-1} k(x,\cdot)\|_{2,n}^2 d\pi(x) \geq \frac{c_{1,2}}{2} \lambda^{-p} - \tau^2 o(1) - \tau o(1) \sqrt{c_{2,2}} \lambda^{-\frac{p}{2}}$$

$$\geq \left(\frac{c_{1,2}}{2} - \tau o(1) \sqrt{c_{2,2}}\right) \lambda^{-p} - \tau^2 o(1)$$

Combined with Eq. (15), it leads to our final bound on the variance term, for a constant $\rho_2 \geq 0$ and for sufficiently large $n \geq 1$, where the hidden index bound depends on $\tau$, we have

$$\frac{\sigma^2}{n^2} \sum_{i=1}^{n} \left\|\left[\hat{C}_{X,\lambda}^{-1} \phi(x_i)\right]\right\|_{L_2(\pi)}^2 \geq \frac{\rho_2}{n\lambda^p}. \tag{16}$$

**Putting it together.** We are now ready to assemble the lower bounds on the variance and on the bias. For a fixed confidence parameter $\tau \geq \log(10)$, for sufficiently large $n > 0$, where the hidden index

bound depends on $\tau$, with probability at least $1 - 10e^{-\tau}$, we have by Eq. (14) and Eq. (16), that for $\lambda = \lambda(n)$ satisfying $1 \geq \lambda \geq n^{-\frac{1}{2+p}}$,

$$\mathbb{E}\left[\left\|[\hat{F}_\lambda] - F_*\right\|^2_{L_2(\pi;\mathcal{Y})} \mid x_1, \ldots, x_n\right] \geq \rho_1 \lambda^2 + \rho_2 n^{-1} \lambda^{-p}$$

where $\rho_1, \rho_2$ have no dependence on $n$. Recall Young's inequality, for $r, q > 1$ satisfying $r^{-1} + q^{-1} = 1$, we have for all $a, b \geq 0$,

$$a + b \geq r^{\frac{1}{r}} q^{\frac{1}{q}} a^{\frac{1}{r}} b^{\frac{1}{q}}.$$

We apply Young's inequality with $r^{-1} = p/(2+p)$ and $q^{-1} = 2/(2+p)$, there exists a constant $c_1 > 0$ such that

$$\rho_1 \lambda^2 + \rho_2 n^{-1} \lambda^{-p} \geq c_1 \left(\lambda^2\right)^{\frac{p}{2+p}} \left(\lambda^{-p} n^{-1}\right)^{\frac{2}{2+p}} = c_1 n^{-\frac{2}{2+p}}.$$

To conclude the proof, let $\lambda = \lambda(n)$ be an arbitrary choice of regularization parameter satisfying $\lambda(n) \to 0$. We have just covered the case $1 \geq \lambda \geq n^{-\frac{1}{2+p}}$ and the case $0 < \lambda \leq n^{-\frac{1}{2+p}}$ is covered by [29, Section B.4]. $\qquad\square$

**Lemma 1.** *For any $\lambda \leq 1$ and $C \in S_2(\mathcal{H}, \mathcal{Y})$, with $C \perp\!\!\!\perp S_2(\overline{ran\ S_\pi}, \mathcal{Y})$[7], we have*

$$\left\|[CC^{-1}_{X,\lambda}]\right\|^2_{S_2(L_2(\pi),\mathcal{Y})} \geq \sum_{i \in I, j \in J} a^2_{ij} \frac{\mu_i}{(\mu_i + 1)^2} > 0,$$

*with $a_{ij} := \langle d_j, C\sqrt{\mu_i}e_i\rangle_\mathcal{Y}$, $i \in I, j \in J$.*

*Proof.* Recell the notations of Section A.3. Define $\{a_{ij}\}_{i \in I \cap I', j \in J}$ such that $a_{ij} := \langle d_j, C\sqrt{\mu_i}e_i\rangle_\mathcal{Y}$ for $i \in I, j \in J$ and $a_{ij} := \langle d_j, C\tilde{e}_i\rangle_\mathcal{Y}$ for $i \in I', j \in J$. Then, on one hand, since $C \in S_2(\mathcal{H}, \mathcal{Y})$,

$$C = \sum_{i \in I, j \in J} a_{ij} d_j \otimes (\sqrt{\mu_i}e_i) + \sum_{i \in I', j \in J} a_{ij} d_j \otimes \tilde{e}_i.$$

On the other hand,

$$C^{-1}_{X,\lambda} = \sum_{i \in I}(\mu_i + \lambda)^{-1}(\sqrt{\mu_i}e_i) \otimes (\sqrt{\mu_i}e_i) + \lambda^{-1} \sum_{i \in I'} \tilde{e}_i \otimes \tilde{e}_i.$$

Therefore, noting that $\tilde{e}_i = 0$ $\pi$−a.e. for all $i \in I'$, we have,

$$[CC^{-1}_{X,\lambda}] = \left[\sum_{i \in I, j \in J} a_{ij}(\mu_i + \lambda)^{-1} d_j \otimes (\sqrt{\mu_i}e_i) + \sum_{i \in I', j \in J} \frac{a_{ij}}{\lambda} d_j \otimes \tilde{e}_i\right]$$

$$= \sum_{i \in I, j \in J} a_{ij} \frac{\sqrt{\mu_i}}{\mu_i + \lambda} d_j \otimes [e_i].$$

Therefore the $S_2(L_2(\pi), \mathcal{Y})$-norm can be evaluated in closed form using Parseval's identity,

$$\left\|[CC^{-1}_{X,\lambda}]\right\|^2_{S_2(L_2(\pi),\mathcal{Y})} = \sum_{i \in I, j \in J} a^2_{ij} \frac{\mu_i}{(\mu_i + \lambda)^2} \geq \sum_{i \in I, j \in J} a^2_{ij} \frac{\mu_i}{(\mu_i + 1)^2},$$

where we used that $\{d_j \otimes [e_i]\}_{j \in J, i \in I}$ is orthonormal in $\mathcal{Y} \otimes L_2(\pi)$, and $\lambda \leq 1$. The right hand side has no dependence on $\lambda$ or $n$. Furthermore, under assumption (EVD+), $\mu_i > 0$ for all $i \in I$, therefore the right hand side term equals zero if and only if $a_{ij} = 0$ for all $i \in I, j \in J$. Since by assumption $C \perp\!\!\!\perp S_2(\overline{ran\ S_\pi}, \mathcal{Y})$, the right hand side is strictly positive. $\qquad\square$

**Lemma 2.** *Suppose Assumption (EVD) holds with $p \in (0, 1]$. Let $C \in S_2(\mathcal{H}, \mathcal{Y})$ such that $[C] \in S_2([\mathcal{H}]^2, \mathcal{Y})$. There is a constant $c_0 > 0$ such that for any $\tau \geq \log(4)$, with probability at least $1 - 4e^{-\tau}$, for $n \geq (c_0 \tau)^{(4+2p)}$ and $1 \geq \lambda \geq n^{-\frac{1}{2+p}}$, we have*

$$\left|\left\|[CC^{-1}_{X\lambda}]\right\|^2_{S_2(L_2(\pi),\mathcal{Y})} - \left\|[C\hat{C}^{-1}_{X\lambda}]\right\|^2_{S_2(L_2(\pi),\mathcal{Y})}\right| \leq \tau^2 o(1)$$

*We have $c_0 := 8\kappa \max\{\sqrt{c_{2,1}}, 1\}$) where $c_{2,1}$ is defined in Lemma 16.*

---
[7] $\perp\!\!\!\perp$ is the notation for "not being orthogonal to".

*Proof.* Using the identity $A^{-1} - B^{-1} = A^{-1}(B - A)B^{-1}$, we obtain

$$C_{X,\lambda}^{-1} - \hat{C}_{X,\lambda}^{-1} = C_{X,\lambda}^{-1}(\hat{C}_X - C_X)\hat{C}_{X,\lambda}^{-1}$$

We apply Lemma 22 with $\gamma = 0$,

$$
\begin{aligned}
\left\| \left[ C \left( C_{X,\lambda}^{-1} - \hat{C}_{X,\lambda}^{-1} \right) \right] \right\|_{S_2(L_2(\pi),\mathcal{Y})} &= \left\| \left[ C\hat{C}_{X,\lambda}^{-1}(\hat{C}_X - C_X)C_{X,\lambda}^{-1} \right] \right\|_{S_2(L_2(\pi),\mathcal{Y})} \\
&= \left\| C\hat{C}_{X,\lambda}^{-1}(\hat{C}_X - C_X)C_{X,\lambda}^{-1}C_X^{\frac{1}{2}} \right\|_{S_2(\mathcal{H},\mathcal{Y})} \\
&\leq \left\| C\hat{C}_{X,\lambda}^{-\frac{1}{2}} \right\|_{S_2(\mathcal{H},\mathcal{Y})} \left\| \hat{C}_{X,\lambda}^{-\frac{1}{2}}C_{X,\lambda}^{\frac{1}{2}} \right\|_{\mathcal{H}\to\mathcal{H}} \\
&\quad \cdot \left\| C_{X,\lambda}^{-\frac{1}{2}}(\hat{C}_X - C_X)C_{X,\lambda}^{-\frac{1}{2}} \right\|_{\mathcal{H}\to\mathcal{H}} \left\| C_{X,\lambda}^{-\frac{1}{2}}C_X^{\frac{1}{2}} \right\|_{\mathcal{H}\to\mathcal{H}} \quad (17)
\end{aligned}
$$

We consider each of the four terms in line (17). The last term is bounded above by 1 and the first term is bounded above by $\lambda^{-\frac{1}{2}}\|C\|_{S_2(\mathcal{H},\mathcal{Y})}$. By Lemma 20 applied with $s = 1/2$, we have for the second term

$$\left\| \hat{C}_{X,\lambda}^{-\frac{1}{2}}C_{X,\lambda}^{\frac{1}{2}} \right\|_{\mathcal{H}\to\mathcal{H}} \leq \left\| \hat{C}_{X,\lambda}^{-1}C_{X,\lambda} \right\|_{\mathcal{H}\to\mathcal{H}}^{\frac{1}{2}}.$$

Then, by Lemma 18, for $\tau \geq \log(2)$, with probability at least $1 - 2e^{-\tau}$, for $\sqrt{n\lambda} \geq 8\tau\kappa\sqrt{\max\{\mathcal{N}(\lambda),1\}}$, we have

$$\left\| \hat{C}_{X,\lambda}^{-1}C_{X,\lambda} \right\|_{\mathcal{H}\to\mathcal{H}} \leq 2.$$

Since $\mathcal{N}(\lambda) \leq c_{2,1}\lambda^{-p}$ by Lemma 16, and $\lambda \leq 1$, it suffices to verify that $\lambda$ satisfies

$$\sqrt{n\lambda} \geq 8\tau\kappa\max\{\sqrt{c_{2,1}},1\}\lambda^{-\frac{p}{2}}.$$

Since $\lambda \geq n^{-\frac{1}{2+p}}$ by assumption, we deduce the sufficient condition $n \geq (\tau c_0)^{2(2+p)}$, where $c_0 := 8\kappa\max\{\sqrt{c_{2,1}},1\}$.

We bound the third term using Lemma 16 [33]. For $\tau \geq \log(2)$, with probability at least $1 - 2e^{-\tau}$, we have

$$\lambda^{-\frac{1}{2}}\left\| C_{X,\lambda}^{-\frac{1}{2}}(C_X - \hat{C}_X)C_{X,\lambda}^{-\frac{1}{2}} \right\|_{\mathcal{H}\to\mathcal{H}} \leq \frac{4\kappa^2\xi_\delta}{3n\lambda^{\frac{3}{2}}} + \sqrt{\frac{2\kappa^2\xi_\delta}{n\lambda^2}},$$

where we define

$$\xi_\delta := \log\frac{2\kappa^2(\mathcal{N}_1(\lambda) + 1)}{e^{-\tau}\|C_X\|_{\mathcal{H}\to\mathcal{H}}}.$$

By assumption $\lambda \geq n^{-\frac{1}{2+p}}$. We thus have

$$n\lambda^{\frac{3}{2}} \geq n^{\frac{1+2p}{4+2p}} \qquad \text{and} \qquad n\lambda^2 \geq n^{\frac{p}{2+p}}.$$

On the other hand, since $1 \geq \lambda \geq n^{-\frac{1}{2+p}}$, using Lemma 16, we have

$$\xi_\delta \leq \log\frac{82(c_{2,1}\lambda^{-p} + 1)}{e^{-\tau}\|C_X\|_{\mathcal{H}\to\mathcal{H}}} \leq \log\frac{2(c_{2,1} + 1)n^{\frac{p}{2+p}}}{e^{-\tau}\|C_X\|_{\mathcal{H}\to\mathcal{H}}} \leq \log\frac{2(c_{2,l} + 1)}{e^{-\tau}\|C_X\|_{\mathcal{H}\to\mathcal{H}}} + \frac{p}{2+p}\log n.$$

The first term does not depend on $n$, and the second term is logarithmic in $n$. Putting everything together with a union bound, we get a bound on (17). With probability at least $1 - 4e^{-\tau}$, for $n \geq (c_0\tau)^{(4+2p)}$, we have

$$\left\| \left[ C \left( C_{X,\lambda}^{-1} - \hat{C}_{X,\lambda}^{-1} \right) \right] \right\|_{S_2(L_2(\pi),\mathcal{Y})} \leq \|C\|_{S_2(\mathcal{H},\mathcal{Y})}\sqrt{2}\left( \frac{4\xi_\delta}{3n^{\frac{0.5+p}{2+p}}} + \sqrt{\frac{2\xi_\delta}{n^{\frac{p}{2+p}}}} \right) = \tau o(1)$$

The derivations in the proof of Lemma 1 show that

$$[CC_{X,\lambda}^{-1}] = \sum_{i\in I, j\in J} a_{ij}\frac{\sqrt{\mu_i}}{\mu_i + \lambda}d_j \otimes [e_i],$$

with $a_{ij} := \langle d_j, C\sqrt{\mu_i}e_i \rangle_{\mathcal{Y}}$, $i \in I, j \in J$. Note that since $[C] \in S_2([\mathcal{H}]^2, \mathcal{Y})$, we have

$$\|[C]\|^2_{S_2([\mathcal{H}]^2,\mathcal{Y})} = \left\| \sum_{i \in I, j \in J} a_{ij} d_j \otimes (\sqrt{\mu_i}e_i) \right\|^2_{S_2([\mathcal{H}]^2,\mathcal{Y})} = \sum_{i \in I, j \in J} \frac{a_{ij}^2}{\mu_i} < +\infty.$$

Hence,

$$\left\| [CC_{X,\lambda}^{-1}] \right\|^2_{S_2(L_2(\pi),\mathcal{Y})} = \sum_{i \in I, j \in J} a_{ij}^2 \frac{\mu_i}{(\mu_i + \lambda)^2} \leq \sum_{i \in I, j \in J} \frac{a_{ij}^2}{\mu_i} = \|[C]\|^2_{S_2([\mathcal{H}]^2,\mathcal{Y})} < +\infty. \qquad (18)$$

Using the equality $a^2 - b^2 = (a-b)(a+b)$ and the reverse triangular inequality, we obtain the following bound, with probability at least $1 - 4e^{-\tau}$, for $n \geq (c_0\tau)^{(4+2p)}$,

$$\left| \left\| [CC_{X,\lambda}^{-1}] \right\|^2_{S_2(L_2(\pi),\mathcal{Y})} - \left\| [C\hat{C}_{X,\lambda}^{-1}] \right\|^2_{S_2(L_2(\pi),\mathcal{Y})} \right|$$

$$\leq \left\| [C(C_{X,\lambda}^{-1} - \hat{C}_{X,\lambda}^{-1})] \right\|_{S_2(L_2(\pi),\mathcal{Y})} \left( \left\| [CC_{X,\lambda}^{-1}] \right\|_{S_2(L_2(\pi),\mathcal{Y})} + \left\| [C\hat{C}_{X,\lambda}^{-1}] \right\|_{S_2(L_2(\pi),\mathcal{Y})} \right)$$

$$\leq \tau o(1) \left( 2 \left\| [CC_{X,\lambda}^{-1}] \right\|_{S_2(L_2(\pi),\mathcal{Y})} + \left\| [C(C_{X,\lambda}^{-1} - \hat{C}_{X,\lambda}^{-1})] \right\|_{S_2(L_2(\pi),\mathcal{Y})} \right)$$

$$\leq \tau o(1) \left( 2\|[C]\|_{S_2([\mathcal{H}]^2,\mathcal{Y})} + \tau o(1) \right)$$

$$= \tau^2 o(1),$$

where in the second last line we used Equation (18). $\qquad \square$

**Lemma 3.** *Fix $x \in \mathcal{X}$ and $f_{x,\lambda}$ as in Definition 8. For $\tau \geq \log(2)$, with probability at least $1 - 2e^{-\tau}$ (note that this event depends on $x$),*

$$\left| \|f_{x,\lambda}\|^2_{2,n} - \|[f_{x,\lambda}]\|^2_{L_2(\pi)} \right| \leq \frac{1}{2}\|[f_{x,\lambda}]\|^2_{L_2(\pi)} + \frac{5\tau\kappa^2}{3\lambda^2 n}.$$

*Proof.* We start with

$$\|f_{x,\lambda}\|_\infty \leq \kappa\|f_{x,\lambda}\|_{\mathcal{H}} \leq \kappa^2\lambda^{-1}.$$

We apply Proposition 3 to $f = f_{x,\lambda}$, with $M = \kappa^2\lambda^{-1}$. For $\tau \geq \log(2)$, with probability at least $1 - 2e^{-\tau}$,

$$\left| \|f_{x,\lambda}\|^2_{2,n} - \|[f_{x,\lambda}]\|^2_{L_2(\pi)} \right| \leq \frac{1}{2}\|[f_{x,\lambda}]\|^2_{L_2(\pi)} + \frac{5\tau\kappa^2}{3\lambda^2 n}.$$

$\qquad \square$

**Lemma 4.** *Suppose that $\mathcal{X}$ is a compact set in $\mathbb{R}^d$ and that $k \in C^\theta(\mathcal{X} \times \mathcal{X})$ for $\theta \in (0,1]$ (Definition 11). Assume that $1 \geq \lambda \geq n^{-\frac{1}{2+p}}$. With probability at least $1 - 2e^{-\tau}$, it holds for all $x \in \mathcal{X}$ simultaneously that*

$$\|C_{X,\lambda}^{-1}k(x,\cdot)\|^2_{2,n} \geq \frac{1}{2}\|[C_{X,\lambda}^{-1}k(x,\cdot)]\|^2_{L_2(\pi)} - \tau o(1),$$

$$\|C_{X,\lambda}^{-1}k(x,\cdot)\|^2_{2,n} \leq \frac{3}{2}\|[C_{X,\lambda}^{-1}k(x,\cdot)]\|^2_{L_2(\pi)} + \tau o(1).$$

*Proof.* The proof follows [29, Lemma C.11]. As we use different notations and tracking of constants, we provide a similar proof in our setting for completeness. By Lemma 24, there exists an $\epsilon$-net $\mathcal{F} \subseteq \mathcal{K}_\lambda \subseteq \mathcal{H}$ with respect to $\|\cdot\|_\infty$ such that there exists a positive constant $c$ with

$$|\mathcal{F}| \leq c(\lambda\epsilon)^{-\frac{2d}{\theta}},$$

for $\epsilon$ to be determined later. Using Lemma 3 and a union bound over the finite set $\mathcal{F}$, with probability at least $1 - 2e^{-\tau}$, it holds simultaneously for all $f \in \mathcal{F}$ that

$$\left| \|f\|^2_{2,n} - \|[f]\|^2_{L_2(\pi)} \right| \leq \frac{1}{2}\|[f]\|^2_{L_2(\pi)} + \frac{5(\tau + \log(|\mathcal{F}|))\kappa^2}{3\lambda^2 n}. \qquad (19)$$

We work in the event where Equation (19) holds for all $f \in \mathcal{F}$. By definition of an $\epsilon$-net and $\mathcal{K}_\lambda$, for any $x \in \mathcal{X}$, there exists some $f \in \mathcal{F}$ such that

$$\|C_{X,\lambda}^{-1}k(x,\cdot) - f\|_\infty \le \epsilon,$$

which in particular implies that

$$\left|\|[C_{X,\lambda}^{-1}k(x,\cdot)]\|_{L_2(\pi)} - \|[f]\|_{L_2(\pi)}\right| \le \epsilon$$
$$\left|\|C_{X,\lambda}^{-1}k(x,\cdot)\|_{2,n} - \|f\|_{2,n}\right| \le \epsilon.$$

Since $\|C_{X,\lambda}^{-1}k(x,\cdot)\|_\infty \le \kappa^2\lambda^{-1}$, using the algebraic identity $a^2 - b^2 = (a-b)(2b+(a-b))$, we obtain

$$\left|\|[C_{X,\lambda}^{-1}k(x,\cdot)]\|_{L_2(\pi)}^2 - \|[f]\|_{L_2(\pi)}^2\right| \le \epsilon(2\kappa^2\lambda^{-1} + \epsilon)$$
$$\left|\|C_{X,\lambda}^{-1}k(x,\cdot)\|_{2,n}^2 - \|f\|_{2,n}^2\right| \le \epsilon(2\kappa^2\lambda^{-1} + \epsilon).$$

We therefore have,

$$\|C_{X,\lambda}^{-1}k(x,\cdot)\|_{2,n}^2 \le \|f\|_{2,n}^2 + \epsilon(2\kappa^2\lambda^{-1} + \epsilon)$$
$$\le \frac{3}{2}\|[f]\|_{L_2(\pi)}^2 + \frac{5(\tau + \log(|\mathcal{F}|)\kappa^2}{3\lambda^2 n} + \epsilon(2\kappa^2\lambda^{-1} + \epsilon)$$
$$\le \frac{3}{2}\|[C_{X,\lambda}^{-1}k(x,\cdot)]\|_{L_2(\pi)}^2 + \frac{5(\tau + \log(|\mathcal{F}|)\kappa^2}{3\lambda^2 n} + 2\epsilon(2\kappa^2\lambda^{-1} + \epsilon).$$

We now choose $\epsilon = \frac{1}{n}$ and bound the error term. Recall that $1 \ge \lambda \ge n^{-\frac{1}{2+p}}$, therefore,

$$\frac{5(\tau + \log(|\mathcal{F}|)\kappa^2}{3\lambda^2 n} + 2\epsilon(2\kappa^2\lambda^{-1} + \epsilon) \le \frac{5(\tau + \log(|\mathcal{F}|)\kappa^2}{3}n^{-\frac{p}{2+p}} + 2\left(2\kappa^2 n^{\frac{-1-p}{2+p}} + \frac{1}{n^2}\right)$$
$$\le \frac{5\kappa^2}{3}\left(\tau + \log(c\lambda^{-\frac{2d}{\theta}}n^{\frac{2d}{\theta}})\right)n^{-\frac{p}{2+p}} + 2\left(2\kappa^2 n^{\frac{-1-p}{2+p}} + \frac{1}{n^2}\right)$$
$$= \tau o(1).$$

$\square$

**Lemma 5.** *For $1 \ge \lambda \ge n^{-\frac{1}{2+p}}$, with probability at least $1 - 4e^{-\tau}$, for $n \ge (c_0\tau)^{4+2p}$, we have for all $x \in \mathcal{X}$ simultaneously*

$$\left\|\hat{C}_X^{\frac{1}{2}}\hat{C}_{X,\lambda}^{-1}(C_X - \hat{C}_X)C_{X,\lambda}^{-1}k(x,\cdot)\right\|_{\mathcal{H}} = \tau o(1), \tag{20}$$

*where $c_0$ is the same constant as in Lemma 2.*

*Proof.*

$$\left\|\hat{C}_X^{\frac{1}{2}}\hat{C}_{X,\lambda}^{-1}(C_X - \hat{C}_X)C_{X,\lambda}^{-1}k(x,\cdot)\right\|_{\mathcal{H}}$$
$$= \left\|\hat{C}_X^{\frac{1}{2}}\hat{C}_{X,\lambda}^{-\frac{1}{2}}\hat{C}_{X,\lambda}^{-\frac{1}{2}}C_{X,\lambda}^{\frac{1}{2}}C_{X,\lambda}^{-\frac{1}{2}}(C_X - \hat{C}_X)C_{X,\lambda}^{-1}k(x,\cdot)\right\|_{\mathcal{H}}$$
$$\le \left\|\hat{C}_X^{\frac{1}{2}}\hat{C}_{X,\lambda}^{-\frac{1}{2}}\right\|_{\mathcal{H}\to\mathcal{H}}\left\|\hat{C}_{X,\lambda}^{-\frac{1}{2}}C_{X,\lambda}^{\frac{1}{2}}\right\|_{\mathcal{H}\to\mathcal{H}}$$
$$\cdot \left\|C_{X,\lambda}^{-\frac{1}{2}}(C_X - \hat{C}_X)C_{X,\lambda}^{-\frac{1}{2}}\right\|_{\mathcal{H}\to\mathcal{H}}\left\|C_{X,\lambda}^{-\frac{1}{2}}k(x,\cdot)\right\|_{\mathcal{H}}$$

We already saw in the proof of Lemma 2 that the first term is bounded by 1 and there is a constant $c_0 > 0$ such that for $\tau \ge \log(2)$, with probability at least $1 - 2e^{-\tau}$, for $n \ge (c_0\tau)^{4+2p}$, the second term is bounded by $\sqrt{2}$. For the third term we also saw in the proof of Lemma 4 that for $\tau \ge \log(2)$, with probability at least $1 - 2e^{-\tau}$, we have

$$\lambda^{-\frac{1}{2}}\left\|C_{X,\lambda}^{-\frac{1}{2}}(C_X - \hat{C}_X)C_{X,\lambda}^{-\frac{1}{2}}\right\|_{\mathcal{H}\to\mathcal{H}} \le \frac{4\kappa^2\xi_\delta}{3n\lambda^{\frac{3}{2}}} + \sqrt{\frac{2\kappa^2\xi_\delta}{n\lambda^2}},$$

where we defined

$$\xi_\delta = \log \frac{2\kappa^2(\mathcal{N}_1(\lambda)+1)}{e^{-\tau}\|C_X\|_{\mathcal{H}\to\mathcal{H}}}.$$

Finally, the fourth term is bounded above by $\lambda^{-\frac{1}{2}}\kappa$. Note that the bound on the fourth term is independent of $x$, so it holds simultaneously for all $x \in \mathcal{X}$. This is in contrast with the setting of Lemma 4 where for each fixed $x \in \mathcal{X}$ corresponds an element in the $\epsilon$-net of $\mathcal{F}$ for which we have a high probability bound, and therefore we must use a union bound in order for the bound to hold simultaneously for all $x \in \mathcal{X}$ in the proof of Lemma 4. As in the proof of Lemma 4 since $1 \geq \lambda \geq n^{-\frac{1}{2+p}}$, we have

$$\xi_\delta \leq \log \frac{2(c_{2,l}+1)}{e^{-\tau}\|C_X\|_{\mathcal{H}\to\mathcal{H}}} + \frac{p}{2+p}\log n.$$

In the bound on $\xi_\delta$ above, the first term does not depend on $n$, and the second term is logarithmic in $n$. Putting everything together by union bound, with probability at least $1 - 4e^{-\tau}$, for $n \geq (c_0\tau)^{4+2p}$, we have

$$\left\|\hat{C}_X^{\frac{1}{2}}\hat{C}_{X,\lambda}^{-1}(C_X - \hat{C}_X)C_{X,\lambda}^{-1}k(x,\cdot)\right\|_{\mathcal{H}} = \tau o(1).$$

$\square$

**Remark 6** (Comparison to [29])**.** *We explicit the differences between our proof strategy and the proof strategy of [29].*

- *Scalar versus vector-valued: lower bounding the bias in our case require us to accommodate for the vector-valued setting (see Lemma 1).*

- *New proof of the bias: we lower bound the bias through Lemma 2, while [29] obtain the lower bound in Lemma C.7; however the proof of Lemma C.7 implicitly uses the equality $\|A^{-1}\| = \|A\|^{-1}$, with $\|\cdot\|$ the operator norm, see Eq. (69) [29] and the preceding equations. It holds that $\|A^{-1}\| \geq \|A\|^{-1}$, but $\|A^{-1}\| \leq \|A\|^{-1}$ may not hold in general. We therefore develop a new proof for this step, leading to Lemma 2.*

- *New proof of the variance: we lower bound the variance in Lemma 5, while [29] lower bound the variance in Lemma C.12; to show Eq. (20), [29] use a covering argument involving $\mathcal{N}(\mathcal{K}_\lambda, \|\cdot\|_{\mathcal{H}}, \epsilon)$ (Lemma C.10). However, a close look at the proof of Lemma C.10 (last inequality of the proof) reveals that $\frac{\lambda_i}{\lambda+\lambda_i}$ was mistaken for $\frac{\lambda}{\lambda+\lambda_i}$ and plugging the correct term in the proof would lead to a vacuous bound. As explained in the proof of Lemma 5, we therefore develop a proof that is free of a covering number argument for this step.*

## C  Learning rates for spectral algorithms

To upper bound the excess-risk, we use a decomposition involving the *approximation error* expressed as $F_\lambda - F_*$ and the *estimation error* expressed as $\hat{F}_\lambda - F_\lambda$.

$$\left\|[\hat{F}_\lambda] - F_*\right\|_\gamma \leq \left\|[\hat{F}_\lambda - F_\lambda]\right\|_\gamma + \left\|[F_\lambda] - F_*\right\|_\gamma,$$

where $\hat{F}_\lambda$ is the empirical estimator based on general spectral regularization (Eq. (11)) and $F_\lambda$ is its counterpart in population (Eq. (10)). Note that this is a different decomposition than the *bias-variance decomposition* used in the proof of Theorem 3.

The proof structure is as follows:

1. Fourier expansion C.1.

2. Approximation Error C.2.

3. Estimation error C.3

## C.1 Fourier expansion

Recall the notations defined in Appendix A.3. The family $\{d_j\}_{j\in J}$ is an ONB of $\mathcal{Y}$, the family $\{\mu_i^{1/2}e_i\}_{i\in I}$ is an ONB of $(\ker I_\pi)^\perp$ and the family $\{\tilde{e}_i\}_{i\in I'}$ is an ONB of $\ker I_\pi$ such that $\left\{\mu_i^{1/2}e_i\right\}_{i\in I} \cup \{\tilde{e}_i\}_{i\in I'}$ forms an ONB of $\mathcal{H}$. Furthermore, recall that $\{\mu_i^{\beta/2}[e_i]\}_{i\in I}$ is an ONB of $[\mathcal{H}]^\beta$, $\beta \geq 0$.

**Lemma 6** (Fourier expansion). *Suppose Assumption* (SRC) *holds with $\beta \geq 0$. By definition of the vector-valued interpolation space and by Theorem 5, we have*

$$F_* = \sum_{i\in I, j\in J} a_{ij}d_j[e_i], \qquad a_{ij} = \langle F_*, d_j[e_i]\rangle_{L_2(\pi;\mathcal{Y})}, \qquad \|F_*\|_\beta^2 = \sum_{i\in I, j\in J} \frac{a_{ij}^2}{\mu_i^\beta}. \tag{21}$$

*Then, we have the following equalities with respect to this Fourier decomposition.*

1. *The Hilbert-Schmidt operator $C_\lambda \in S_2(\mathcal{H}, \mathcal{Y})$, Eq.* (10), *can be written as*

$$C_\lambda = \sum_{i\in I, j\in J} a_{ij} g_\lambda(\mu_i)\sqrt{\mu_i}\, d_j \otimes \sqrt{\mu_i}e_i. \tag{22}$$

2. *The Hilbert-Schmidt operator $(C_{YX} - C_\lambda C_X) C_{X,\lambda}^{-\frac{1}{2}} \in S_2(\mathcal{H}, \mathcal{Y})$ can be written as*

$$(C_{YX} - C_\lambda C_X) C_{X,\lambda}^{-\frac{1}{2}} = \sum_{i\in I, j\in J} a_{ij} r_\lambda(\mu_i)(\mu_i + \lambda)^{-\frac{1}{2}}\sqrt{\mu_i}\,(d_j \otimes \sqrt{\mu_i}e_i) \tag{23}$$

3. *The Hilbert-Schmidt operator $C_{YX} \in S_2(\mathcal{H}, \mathcal{Y})$ can be written as*

$$C_{YX} = \left(\sum_{i\in I, j\in J} a_{ij} \mu_i^{-\frac{\beta}{2}} d_j \otimes \sqrt{\mu_i}e_i\right) C_X^{\frac{\beta+1}{2}} \tag{24}$$

*Proof.* We first derive the Fourier expansion of $C_{YX}$,

$$\begin{aligned}
C_{YX} &= \mathbb{E}_{X,Y}\left[Y \otimes \phi(X)\right] \\
&= \mathbb{E}_X\left[F_*(X) \otimes \phi(X)\right] \tag{25} \\
&= \mathbb{E}_X\left[\sum_{i\in I, j\in J} a_{ij} e_i(X) d_j \otimes \phi(X)\right] \\
&= \mathbb{E}_X\left[\sum_{i\in I, j\in J} a_{ij} d_j \otimes \left(\sum_{k\in I}\sqrt{\mu_k}e_k(X)\sqrt{\mu_k}e_k\right)e_i(X)\right] \\
&= \sum_{ijk} a_{ij}\sqrt{\mu_k}\cdot \mathbb{E}_X[e_k(X)e_i(X)]\cdot d_j \otimes (\sqrt{\mu_k}e_k) \\
&= \sum_{i\in I, j\in J} a_{ij}\sqrt{\mu_i}\, d_j \otimes (\sqrt{\mu_i}e_i), \tag{26}
\end{aligned}$$

where in Eq. (25) we used the tower property of conditional expectation and in Eq. (26) we used the fact that $\{[e_i]\}_{i\in I}$ forms an orthonormal system in $L_2(\pi)$. We can manipulate Eq. (26) to derive Eq. (24),

$$\begin{aligned}
C_{YX} &= \sum_{i\in I, j\in J} a_{ij}\mu_i^{\frac{1}{2}-\frac{\beta+1}{2}} d_j \otimes \left(C_X^{\frac{\beta+1}{2}}(\sqrt{\mu_i}e_i)\right) \\
&= \left(\sum_{i\in I, j\in J} a_{ij}\mu_i^{-\frac{\beta}{2}} d_j \otimes \sqrt{\mu_i}e_i\right) C_X^{\frac{\beta+1}{2}}.
\end{aligned}$$

By the spectral decomposition of $C_X$ Eq. (6) and spectral calculus (Definition 9), we have that

$$g_\lambda(C_X) = \sum_{i\in I} g_\lambda(\mu_i)\sqrt{\mu_i}e_i \otimes \sqrt{\mu_i}e_i + g_\lambda(0)\sum_{i\in I'}\tilde{e}_i \otimes \tilde{e}_i, \tag{27}$$

$$r_\lambda(C_X) = \sum_{i\in I} r_\lambda(\mu_i)\sqrt{\mu_i}e_i \otimes \sqrt{\mu_i}e_i + \sum_{i\in I'}\tilde{e}_i \otimes \tilde{e}_i. \tag{28}$$

where we used $r_\lambda(0) = 1$.

**Proof of Eq.** (22). Using Eq. (26) and (27), we have

$$
\begin{aligned}
C_\lambda &= \left( \sum_{i \in I, j \in J} a_{ij} \sqrt{\mu_i} d_j \otimes (\sqrt{\mu_i} e_i) \right) \left( \sum_{k \in I} g_\lambda(\mu_k)(\sqrt{\mu_k} e_k) \otimes (\sqrt{\mu_k} e_k) + g_\lambda(0) \sum_{l \in I'} \tilde{e}_l \otimes \tilde{e}_l \right) \\
&= \sum_{ijk} a_{ij} \sqrt{\mu_i} g_\lambda(\mu_k) \delta_{ik} d_j \otimes (\sqrt{\mu_k} e_k) \qquad\qquad (29) \\
&= \sum_{i \in I, j \in J} a_{ij} \sqrt{\mu_i} g_\lambda(\mu_i) d_j \otimes (\sqrt{\mu_i} e_i),
\end{aligned}
$$

where in Eq. (29), we recall the fact that $\{\sqrt{\mu_i} e_i\}_{i \in I}$ forms an ONB of $(\ker I_\pi)^\perp$ and $\{\tilde{e}_i\}_{i \in I'}$ forms an ONB of $\ker I_\pi$.

**Proof of Eq.** (23). Using Eq. (26) and (28), we have

$$
\begin{aligned}
\left( C_{YX} - C_\lambda C_X \right) C_{X,\lambda}^{-\frac{1}{2}} &= C_{YX} r_\lambda(C_X) C_{X,\lambda}^{-\frac{1}{2}} \\
&= \left( \sum_{i \in I, j \in J} a_{ij} \sqrt{\mu_i} d_j \otimes (\sqrt{\mu_i} e_i) \right) \left( \sum_{k \in I} r_\lambda(\mu_k) \sqrt{\mu_k} e_k \otimes \sqrt{\mu_k} e_k + \sum_{l \in I'} \tilde{e}_l \otimes \tilde{e}_l \right) C_{X,\lambda}^{-\frac{1}{2}} \\
&= \left( \sum_{ijk} a_{ij} \sqrt{\mu_i} r_\lambda(\mu_k) d_j \otimes (\sqrt{\mu_k} e_k) \delta_{ik} \right) C_{X,\lambda}^{-\frac{1}{2}} \\
&= \sum_{i \in I, j \in J} a_{ij} \sqrt{\mu_i} r_\lambda(\mu_i) d_j \otimes (C_{X,\lambda}^{-\frac{1}{2}}(\sqrt{\mu_i} e_i)) \\
&= \sum_{i \in I, j \in J} a_{ij} \sqrt{\mu_i} (\mu_i + \lambda)^{-\frac{1}{2}} r_\lambda(\mu_i) d_j \otimes (\sqrt{\mu_i} e_i),
\end{aligned}
$$

$\square$

**Lemma 7.** *Suppose Assumption* (SRC) *holds with $\beta \geq 0$, then the following bound is satisfied, for all $\lambda > 0$ and $0 \leq \gamma \leq 1$, we have*

$$
\| [F_\lambda] \|_\gamma^2 \leq E^2 \| F_* \|_{\min\{\gamma, \beta\}}^2 \lambda^{-(\gamma - \beta)_+}.
$$

*For the definition of E, see Eq.* (8).

*Proof.* We adopt the notations of Lemma 6. By Parseval's identity and Eq. (22), we have

$$
\begin{aligned}
\| [F_\lambda] \|_\gamma^2 &= \| C_\lambda \|_{S_2([\mathcal{H}]^\gamma, \mathcal{Y})}^2 \\
&= \sum_{i \in I, j \in J} a_{ij}^2 g_\lambda(\mu_i)^2 \mu_i^{2-\gamma}.
\end{aligned}
$$

In the case of $\gamma \leq \beta$, we bound $g_\lambda(\mu_i) \mu_i \leq E$ using Eq. (8). Then, by Eq. (21),

$$
\| [F_\lambda] \|_\gamma^2 \leq E^2 \sum_{i \in I, j \in J} \frac{a_{ij}^2}{\mu_i^\gamma} = E^2 \| F_* \|_\gamma^2.
$$

In the case of $\gamma > \beta$, we apply Eq. (8) to $g_\lambda(\mu_i) \mu_i^{1 - \frac{\gamma - \beta}{2}} \leq E \lambda^{-\frac{\gamma - \beta}{2}}$ to obtain, using Eq. (21) again,

$$
\begin{aligned}
\| [F_\lambda] \|_\gamma^2 &= \sum_{i \in I, j \in J} g_\lambda(\mu_i)^2 \mu_i^{2-(\gamma - \beta)} \mu_i^{-\beta} a_{ij}^2 \\
&\leq E^2 \lambda^{-(\gamma - \beta)} \sum_{i \in I, j \in J} \mu_i^{-\beta} a_{ij}^2 \\
&= E^2 \lambda^{-(\gamma - \beta)} \| F_* \|_\beta^2.
\end{aligned}
$$

$\square$

**Lemma 8.** *Suppose Assumption* (SRC) *holds for* $0 \le \beta \le 2\rho$*, with* $\rho$ *the qualification. Then, the following bound is satisfied, for all* $\lambda > 0$*, we have*

$$\left\| (C_{YX} - C_\lambda C_X) C_{X,\lambda}^{-\frac{1}{2}} \right\|_{S_2(\mathcal{H},\mathcal{Y})} \le \omega_\rho \|F_*\|_\beta \lambda^{\frac{\beta}{2}}.$$

*For the definition of* $\omega_\rho$*, see Eq.* (9)*.*

*Proof.* Recall that in Lemma 6 we used the decomposition

$$F_* = \sum_{i \in I, j \in J} a_{ij} d_j [e_i],$$

where Assumption (SRC) implies that $\|F_*\|_\beta^2 = \sum_{ij} \frac{a_{ij}^2}{\mu_i^\beta} < \infty$. Using Eq. (23) in Lemma 6 and Parseval's identity w.r.t. the ONS $\{d_j \otimes \mu_i^{1/2} e_i\}_{i \in I, j \in J}$ in $S_2(\mathcal{H}, \mathcal{Y})$, we have

$$\begin{aligned}
\left\| (C_{YX} - C_\lambda C_X) C_{X,\lambda}^{-\frac{1}{2}} \right\|_{S_2(\mathcal{H},\mathcal{Y})} &= \left( \sum_{i \in I, j \in J} a_{ij}^2 r_\lambda^2(\mu_i)(\mu_i + \lambda)^{-1}\mu_i \right)^{\frac{1}{2}} \\
&\le \left( \sum_{i \in I, j \in J} \frac{a_{ij}^2}{\mu_i^\beta} r_\lambda^2(\mu_i)\mu_i^\beta \right)^{\frac{1}{2}} \\
&\le \|F_*\|_\beta \sup_{i \in I} r_\lambda(\mu_i)\mu_i^{\frac{\beta}{2}} \\
&\le \|F_*\|_\beta \omega_\rho \lambda^{\frac{\beta}{2}}.
\end{aligned}$$

$\square$

**Lemma 9.** *Suppose Assumption* (SRC) *holds with* $\beta \ge 0$*, then for all* $\lambda > 0$*, we have*

$$\left\| C_\lambda r_\lambda(\hat{C}_X) \hat{C}_{X,\lambda}^{\frac{1}{2}} \right\|_{S_2(\mathcal{H},\mathcal{Y})} \le B \left\| \hat{C}_{X,\lambda}^{\frac{1}{2}} r_\lambda(\hat{C}_X) g_\lambda(C_X) C_X^{\frac{\beta+1}{2}} \right\|_{\mathcal{H} \to \mathcal{H}},$$

*where* $\|F_*\|_\beta = B < \infty$*.*

*Proof.* Recall that Lemma 6 we used the decomposition

$$F_* = \sum_{i \in I, j \in j} a_{ij} d_j [e_i],$$

where $\|F_*\|_\beta^2 = \sum_{ij} \frac{a_{ij}^2}{\mu_i^\beta} = B^2 < \infty$. Using Eq. (24) in Lemma 6 and $C_\lambda = C_{YX} g_\lambda(C_X)$, we have

$$\begin{aligned}
\left\| C_\lambda r_\lambda(\hat{C}_X) \hat{C}_{X,\lambda}^{\frac{1}{2}} \right\|_{S_2(\mathcal{H},\mathcal{Y})} &= \left\| \left( \sum_{ij} a_{ij} \mu_i^{-\frac{\beta}{2}} d_j \otimes \sqrt{\mu_i} e_i \right) C_X^{\frac{\beta+1}{2}} g_\lambda(C_X) r_\lambda(\hat{C}_X) \hat{C}_{X,\lambda}^{\frac{1}{2}} \right\|_{S_2(\mathcal{H},\mathcal{Y})} \\
&\le B \left\| C_X^{\frac{\beta+1}{2}} g_\lambda(C_X) r_\lambda(\hat{C}_X) \hat{C}_{X,\lambda}^{\frac{1}{2}} \right\|_{\mathcal{H} \to \mathcal{H}},
\end{aligned}$$

where we notice that the $S_2(\mathcal{H}, \mathcal{Y})$ norm of the first term is exactly the $\beta$ norm of $F_*$, which is given by $B$. Recalling that $C_X, \hat{C}_X$ are self adjoint, we prove the final result by taking the adjoint and using that an operator has the same operator norm as its adjoint. $\square$

## C.2 Approximation Error

**Lemma 10.** *Let* $F_\lambda$ *be given by Eq.* (10) *based on a general spectral filter satisfying Definition 2 with qualification* $\rho \ge 0$*. Suppose Assumption* (SRC) *holds with parameter* $\beta \ge 0$ *and define* $\beta_\rho = \min\{\beta, 2\rho\}$*, then the following bound is satisfied, for all* $\lambda > 0$ *and* $0 \le \gamma \le \beta_\rho$*,*

$$\|[F_\lambda] - F_*\|_\gamma^2 \le \omega_\rho^2 \|F_*\|_{\beta_\rho}^2 \lambda^{\beta_\rho - \gamma}.$$

*Proof.* In Eq. (10), we defined $F_\lambda(\cdot) = C_\lambda \phi(\cdot)$. On the other hand, in Lemma 6 we obtained the Fourier expansion of $C_\lambda$ leading to Eq. (22). Thus we have for $\pi$−almost all $x \in \mathcal{X}$,

$$F_\lambda(x) = \sum_{i \in I, j \in J} a_{ij} \mu_i g_\lambda(\mu_i) d_j e_i(x).$$

Therefore,

$$[F_\lambda] - F_* = \sum_{i \in I, j \in J} a_{ij}(1 - \mu_i g_\lambda(\mu_i)) d_j[e_i] = \sum_{i \in I, j \in J} a_{ij} r_\lambda(\mu_i) d_j[e_i].$$

Suppose $\beta \le 2\rho$, using Parseval's identity w.r.t. the ONB $\{d_j \mu_i^{\gamma/2}[e_i]\}_{i \in I, j \in J}$ of $[\mathcal{G}]^\gamma$, we have

$$
\begin{aligned}
\|[F_\lambda] - F_*\|_\gamma^2 &= \left\| \sum_{i \in I, j \in J} \frac{a_{ij}}{\mu_i^{\gamma/2}} r_\lambda(\mu_i) d_j \mu_i^{\gamma/2}[e_i] \right\|_\gamma^2 \\
&= \sum_{i \in I, j \in J} \frac{a_{ij}^2}{\mu_i^\gamma} r_\lambda^2(\mu_i) \\
&= \sum_{i \in I, j \in J} \frac{a_{ij}^2}{\mu_i^\beta} r_\lambda^2(\mu_i) \mu_i^{\beta - \gamma} \\
&\le \omega_\rho^2 \lambda^{\beta - \gamma} \sum_{i \in I, j \in J} \frac{a_{ij}^2}{\mu_i^\beta} \\
&= \|F_*\|_\beta^2 \omega_\rho^2 \lambda^{\beta - \gamma}
\end{aligned}
$$

where we used Eq. (9) in the definition of a filter function, together with $0 \le \beta \le 2\rho$ and $0 \le \gamma \le \beta$, which taken together implies that $0 \le \frac{\beta - \gamma}{2} \le \rho$. Finally, if $\beta \ge 2\rho$, then since $[\mathcal{G}]^\beta \subseteq [\mathcal{G}]^{2\rho}$, we can perform the last derivations again with $\tilde{\beta} = 2\rho$ to obtain the final result. $\qquad\square$

## C.3 Estimation error

Before proving the main results we recall two *embedding properties* for the vector-valued interpolation space $[\mathcal{G}]^\beta$ (Definition 1). The first embedding property lifts the property (EMB) defined for the scalar-valued RKHS $[\mathcal{H}]^\alpha$ to the vector-valued RKHS $[\mathcal{G}]^\alpha$.

**Lemma 11** ($L_\infty$-embedding property - Lemma 4 [31]). *Under* (EMB) *the inclusion operator* $\mathcal{I}_\pi^{\alpha, \beta} : [\mathcal{G}]^\alpha \hookrightarrow L_\infty(\pi; \mathcal{Y})$ *is bounded with operator norm* $A$,

**Theorem 6** ($L_q$-embedding property - Theorem 3 [31]). *Let Assumption* (EMB) *be satisfied with parameter* $\alpha \in (0, 1]$. *For any* $\beta \in [0, \alpha)$, *the inclusion map*

$$\mathcal{I}_\pi^{q_{\alpha, \beta}} : [\mathcal{G}]^\beta \hookrightarrow L_{q_{\alpha, \beta}}(\pi; \mathcal{Y})$$

*is bounded, where* $q_{\alpha, \beta} := \frac{2\alpha}{\alpha - \beta}$.

The $L_q$-embedding property was first introduced in the scalar-valued setting in [55] and then lifted to the vector-valued setting by [31]. Its role is to replace a boundedness condition on the ground truth function $F_*$. We now explain how the $L_q$-embedding property can be combined with Assumption (EMB) and a truncation technique.

**Lemma 12.** *Recall that* $\pi$ *is the marginal measure of* $X$ *on* $\mathcal{X}$. *For* $t \ge 0$, *define the measurable set* $\Omega_t$ *as follows*

$$\Omega_t := \{x \in \mathcal{X} : \|F_*(x)\|_\mathcal{Y} \le t\}$$

*Let* $q > 0$. *Assume that* $F_* \in L_q(\pi; \mathcal{Y})$. *In other words, there exists some constant* $c_q > 0$ *such that*

$$\|F_*\|_{L_q(\pi; \mathcal{Y})} = \left( \int_\mathcal{X} \|F_*(x)\|_\mathcal{Y}^q \mathrm{d}\pi(x) \right)^{\frac{1}{q}} = c_q < +\infty,$$

*Then we have the following conclusions*

1. *The* $\pi$-*measure of the complement of* $\Omega$ *can be bounded by*

$$\pi(\{x \notin \Omega_t\}) \le \frac{c_q^q}{t^q}.$$

2. *Recall that $\{x_i\}_{i=1}^n$ are i.i.d. samples distributed according to $\pi$. If $t = n^{\frac{1}{\tilde{q}}}$ for $\tilde{q} < q$, then we can conclude as follows. For a fixed parameter $\tau > 0$, for all sufficiently large $n$, where the hidden index bound depends on $q\tilde{q}^{-1}$ and $\tau$, we have*

$$\pi^{\otimes n} \left( \cap_{i=1}^n \{x_i \in \Omega_t\} \right) \geq 1 - e^{-\tau}.$$

*Proof.* The first claim is a straightforward application of Markov's inequality, as follows

$$\pi(\{x \notin \Omega_t\}) = \pi \left( \|F_*(x)\|_{\mathcal{Y}} > t \right) \leq \frac{\mathbb{E}_\pi \left[ \|F_*(X)\|_{\mathcal{Y}}^q \right]}{t^q} = \frac{c_q^q}{t^q}.$$

To show the second claim, we first evaluate the probability that there exists some $x_i$'s that lies outside $\Omega_t$,

$$\begin{aligned}
\pi^{\otimes n} \left( \cup_{i=1}^n \{x_i \notin \Omega_t\} \right) &= 1 - \pi^{\otimes n} \left( \cap_{i=1}^n \{x_i \in \Omega_t\} \right) \\
&= 1 - \pi(\{x_i \in \Omega_t\})^n \\
&\leq 1 - \left( 1 - \frac{c_q^q}{t^q} \right)^n \\
&\leq \frac{c_q^q n}{t^q},
\end{aligned}$$

where in the last inequality we used Bernoulli's inequality, which states that for $r \geq 1$ and $0 \leq x \leq 1$,

$$(1 - x)^r \geq 1 - rx.$$

By assumption $t = n^{\frac{1}{q_t}}$ for some fixed $q > q_t > 0$. We thus have

$$\pi^{\otimes n} \left( \cup_{i=1}^n \{x_i \notin \Omega_t\} \right) \leq c_q^q n^{1 - \frac{q}{q_t}} \leq e^{-\tau},$$

for sufficiently large $n$, where the hidden index bound depends on $\frac{q}{q_t}$ and $\tau$. $\qquad\square$

We adapt [31, Lemma 5] to the spectral algorithms setting.

**Lemma 13.** *Suppose Assumptions* (SRC) *and* (EMB) *hold for some $0 \leq \beta \leq 2\rho$, with $\rho$ the qualification, then the following bounds are satisfied, for all $0 < \lambda \leq 1$,*

$$\|[F_\lambda] - F_*\|_{L_\infty}^2 \leq \left( \|F_*\|_{L_\infty} + A \max\{E, \omega_\rho\} \|F_*\|_\beta \right)^2 \lambda^{\beta - \alpha}, \tag{30}$$

$$\|[F_\lambda]\|_{L_\infty}^2 \leq A^2 E^2 \|F_*\|_{\min\{\alpha,\beta\}}^2 \lambda^{-(\alpha - \beta)_+}, \tag{31}$$

*Proof.* We use Lemma 11 and Lemma 7 to write:

$$\|[F_\lambda]\|_\infty^2 \leq A^2 \|[F_\lambda]\|_\alpha^2 \leq A^2 E^2 \|F_*\|_{\min\{\alpha,\beta\}}^2 \lambda^{-(\alpha - \beta)_+}.$$

This proves Eq. (31). To show Eq. (30), in the case $\beta \leq \alpha$ we use the triangle inequality, Eq. (31) and $\lambda \leq 1$ to obtain

$$\begin{aligned}
\|[F_\lambda] - F_*\|_\infty &\leq \|F_*\|_\infty + \|[F_\lambda]\|_\infty \\
&\leq \left( \|F_*\|_\infty + AE\|F_*\|_\beta \right) \lambda^{-\frac{\alpha - \beta}{2}}.
\end{aligned}$$

In the case $\beta > \alpha$, Eq. (30) is a consequence of Lemma 11 and Lemma 10 with $\gamma = \alpha$ (here we use the assumption $0 \leq \beta \leq 2\rho$),

$$\|[F_\lambda] - F_*\|_\infty^2 \leq A^2 \|[F_\lambda] - F_*\|_\alpha^2 \leq A^2 \omega_\rho^2 \|F_*\|_\beta^2 \lambda^{\beta - \alpha} \leq \left( \|F_*\|_\infty + A\omega_\rho \|F_*\|_\beta \right)^2 \lambda^{\beta - \alpha}.$$

$\qquad\square$

We adapt [54, Theorem 13] to the vector-valued setting.

**Theorem 7.** *Suppose that Assumptions* (EMB), (EVD), (MOM) *and* (SRC) *hold for* $0 \leq \beta \leq 2\rho$, *where $\rho$ is the qualification, and $p \leq \alpha \leq 1$. Denote, for $i = 1, \ldots, n$,*

$$\xi_i = \xi(x_i, y_i) = \left((y_i - C_\lambda \phi(x_i)) \otimes \phi(x_i)\right) C_{X,\lambda}^{-\frac{1}{2}},$$

*and for $t \geq 0$,*

$$\Omega_t = \{x \in \mathcal{X} : \|F_*(x)\|_{\mathcal{Y}} \leq t\}$$

*Then for all $\tau \geq 1$, with probability at least $1 - 2e^{-\tau}$, we have*

$$\left\| \frac{1}{n} \sum_{i=1}^{n} \xi_i \mathbb{1}\{x_i \in \Omega_t\} - \mathbb{E}[\xi(X, Y)\mathbb{1}\{X \in \Omega_t\}] \right\|_{S_2(\mathcal{H}, \mathcal{Y})}$$

$$\leq \tau \left( c_1 \lambda^{\frac{\beta}{2} - \alpha} n^{-1} + c_2 \lambda^{-\frac{\alpha}{2}} n^{-1}(t + R + A) + \frac{c_3 \sqrt{\mathcal{N}_1(\lambda)}}{\sqrt{n}} + \frac{c_4}{\sqrt{n}\lambda^{\frac{\alpha - \beta}{2}}} \right)$$

*where $R$ is the constant from Assumption* (MOM)*, and*

$$c_1 = 8\sqrt{2}A^2 \max\{E, \omega_\rho\} \|F_*\|_\beta$$
$$c_2 = 8\sqrt{2}A$$
$$c_3 = 8\sqrt{2}\sigma$$
$$c_4 = 8\sqrt{2}A\|F_*\|_\beta \omega_\rho$$

*where $A$ is the constant from Assumption* (EMB)*, and $E, \omega_\rho$ are defined in Eq.* (8) *and* (9) *respectively.*

*Proof.* We wish to apply vector-valued Bernstein's inequality, namely Theorem 10. We thus compute,

$$\mathbb{E}\left[\|\xi(X, Y)\mathbb{1}\{X \in \Omega_t\}\|_{S_2(\mathcal{H}, \mathcal{Y})}^m\right] = \mathbb{E}\left[\mathbb{1}\{X \in \Omega_t\} \left\|(Y - C_\lambda\phi(X)) \otimes \left(C_{X,\lambda}^{-\frac{1}{2}}\phi(X)\right)\right\|_{S_2(\mathcal{H}, \mathcal{Y})}^m\right]$$

$$= \mathbb{E}\left[\mathbb{1}\{X \in \Omega_t\} \|(Y - C_\lambda\phi(X))\|_{\mathcal{Y}}^m \left\|C_{X,\lambda}^{-\frac{1}{2}}\phi(X)\right\|_{\mathcal{H}}^m\right]$$

$$= \int_{\Omega_t} \left\|C_{X,\lambda}^{-\frac{1}{2}}\phi(x)\right\|_{\mathcal{H}}^m \int_{\mathcal{Y}} \|y - C_\lambda\phi(x)\|_{\mathcal{Y}}^m \, \mathrm{d}p(x, \mathrm{d}y)\mathrm{d}\pi(x). \tag{32}$$

First we consider the inner integral, by Assumption (MOM),

$$\int_{\mathcal{Y}} \|(y - C_\lambda\phi(x))\|_{\mathcal{Y}}^m \, \mathrm{d}p(x, \mathrm{d}y) \leq 2^{m-1}\left(\int_{\mathcal{Y}} \|y - F_*(x)\|_{\mathcal{Y}}^m + \|F_\lambda(x) - F_*(x)\|_{\mathcal{Y}}^m\right)\mathrm{d}p(x, \mathrm{d}y)$$

$$= m!\sigma^2(2R)^{m-2} + 2^{m-1}\|F_\lambda(x) - F_*(x)\|_{\mathcal{Y}}^m.$$

Plugging the above inequality into Eq. (32), as well as introducing the shorthand,

$$h_x := C_{X,\lambda}^{-\frac{1}{2}}\phi(x),$$

we have

$$\mathbb{E}\left[\|\xi(X, Y)\mathbb{1}\{X \in \Omega_t\}\|_{S_2(\mathcal{H}, \mathcal{Y})}^m\right] \leq m!\sigma^2(2R)^{m-2} \int_{\Omega_t} \|h_x\|_{\mathcal{H}}^m \mathrm{d}\pi(x) \tag{33}$$

$$+ 2^{m-1} \int_{\Omega_t} \|h_x\|_{\mathcal{H}}^m \|F_\lambda(x) - F_*(x)\|_{\mathcal{Y}}^m \, \mathrm{d}\pi(x).$$

We bound term (33) using Lemma 15 and Lemma 17 with $l = 1$. We have,

$$\int_{\Omega_t} \|h_x\|_{\mathcal{H}}^m \mathrm{d}\pi(x) \leq (A\lambda^{-\frac{\alpha}{2}})^{m-2}\mathcal{N}_1(\lambda).$$

Therefore we bound term (33) as follows,

$$m!\sigma^2(2R)^{m-2} \int_{\Omega_t} \|h_x\|_{\mathcal{H}}^m \mathrm{d}\pi(x) \leq m!\sigma^2 \left(\frac{2AR}{\lambda^{\frac{\alpha}{2}}}\right)^{m-2} \mathcal{N}_1(\lambda).$$

If $\beta \geq \alpha$, by Assumption (EMB),

$$\|F_*\|_\infty \leq A\|F_*\|_\alpha \leq A\|F_*\|_\beta.$$

Hence by Lemma 13,

$$\|[F_\lambda] - F_*\|_\infty \leq (\|F_*\|_\infty + A\max\{E, \omega_\rho\}\|F_*\|_\beta)\lambda^{\frac{\beta-\alpha}{2}} \leq A(1 + \max\{E, \omega_\rho\})\|F_*\|_\beta\lambda^{\frac{\beta-\alpha}{2}}.$$

If $\beta < \alpha$, by Lemma 13, we have for $\pi$-almost all $x \in \Omega_t$,

$$\|F_*(x) - F_\lambda(x)\|_{\mathcal{Y}} \leq t + \|[F_\lambda]\|_{L_\infty(\pi;\mathcal{Y})} \leq t + AE\|F_*\|_\beta\lambda^{\frac{\beta-\alpha}{2}}.$$

Therefore, for all $\beta \in [0, 2\rho]$,

$$\|(F_* - [F_\lambda])\mathbb{1}_{X \in \Omega_t}\|_{L_\infty(\pi;\mathcal{Y})} \leq t + A(1 + \max\{E, \omega_\rho\}\|F_*\|_\beta\lambda^{\frac{\beta-\alpha}{2}}) =: \chi(t, \lambda).$$

Using Lemma 17 with $l = 1$, we have,

$$2^{m-1} \int_{\Omega_t} \|h_x\|_{\mathcal{H}}^m \|F_*(x) - F_\lambda(x)\|_{\mathcal{Y}}^m \mathrm{d}\pi(x)$$

$$\leq 2^{m-1}\chi(t, \lambda)^{m-2}(A\lambda^{-\frac{\alpha}{2}})^m \|F_* - [F_\lambda]\|_{L_2(\pi;\mathcal{Y})}^2$$

$$= \left(\frac{2\chi(t, \lambda)A}{\lambda^{\frac{\alpha}{2}}}\right)^{m-2} \|F_* - [F_\lambda]\|_{L_2(\pi;\mathcal{Y})}^2 \frac{2A^2}{\lambda^\alpha}$$

$$\leq m! \left(\frac{2\chi(t, \lambda)A}{\lambda^{\frac{\alpha}{2}}}\right)^{m-2} \|F_* - [F_\lambda]\|_{L_2(\pi;\mathcal{Y})}^2 \frac{2A^2}{\lambda^\alpha}.$$

Putting everything together,

$$\mathbb{E}\left[\|\xi(X, Y)\mathbb{1}\{X \in \Omega_t\}\|_{S_2(\mathcal{H},\mathcal{Y})}^m\right] \leq m! \left(\frac{2(R + \chi(t, \lambda))A}{\lambda^{\frac{\alpha}{2}}}\right)^{m-2} \left(\sigma^2\mathcal{N}_1(\lambda) + \|F_* - [F_\lambda]\|_{L_2(\pi;\mathcal{Y})}^2 \frac{2A^2}{\lambda^\alpha}\right).$$

We now apply Theorem 10 with

$$L \leftarrow \frac{2(R + \chi(t, \lambda))A}{\lambda^{\frac{\alpha}{2}}}$$

$$\sigma \leftarrow 2\sigma\sqrt{\mathcal{N}_1(\lambda)} + \|F_* - [F_\lambda]\|_{L_2(\pi;\mathcal{Y})}\frac{2A}{\lambda^{\frac{\alpha}{2}}}$$

We bound $\|F_* - [F_\lambda]\|_{L_2(\pi;\mathcal{Y})}$ using Lemma 10 with $\gamma = 0$,

$$\|F_* - [F_\lambda]\|_{L_2(\pi;\mathcal{Y})} \leq \omega_\rho\|F_*\|_\beta\lambda^{\frac{\beta}{2}}.$$

The conclusion is, for all $\tau \geq 1$, with probability at least $1 - 2e^{-\tau}$, we have

$$\left\|\frac{1}{n}\sum_{i=1}^n \xi_i\mathbb{1}\{x_i \in \Omega_t\} - \mathbb{E}[\xi(X, Y)\mathbb{1}\{X \in \Omega_t\}]\right\|_{S_2(\mathcal{H},\mathcal{Y})}$$

$$\leq 4\sqrt{2}\tau\left(\frac{2\sigma\sqrt{\mathcal{N}_1(\lambda)} + \|F_* - [F_\lambda]\|_{L_2(\pi;\mathcal{Y})}\frac{2A}{\lambda^{\frac{\alpha}{2}}}}{\sqrt{n}} + \frac{2(R + \chi(t, \lambda))A}{n\lambda^{\frac{\alpha}{2}}}\right)$$

$$\leq 4\sqrt{2}\tau\left(\frac{2\sigma}{\sqrt{n}}\sqrt{\mathcal{N}_1(\lambda)} + \frac{2A\|F_*\|_\beta\omega_\rho}{\sqrt{n}\lambda^{\frac{\alpha-\beta}{2}}} + \frac{2(R + t + A)A}{n\lambda^{\frac{\alpha}{2}}} + \frac{2A^2\max\{E, \omega_\rho\}\|F_*\|_\beta}{n\lambda^{\alpha-\frac{\beta}{2}}}\right).$$

$$\square$$

**Lemma 14.** *Suppose that the same assumptions and notations listed in Theorem 7 hold.*

1. *Suppose $\beta + p > \alpha$, and $\lambda \asymp n^{-\frac{1}{\beta+p}}$. For any fixed $\tau \geq 1$, with probability at least $1 - 2e^{-\tau}$, suppose that the truncation level $t$ satisfies*

$$t \leq n^{\frac{1}{2}\left(1 + \frac{p-\alpha}{p+\beta}\right)},$$

   *then there exists a constant $c > 0$ such that*

$$\left\|\frac{1}{n}\sum_{i=1}^n \xi_i\mathbb{1}\{x_i \in \Omega_t\} - \mathbb{E}[\xi(X, Y)\mathbb{1}\{X \in \Omega_t\}]\right\|_{S_2(\mathcal{H},\mathcal{Y})} \leq c\tau n^{-\frac{1}{2}\frac{\beta}{\beta+p}}.$$

2. *Suppose $\beta + p \leq \alpha$, and $\lambda \asymp \left(\frac{n}{\log^\theta(n)}\right)^{\frac{1}{\alpha}}$ for some $\theta > 1$. For any fixed $\tau \geq 1$, with probability at least $1 - 2e^{-\tau}$, suppose that the truncation level $t$ satisfies*

$$t \leq n^{\frac{1}{2}\left(1 - \frac{\beta}{\alpha}\right)},$$

*then there exists a constant $c > 0$ such that*

$$\left\| \frac{1}{n} \sum_{i=1}^{n} \xi_i \mathbb{1}\{x_i \in \Omega\} - \mathbb{E}[\xi(X,Y)\mathbb{1}\{X \in \Omega\}] \right\|_{\mathcal{G}} \leq c\tau \left(\frac{n}{\log^\theta(n)}\right)^{-\frac{\beta}{2\alpha}}$$

*Proof.* Note that Theorem 7 yields the same conclusion as in the scalar-valued case proved in [54, Theorem 13]. The Lemma then follows from the analysis for the scalar-valued case in the proof of [54, Theorem 15]. $\qquad\square$

We adapt [55, Theorem 15] to the vector-valued setting.

**Theorem 8.** *Suppose that Assumptions* (EMB)*,* (EVD)*,* (MOM) *and* (SRC) *hold for $0 \leq \beta \leq 2\rho$, where $\rho$ is the qualification, and $p \leq \alpha \leq 1$.*

1. *In the case of $\beta + p > \alpha$, choosing $\lambda \asymp n^{-\frac{1}{\beta+p}}$, for any fixed $\tau \geq \log(4)$, when $n$ is sufficiently large, with probability at least $1 - 4e^{-\tau}$, we have*

$$\left\| \left(\left(\hat{C}_{YX} - C_\lambda \hat{C}_X\right) - (C_{YX} - C_\lambda C_X)\right) C_{X,\lambda}^{-\frac{1}{2}} \right\|_{S_2(\mathcal{H},\mathcal{Y})} \leq c\tau n^{-\frac{1}{2}\frac{\beta}{\beta+p}} \qquad (34)$$

*where $c$ is a constant independent of $n, \tau, \lambda$.*

2. *In the case of $\beta + p \leq \alpha$, choosing $\lambda \asymp \left(\frac{n}{\log^\theta(n)}\right)^{-\frac{1}{\alpha}}$ for some $\theta > 1$. We make the additional assumption that there exists some $\alpha' < \alpha$ such that Assumption* (MOM) *is satisfied for $\alpha' < \alpha$. Then, for any fixed $\tau \geq \log(4)$, when $n$ is sufficiently large, where the hidden index bound depends on $\alpha - \alpha'$, with probability at least $1 - 4e^{-\tau}$, we have*

$$\left\| \left(\left(\hat{C}_{YX} - C_\lambda \hat{C}_X\right) - (C_{YX} - C_\lambda C_X)\right) C_{X,\lambda}^{-\frac{1}{2}} \right\|_{S_2(\mathcal{H},\mathcal{Y})} \leq c\tau \left(\frac{n}{\log^\theta(n)}\right)^{-\frac{\beta}{2\alpha}} \qquad (35)$$

*where $c$ is a constant independent of $n, \tau, \lambda$.*

*Proof.* By assumption (EMB) and Theorem 6, if $\beta < \alpha$, then the inclusion map

$$\mathcal{I}_\pi^{q_{\alpha,\beta}} : [\mathcal{G}]^\beta \hookrightarrow L_{q_{\alpha,\beta}}(\pi; \mathcal{Y})$$

is bounded, where $q_{\alpha,\beta} := \frac{2\alpha}{\alpha-\beta}$. If $\beta \geq \alpha$, then by Lemma 11 the inclusion map

$$\mathcal{I}_\pi^\infty : [\mathcal{G}]^\beta \hookrightarrow L_\infty(\pi; \mathcal{Y})$$

is bounded and therefore $[\mathcal{G}]^\beta$ is continuously embedded into $L_q(\pi; \mathcal{Y})$ for any $q \geq 1$. In the rest of the proof, we will use $q$ to denote $q_{\alpha,\beta}$, unless otherwise specified. Furthermore, we will use $c_q = \|F_*\|_{L_q(\pi;\mathcal{Y})}$.

We first consider the case $\beta + p > \alpha$. We can easily verify using $\beta + p > \alpha$ that the following inequality holds

$$\frac{1}{2}\left(1 + \frac{p-\alpha}{p+\beta}\right) > \frac{1}{2}\left(\frac{p}{p+\beta}\right) > \frac{\alpha - \beta}{2\alpha} = \frac{1}{q_{\alpha,\beta}}.$$

Choose $t = n^{\tilde{q}^{-1}}$, where

$$\frac{1}{\tilde{q}} = \frac{1}{2}\left(\frac{1}{2}\left(1 + \frac{p-\alpha}{p+\beta}\right) + \frac{1}{q}\right).$$

We thus have

$$n^{\frac{1}{2}\left(1 + \frac{p-\alpha}{p+\beta}\right)} > t = n^{\tilde{q}^{-1}} > n^{\frac{1}{q_{\alpha,\beta}}}.$$

Thus the assumptions for both Lemma 12 and Lemma 14 are satisfied.

We then consider the case $\beta + p \leq \alpha$. We now apply Assumption (EMB) and Theorem 6 to $\alpha'$ instead of $\alpha$. We obtain that the inclusion map $I_\pi^{q_{\alpha',\beta}}$ is bounded, where we recall that $q_{\alpha',\beta}$ is defined to be $\frac{2\alpha'}{\alpha'-\beta}$. Since $x \mapsto \frac{2x}{x-\beta}$ is monotonically decreasing for $x > \beta$, we obtain the inequality

$$\frac{2\alpha'}{\alpha'-\beta} > \frac{2\alpha}{\alpha-\beta}.$$

We choose $t = n^{\frac{1}{q_{\alpha,\beta}}}$. By construction, $t$ satisfies the assumptions in Lemma 14. Furthermore, the assumptions of Lemma 12 are satisfied, with $F_* \in L_{q'}(\pi, \mathcal{Y})$, and $t = n^{\frac{1}{q_{\alpha,\beta}}}$.

Having established the applicability of Lemma 14 and Lemma 12, let us turn our attention to proving the results of the Theorem. Denote

$$\xi(x,y) = (y - C_\lambda \phi(x)) \otimes \left( C_{X,\lambda}^{-\frac{1}{2}} \phi(x) \right)$$

We compute

$$\mathbb{E}[\xi(x,y)] = (C_{YX} - C_\lambda C_X) C_{X,\lambda}^{-\frac{1}{2}}$$

1. The $\beta + p > \alpha$ case. Have

$$\left\| \frac{1}{n} \sum_{i=1}^n \xi_i - \mathbb{E}[\xi(x,y)] \right\|_{S_2(\mathcal{H},\mathcal{Y})} \leq \left\| \frac{1}{n} \sum_{i=1}^n \xi_i \mathbb{1}\{x_i \in \Omega_t\} - \mathbb{E}[\xi(x,y)\mathbb{1}\{x \in \Omega_t\}] \right\|_{S_2(\mathcal{H},\mathcal{Y})}$$
$$+ \left\| \frac{1}{n} \sum_{i=1}^n \xi_i \mathbb{1}\{x_i \in \Omega_t^c\} \right\|_{S_2(\mathcal{H},\mathcal{Y})}$$
$$+ \|\mathbb{E}[\xi(x,y)\mathbb{1}\{x \in \Omega_t^c\}\|_{S_2(\mathcal{H},\mathcal{Y})}$$

We can bound the first term with probability at least $1 - 2e^{-\tau}$ by $c\tau \left( \frac{n}{\log^\theta(n)} \right)^{-\frac{\beta}{2\alpha}}$, according to Lemma 14. By Lemma 12, with probability at least $1 - e^{-\tau}$, for sufficiently large $n$, $x_i \in \Omega_t$ for all $i \in [n]$, whereby the second term is zero. It remains to bound the third term, where our bound will be deterministic. Using Jensen's inequality, we have,

$$\|\mathbb{E}[\xi(x,y)\mathbb{1}\{x \in \Omega_t^c\}]\|_{S_2(\mathcal{H},\mathcal{Y})} \leq \mathbb{E}\left[ \|\xi(x,y)\mathbb{1}\{x \in \Omega_t^c\}\|_{S_2(\mathcal{H},\mathcal{Y})} \right]$$
$$= \mathbb{E}\left[ \|(y - C_\lambda \phi(x))\mathbb{1}\{x \notin \Omega_t\}\|_{\mathcal{Y}} \cdot \left\| C_{X,\lambda}^{-\frac{1}{2}} \phi(x) \right\|_{\mathcal{H}} \right]$$
$$\leq A\lambda^{-\frac{\alpha}{2}} \mathbb{E}\left[ \|(y - C_\lambda \phi(x))\mathbb{1}\{x \notin \Omega_t\}\|_{\mathcal{Y}} \right]$$

where in the third line we used Lemma 17. We first split the second term into an approximation error and a noise term using triangle inequality.

$$\mathbb{E}\left[ \|(y - C_\lambda \phi(x))\mathbb{1}\{x \notin \Omega_t\}\|_{\mathcal{Y}} \right] \leq \mathbb{E}\left[ \|(y - F_*(x))\mathbb{1}\{x \notin \Omega_t\}\|_{\mathcal{Y}} \right] + \mathbb{E}\left[ \|(F_*(x) - F_\lambda(x))\mathbb{1}\{x \notin \Omega_t\}\|_{\mathcal{Y}} \right]$$

We bound the first term using the tower property of conditional expectation,

$$\mathbb{E}\left[ \|(y - C_\lambda \phi(x))\mathbb{1}\{x \notin \Omega_t\}\|_{\mathcal{Y}} \right] \leq \mathbb{E}_\pi \left[ \mathbb{E}[\|y - C_\lambda \phi(x)\|_{\mathcal{Y}} \mid x]\mathbb{1}\{x \notin \Omega_t\} \right]$$
$$\leq \mathbb{E}_\pi \left[ \mathbb{E}[\|y - C_\lambda \phi(x)\|_{\mathcal{Y}}^2 \mid x]^{\frac{1}{2}}\mathbb{1}\{x \notin \Omega_t\} \right]$$
$$\leq \sigma\pi(x \notin \Omega_t)$$
$$\leq \frac{\sigma c_q^q}{t^q}.$$

where in the third inequality we used Assumption (MOM) with $q = 2$, and in the fourth inequality we used Lemma 12. We bound the second term using Cauchy-Schwarz inequality and Lemma 10 with $\gamma = 0$,

$$\mathbb{E}\left[ \|(F_*(x) - F_\lambda(x))\mathbb{1}\{x \notin \Omega_t\}\|_{\mathcal{Y}} \right] \leq \mathbb{P}(x \notin \Omega_t)^{\frac{1}{2}} \|F_*\|_\beta \omega_\rho \lambda^{\frac{\beta}{2}}$$

Therefore, using Lemma 12, we have,

$$\|\mathbb{E}[\xi(x,y)\mathbb{1}\{x \in \Omega_t^c\}\|_{S_2(\mathcal{H},\mathcal{Y})} \leq A\lambda^{-\frac{\alpha}{2}} \left( \frac{\sigma c_q^q}{t^q} + \frac{c_q^{\frac{q}{2}}}{t^{\frac{q}{2}}} \|F_*\|_\beta \omega_\rho \lambda^{\frac{\beta}{2}} \right). \tag{36}$$

We now plug in $\lambda \asymp n^{-\frac{1}{\beta+p}}$. Recall that by construction $t > n^{\frac{1}{q}}$. Thus,

$$\lambda^{-\frac{\alpha}{2}}t^{-q} \lesssim n^{-1}n^{\frac{\alpha}{2(\beta+p)}} < n^{-1}n^{\frac{\beta+p}{2(\beta+p)}} = n^{-\frac{1}{2}} \leq n^{-\frac{1}{2}\frac{\beta}{\beta+p}}$$

$$\lambda^{\frac{\beta-\alpha}{2}}t^{-\frac{q}{2}} \lesssim n^{-\frac{1}{2}}n^{\frac{-(\beta-\alpha)}{2(\beta+p)}} < n^{-\frac{1}{2}}n^{\frac{p}{2(\beta+p)}} = n^{-\frac{1}{2}\frac{\beta}{\beta+p}}$$

We've therefore proved inequality (34).

2. The $\beta + p \leq \alpha$ case. We proceed similarly to the $\beta + p > \alpha$ case. We have,

$$\left\|\frac{1}{n}\sum_{i=1}^{n}\xi_i - \mathbb{E}[\xi(x,y)]\right\|_{S_2(\mathcal{H},\mathcal{Y})} \leq \left\|\frac{1}{n}\sum_{i=1}^{n}\xi_i\mathbb{1}\{x_i \in \Omega_t\} - \mathbb{E}[\xi(x,y)\mathbb{1}\{x \in \Omega_t\}]\right\|_{S_2(\mathcal{H},\mathcal{Y})}$$

$$+ \left\|\frac{1}{n}\sum_{i=1}^{n}\xi_i\mathbb{1}\{x_i \in \Omega_t^c\}\right\|_{S_2(\mathcal{H},\mathcal{Y})}$$

$$+ \|\mathbb{E}[\xi(x,y)\mathbb{1}\{x \in \Omega_t^c\}]\|_{S_2(\mathcal{H},\mathcal{Y})}$$

We can bound the first term with probability at least $1 - 2e^{-\tau}$ by $c\tau\left(\frac{n}{\log^\theta(n)}\right)^{-\frac{\beta}{2\alpha}}$, according to Lemma 14. By Lemma 12, with probability at least $1 - e^{-\tau}$, for sufficiently large $n$, $x_i \in \Omega_t$ for all $i \in [n]$, whereby the second term is zero. We bound the third term by Eq. (36). We now plug in $\lambda \asymp \left(\frac{n}{\log^\theta(n)}\right)^{-\frac{1}{\alpha}}$. Recall that by construction $t > n^{\frac{1}{q}}$. Thus,

$$\lambda^{-\frac{\alpha}{2}}t^{-q} \lesssim n^{-1}\left(\frac{n}{\log^\theta(n)}\right)^{\frac{1}{2}} < \left(\frac{n}{\log^\theta(n)}\right)^{-\frac{1}{2}} \leq \left(\frac{n}{\log^\theta(n)}\right)^{-\frac{\beta}{2\alpha}}$$

$$\lambda^{\frac{\beta-\alpha}{2}}t^{-\frac{q}{2}} \lesssim n^{-\frac{1}{2}}\left(\frac{n}{\log^\theta(n)}\right)^{\frac{\alpha-\beta}{2\alpha}} < \left(\frac{n}{\log^\theta(n)}\right)^{\frac{-\beta}{2\alpha}}$$

We have therefore proved inequality (35). $\qquad\square$

We adapt [54, Theorem 16] to the vector-valued setting.

**Theorem 9** (Bound of estimation error). *Suppose that assumptions* (EMB), (EVD), (MOM) *and* (SRC) *hold for $0 \leq \beta \leq 2\rho$, where $\rho$ is the qualification, and $p \leq \alpha < 1$. For $0 \leq \gamma \leq 1$, with $\gamma \leq \beta$,*

1. *In the case of $\beta + p > \alpha$, by choosing $\lambda \asymp n^{-\frac{1}{\beta+p}}$, for any fixed $\tau \geq \log(4)$, when $n$ is sufficiently large, with probability at least $1 - 4e^{-\tau}$, we have*

$$\|[\hat{C}_\lambda - C_\lambda]\|_{S_2([\mathcal{H}]^\gamma,\mathcal{Y})}^2 \leq c\tau^2 n^{-\frac{\beta-\gamma}{\beta+p}},$$

   *where $c$ is a constant independent of $n, \tau$.*

2. *In the case of $\beta + p \leq \alpha$, by choosing $\lambda \asymp \left(\frac{n}{\log^\theta(n)}\right)^{-\frac{1}{\alpha}}$, for any fixed $\tau \geq \log(4)$, when $n$ is sufficiently large, with probability at least $1 - 4e^{-\tau}$, we have*

$$\|[\hat{C}_\lambda - C_\lambda]\|_{S_2([\mathcal{H}]^\gamma,\mathcal{Y})}^2 \leq c\tau^2 \left(\frac{n}{\log^\theta(n)}\right)^{-\frac{\beta-\gamma}{\alpha}}$$

   *where $c$ is a constant independent of $n, \tau$.*

*Proof.* Firstly, we establish the applicability of Lemma 19.

1. The $\beta + p > \alpha$ case. Have $\lambda \asymp n^{-\frac{1}{\beta+p}}$, hence

$$n\lambda^\alpha \gtrsim n^{\frac{\beta+p-\alpha}{\beta+p}}$$

whereas using $\lambda \leq \|C_X\|_{\mathcal{H}\to\mathcal{H}}$ for sufficiently large $n$, as well as Lemma 16,

$$8A^2\tau\log\left(2e\mathcal{N}(\lambda)\frac{\|C_X\|_{\mathcal{H}\to\mathcal{H}}+\lambda}{\|C_X\|_{\mathcal{H}\to\mathcal{H}}}\right) \leq 8A^2\tau\log(4ec_{2,1}\lambda^{-p}) \lesssim 8A^2\tau\left(\log(4ec_{2,1}) + \frac{p}{\beta+p}\log(n)\right)$$

Therefore, for a fixed $\tau > 0$, for all sufficiently large $n$, Eq. (40) in Lemma 19 is satisfied.

2. The $\beta + p \leq \alpha$ case. Have $\lambda \asymp \left(\frac{n}{\log^\theta(n)}\right)^{-\frac{1}{\alpha}}$ for some $\theta > 1$, hence

$$n\lambda^\alpha \geq \log^\theta(n)$$

whereas similar to the $\beta + p > \alpha$ case, we ahve

$$8A^2\tau \log\left(2e\mathcal{N}(\lambda)\frac{\|C_X\|_{\mathcal{H}\to\mathcal{H}} + \lambda}{\|C_X\|_{\mathcal{H}\to\mathcal{H}}}\right) \lesssim 8A^2\tau\left(\log(4ec_{2,1}) + \frac{p}{\alpha}\log\left(\frac{n}{\log^\theta(n)}\right)\right)$$

Therefore, for a fixed $\tau > 0$, for all sufficiently large $n$, Eq. (40) in Lemma 19 is satisfied.

We thus conclude for all $\alpha \in (0,1]$, with probability $\geq 1 - 2e^{-\tau}$, Eq. (41) and (42) are satisfied simultaneously.

We exploit the following decomposition

$$\left\|[\hat{C}_\lambda - C_\lambda]\right\|_{S_2([\mathcal{H}]^\gamma, \mathcal{Y})} \leq \left\|(\hat{C}_\lambda - C_\lambda) C_X^{\frac{1-\gamma}{2}}\right\|_{S_2(\mathcal{H}, \mathcal{Y})}$$

$$\leq \left\|(\hat{C}_\lambda - C_\lambda) \hat{C}_{X,\lambda}^{\frac{1}{2}}\right\|_{S_2(\mathcal{H},\mathcal{Y})} \cdot \left\|\hat{C}_{X,\lambda}^{-\frac{1}{2}} C_{X,\lambda}^{\frac{1}{2}}\right\|_{\mathcal{H}\to\mathcal{H}} \cdot \left\|C_{X,\lambda}^{-\frac{1}{2}} C_X^{\frac{1-\gamma}{2}}\right\|_{\mathcal{H}\to\mathcal{H}}$$

$$\leq \left\|(\hat{C}_\lambda - C_\lambda) \hat{C}_{X,\lambda}^{\frac{1}{2}}\right\|_{S_2(\mathcal{H},\mathcal{Y})} \cdot 3 \cdot \sup_{i\in\mathbb{N}} \frac{\mu_i^{\frac{1-\gamma}{2}}}{\sqrt{\mu_i + \lambda}}$$

$$\leq \left\|(\hat{C}_\lambda - C_\lambda) \hat{C}_{X,\lambda}^{\frac{1}{2}}\right\|_{S_2(\mathcal{H},\mathcal{Y})} \cdot 3\lambda^{-\frac{\gamma}{2}}, \tag{37}$$

where in the first inequality we used Lemma 22, in the third inequality we used Eq. (42) and in the last inequality we used Lemma 21. We consider the following decomposition

$$\hat{C}_\lambda - C_\lambda = \hat{C}_\lambda - C_\lambda\left(\hat{C}_X g_\lambda\left(\hat{C}_X\right) + r_\lambda\left(\hat{C}_X\right)\right)$$

$$= \left(\hat{C}_{YX} - C_\lambda\hat{C}_X\right) g_\lambda\left(\hat{C}_X\right) - C_\lambda r_\lambda\left(\hat{C}_X\right).$$

Hence

$$\left\|[\hat{C}_\lambda - C_\lambda]\right\|_{S_2([\mathcal{H}]^\gamma, \mathcal{Y})}^2 \leq 18\lambda^{-\gamma}\left((I)^2 + (II)^2\right),$$

where

$$(I) = \left\|\left(\hat{C}_{YX} - C_\lambda\hat{C}_X\right) \hat{C}_{X,\lambda}^{\frac{1}{2}} g_\lambda\left(\hat{C}_X\right)\right\|_{S_2(\mathcal{H},\mathcal{Y})}$$

$$(II) = \left\|C_\lambda r_\lambda\left(\hat{C}_X\right) \hat{C}_{X,\lambda}^{\frac{1}{2}}\right\|_{S_2(\mathcal{H},\mathcal{Y})}.$$

**Term (I)**. The high level idea is to bound (I) by exploiting the first axiom of the filter function (8), where $g_\lambda(\hat{C}_X)$ is intuitively a regularized inverse of $\hat{C}_X$, by grouping it with $\hat{C}_{X,\lambda}$.

$$(I) \leq \left\|\left(\hat{C}_{YX} - C_\lambda\hat{C}_X\right) C_{X,\lambda}^{-\frac{1}{2}}\right\|_{S_2(\mathcal{H},\mathcal{Y})} \cdot \left\|C_{X,\lambda}^{\frac{1}{2}}\hat{C}_{X,\lambda}^{-\frac{1}{2}}\right\|_{\mathcal{H}\to\mathcal{H}} \cdot \left\|\hat{C}_{X,\lambda} g_\lambda\left(\hat{C}_X\right)\right\|_{\mathcal{H}\to\mathcal{H}}$$

$$\leq \left\|\left(\hat{C}_{YX} - C_\lambda\hat{C}_X\right) C_{X,\lambda}^{-\frac{1}{2}}\right\|_{S_2(\mathcal{H},\mathcal{Y})} \cdot \sqrt{3} \sup_{t\in[0,\kappa^2]}(t + \lambda)g_\lambda(t)$$

$$\leq \left\|\left(\hat{C}_{YX} - C_\lambda\hat{C}_X\right) C_{X,\lambda}^{-\frac{1}{2}}\right\|_{S_2(\mathcal{H},\mathcal{Y})} \cdot 2\sqrt{3}E.$$

where the second inequality follows from Eq. (42). We consider the following decomposition

$$\left\|\left(\hat{C}_{YX} - C_\lambda\hat{C}_X\right) C_{X,\lambda}^{-\frac{1}{2}}\right\|_{S_2(\mathcal{H},\mathcal{Y})}^2 \leq 2\left\|\left(\left(\hat{C}_{YX} - C_\lambda\hat{C}_X\right) - (C_{YX} - C_\lambda C_X)\right) C_{X,\lambda}^{-\frac{1}{2}}\right\|_{S_2(\mathcal{H},\mathcal{Y})}^2$$

$$+ 2\left\|(C_{YX} - C_\lambda C_X) C_{X,\lambda}^{-\frac{1}{2}}\right\|_{S_2(\mathcal{H},\mathcal{Y})}^2$$

We bound the first term by Theorem 8 and the second term by Lemma 8. This yields, for $\tau \geq \log(4)$, with probability at least $1 - 4e^{-\tau}$, for some constant $c > 0$ which does not depend on $n, \tau, \lambda$,

$$\left\| \left( \hat{C}_{YX} - C_\lambda \hat{C}_X \right) C_{X,\lambda}^{-\frac{1}{2}} \right\|_{S_2(\mathcal{H},\mathcal{Y})}^2 \leq 2\omega_\rho^2 \|F_*\|_\beta^2 \lambda^\beta + \begin{cases} c\tau^2 n^{-\frac{\beta}{\beta+p}} & \beta + p \geq \alpha \\ c\tau^2 \left( \frac{n}{\log^\theta(n)} \right)^{-\frac{\beta}{\alpha}} & \beta + p < \alpha \end{cases}$$

$$\leq \begin{cases} \tau^2 (c + 2\|F_*\|_\beta^2 \omega_\rho^2 \lambda^\beta) n^{-\frac{\beta}{\beta+p}} & \beta + p \geq \alpha \\ \tau^2 (c + 2\|F_*\|_\beta^2 \omega_\rho^2 \lambda^\beta) \left( \frac{n}{\log^\theta(n)} \right)^{-\frac{\beta}{\alpha}} & \beta + p < \alpha \end{cases}$$

where we used that $\tau > 1$. So collecting all the relevant constants together, we can write the upper bound of term (I) as follows: with probability at least $1 - 4e^{-\tau}$, for some constant $c' > 0$ (different from the $c$ before) which does not depend on $n, \tau, \lambda$, we have

$$(I) \leq c'\tau \cdot \begin{cases} n^{-\frac{1}{2}\frac{\beta}{\beta+p}} & \beta + p \geq \alpha \\ \left( \frac{n}{\log^\theta(n)} \right)^{-\frac{\beta}{2\alpha}} & \beta + p < \alpha. \end{cases}$$

**Term (II)**. Using Lemma 9, we have

$$(II) \leq B \left\| \hat{C}_{X,\lambda}^{\frac{1}{2}} r_\lambda(\hat{C}_X) g_\lambda(C_X) C_X^{\frac{\beta+1}{2}} \right\|_{\mathcal{H}\to\mathcal{H}}$$

The second term is the same as the scalar-valued case, which is bounded in Step 3 of the proof of [54, Theorem 16]. We define

$$\Delta_1 := 32 \max\left\{ \frac{\beta-1}{2}, 1 \right\} E\omega_\rho \kappa^{\beta-1} \lambda^{\frac{1}{2}} n^{-\frac{\min(\beta,3)-1}{4}}$$

By the proof of [54, Theorem 16], we have, with probability at least $1 - 6e^{-\tau}$

$$(II) \leq 6B\omega_\rho E \lambda^{\frac{\beta}{2}} + \Delta_1 B\tau \mathbb{1}\{\beta > 2\}.$$

1. Case $\beta + p > \alpha$. In this case $\lambda \asymp n^{-\frac{1}{\beta+p}}$. We note that for $\beta > 2$, $\Delta_1$ as a function of $n$ can be written as

$$\Delta_1 \asymp n^{-\frac{1}{2(\beta+p)} - \frac{\min(\beta,3)-1}{4}}$$

Note that

$$\frac{1}{2(\beta+p)} + \frac{\min(\beta,3)-1}{4} - \frac{\beta}{2(\beta+p)} = \frac{1}{2}\left( \frac{p}{\beta+p} + \frac{\min(\beta,3)-1}{2} \right) > 0$$

Hence

$$\Delta_1 \lesssim n^{-\frac{\beta}{2(\beta+p)}}$$

Therefore we have shown that there exists some constant $c'' > 0$, independent of $n, \lambda, \tau$, such that with probability at least $1 - 6e^{-\tau}$, for sufficiently large $n$,

$$\left\| \left[ \hat{C}_\lambda - C_\lambda \right] \right\|_{S_2([\mathcal{H}]^\gamma,\mathcal{Y})} \leq c''\tau n^{-\frac{1}{2}\frac{\beta-\gamma}{\beta+p}}.$$

2. Case $\beta + p \leq \alpha$. In this case $\beta \leq \alpha \leq 1$, and $\lambda \asymp \left( \frac{n}{\log^\theta(n)} \right)^{-\frac{1}{\alpha}}$. We have also shown that there exists some constant $c'' > 0$, independent of $n, \lambda, \tau$, such that with probability at least $1 - 6e^{-\tau}$, for sufficiently large $n$,

$$\left\| \left[ \hat{C}_\lambda - C_\lambda \right] \right\|_{S_2([\mathcal{H}]^\gamma,\mathcal{Y})} \leq c''\tau \left( \frac{n}{\log^\theta(n)} \right)^{-\frac{\beta-\gamma}{2\alpha}}$$

$\square$

Putting together Lemma 10 and Theorem 9, we have proved Theorem 4.

# D  Auxiliary Results

## D.1  Spectral Calculus, Proof of Proposition 1 and Empirical Solution

**Definition 9** (Spectral Calculus; see 16, Chapter 2.3). *Let $H$ be a Hilbert space. Consider $g : \mathbb{R} \to \mathbb{R}$ and a self-adjoint compact operator $A : H \to H$ admitting a spectral decomposition written as*

$$A = \sum_{i \in I} \lambda_i h_i \otimes h_i.$$

*We then define $g(A) : H \to H$ as*

$$g(A) := \sum_{i \in I} g(\lambda_i) h_i \otimes h_i$$

*whenever this series converges in operator norm.*

*Proof of Proposition 1.* We define the sampling operator $S : \mathbb{R}^n \to \mathcal{H}$ and it dual $S^* : \mathcal{H} \to \mathbb{R}^n$,

$$S : \mathbb{R}^n \to \mathcal{H}, \qquad\qquad\qquad S^* : \mathcal{H} \to \mathbb{R}^n,$$

$$\alpha \mapsto \sum_{i=1}^n \alpha_i \phi(x_i) \qquad\qquad\qquad f \mapsto (f(x_i))_{i=1}^n$$

We can verify that $\hat{C}_X = n^{-1} S S^*$ and $\mathbf{K} = S^* S$. Let $\mathbf{Y} = (y_i)_{i=1}^n \in \mathbb{R}^n$. We have, for all $x \in \mathcal{X}$,

$$
\begin{aligned}
\hat{F}_\lambda(x) &= \hat{C}_\lambda \phi(X) \\
&= \left( \frac{1}{n} \sum_{i=1}^n y_i \otimes \phi(x_i) \right) g_\lambda(\hat{C}_X) \phi(X) \\
&= \sum_{i=1}^n y_i \left\langle \phi(x_i), \frac{1}{n} g_\lambda(n^{-1} S S^*) \phi(X) \right\rangle_{\mathcal{H}} \\
&= \mathbf{Y}^T S^* \left( \frac{1}{n} g_\lambda(n^{-1} S S^*) \phi(X) \right) && (38) \\
&= \mathbf{Y}^T \frac{1}{n} g_\lambda(n^{-1} S^* S) S^* \phi(X) && (39) \\
&= \mathbf{Y}^T \frac{1}{n} g_\lambda(n^{-1} \mathbf{K}) S^* \phi(X) \\
&= \mathbf{Y}^T \frac{1}{n} g_\lambda(n^{-1} \mathbf{K}) \mathbf{k}_x.
\end{aligned}
$$

To go from (38) to (39), we make the following observation. Consider the singular value decomposition of the compact operator $S$, there is $m \leq n$ such that

$$S = \sum_{i=1}^m \sqrt{\sigma_i} f_i \otimes e_i$$

where $(e_i)_i, (f_i)_i$ are orthonormal sequences in $\mathbb{R}^n$ and $\mathcal{H}$ respectively. We then have

$$SS^* = \sum_{i=1}^m \sigma_i f_i \otimes f_i, \quad S^* S = \sum_{i=1}^m \sigma_i e_i \otimes e_i.$$

Therefore, we deduce

$$
\begin{aligned}
S^* g_\lambda\left( \frac{SS^*}{n} \right) &= \left( \sum_i \sqrt{\sigma_i} e_i \otimes f_i \right) \left( \sum_j g_\lambda\left( \frac{\sigma_j}{n} \right) f_j \otimes f_j \right) \\
&= \sum_{i,j} g_\lambda\left( \frac{\sigma_j}{n} \right) \sqrt{\sigma_i} e_i \otimes f_j \langle f_i, f_j \rangle_{\mathcal{H}} \\
&= \sum_i g_\lambda\left( \frac{\sigma_i}{n} \right) \sqrt{\sigma_i} e_i \otimes f_i
\end{aligned}
$$

Similarly,

$$g_\lambda\left(\frac{S^*S}{n}\right)S^* = \left(\sum_j g_\lambda\left(\frac{\sigma_j}{n}\right)e_j \otimes e_j\right)\left(\sum_i \sqrt{\sigma_i}e_i \otimes f_i\right)$$

$$= \sum_{i,j} g_\lambda\left(\frac{\sigma_j}{n}\right)\sqrt{\sigma_i}e_j \otimes f_i \langle e_i, e_j\rangle_{\mathbb{R}^n}$$

$$= \sum_i g_\lambda\left(\frac{\sigma_i}{n}\right)\sqrt{\sigma_i}e_i \otimes f_i$$

Hence we have proved

$$S^* g_\lambda\left(\frac{SS^*}{n}\right) = g_\lambda\left(\frac{S^*S}{n}\right)S^*$$

as desired. $\qquad\square$

**Proposition 2.** *Any minimizer $F \in \mathcal{G}$ of*

$$\mathcal{E}_n(F) := \frac{1}{n}\sum_{i=1}^n \|y_i - F(x_i)\|_{\mathcal{Y}}^2$$

*on $\mathcal{G}$ must satisfy*

$$\hat{C}_{YX} = \hat{C}\hat{C}_X, \qquad C \in S_2(\mathcal{H}, \mathcal{Y}),$$

*where $F(\cdot) = C\phi(\cdot)$.*

*Proof.* By Corollary 1, it is equivalent to solve the following optimization problem on $S_2(\mathcal{H}, \mathcal{Y})$,

$$\min_{C \in S_2(\mathcal{H},\mathcal{Y})} \frac{1}{n}\sum_{i=1}^n \|y_i - C\phi(x_i)\|_{\mathcal{Y}}^2.$$

Recall for a Hilbert-Schmidt operator $L \in S_2(\mathcal{H}, \mathcal{Y})$, we have

$$\langle L, a \otimes b\rangle_{S_2(\mathcal{H},\mathcal{Y})} = \langle a, Lb\rangle_{\mathcal{Y}}.$$

Using this, we re-write the objective as an inner product in $S_2(\mathcal{H}, \mathcal{Y})$:

$$\frac{1}{n}\sum_{i=1}^n \|y_i - C\phi(x_i)\|_{\mathcal{Y}}^2 = \frac{1}{n}\sum_{i=1}^n -2\langle C, y_i \otimes \phi(x_i)\rangle_{S_2(\mathcal{H},\mathcal{Y})} + \langle C, (C\phi(x_i)) \otimes \phi(x_i)\rangle_{S_2(\mathcal{H},\mathcal{Y})} + \text{constant}$$

$$= -2\langle C, \hat{C}_{YX}\rangle_{S_2(\mathcal{H},\mathcal{Y})} + \langle C, C\hat{C}_X\rangle_{S_2(\mathcal{H},\mathcal{Y})}$$

Taking the Fréchet derivative with respect to $C$ and setting in to zero, we obtain the following first order condition

$$\hat{C}_{YX} = C\hat{C}_X$$

$\qquad\square$

### D.2 Properties Related to Assumptions (EMB) and (EVD)

**Lemma 15** (Lemma 13 [17])**.** *Under* (EMB)*, the following inequality is satisfied, for $\lambda > 0$ and $\pi$-almost all $x \in \mathcal{X}$,*

$$\left\|(C_X + \lambda Id_{\mathcal{H}})^{-\frac{1}{2}} k(x, \cdot)\right\|_{\mathcal{H}} \le A\lambda^{-\frac{\alpha}{2}}.$$

**Definition 10** (*l*-effective dimension)**.** *For $l \ge 1$, the l-effective dimension $\mathcal{N}_l : (0, \infty) \to [0, \infty)$ is defined by*

$$\mathcal{N}_l(\lambda) := \text{Tr}\left[C_X^l C_{X,\lambda}^{-l}\right] = \sum_{i \ge 1}\left(\frac{\mu_i}{\mu_i + \lambda}\right)^l$$

The 1-effective dimension is widely considered in the statistical analysis of kernel ridge regression (see [6], [5], [32], [34], [17]). The following lemma provides upper and lower bounds for the $l$−effective dimension.

**Lemma 16.** *Suppose Assumption* (EVD) *holds with parameter* $p \in (0,1]$, *for any* $\lambda \in (0,1]$, *there exists a constant* $c_{2,l} > 0$ *independent of* $\lambda$ *such that*

$$\mathcal{N}_l(\lambda) \le c_{2,l}\lambda^{-p}.$$

*If furthermore, Assumption* (EVD+) *holds with parameter* $p \in (0,1)$, *for any* $\lambda \in (0,1]$, *there exists a constant* $c_{1,l} > 0$ *independent of* $\lambda$ *such that*

$$c_{1,l}\lambda^{-p} \le \mathcal{N}_l(\lambda) \le c_{2,l}\lambda^{-p}.$$

The proof can be found in [29] (Proposition D.1), but as the proof is incomplete we provide a full proof for completeness. This allows us to detect that the value $p = 1$ in Assumption (EVD+) is not compatible with the assumption of a bounded kernel (see Remark 7 below).

*Proof.*

$$\mathcal{N}_l(\lambda) \le \sum_{i \ge 1} \left( \frac{D_2}{D_2 + \lambda i^{\frac{1}{p}}} \right)^l \qquad \left( x \mapsto \frac{x}{x + \lambda} \text{ is monotonically increasing} \right)$$

$$\le \int_0^{+\infty} \left( \frac{D_2}{D_2 + \lambda x^{\frac{1}{p}}} \right)^l \mathrm{d}x \qquad \left( i \mapsto \left( \frac{D_2}{D_2 + \lambda i^{1/p}} \right)^l \text{ is positive and decreasing} \right)$$

$$= \int_0^{+\infty} \left( \frac{D_2}{D_2 + y^{\frac{1}{p}}} \right)^l \frac{\mathrm{d}y}{\lambda^p} \qquad (y^{1/p} = \lambda x^{1/p})$$

$$\le \lambda^{-p} \left( 1 + \int_1^{+\infty} \left( \frac{D_2}{D_2 + y^{\frac{1}{p}}} \right)^l \mathrm{d}y \right)$$

Let us now consider the integral. Let us first consider $p \le 1 < l$,

$$\int_1^{\infty} \left( \frac{D_2}{D_2 + y^{\frac{1}{p}}} \right)^l \mathrm{d}y \le D_2^l \int_1^{\infty} y^{-\frac{l}{p}} \mathrm{d}y$$

$$= D_2^l \frac{p}{l - p}$$

Therefore, using $\lambda \le 1$, we can take $c_{2,l} = 1 + D_2^l \frac{p}{l-p}$. The remaining edge case $p = 1 = l$, is covered by [17, Lemma 11] with $c_{2,1} = \|C_X\|_{S_1(\mathcal{H})}$. For the lower bound, we proceed similarly. For $p \in (0,1)$ (and therefore $p < l$),

$$\mathcal{N}_l(\lambda) \ge \sum_{i \ge 1} \left( \frac{D_1}{D_1 + \lambda i^{\frac{1}{p}}} \right)^l \qquad \left( x \mapsto \frac{x}{x + \lambda} \text{ is monotonically increasing} \right)$$

$$\ge \int_1^{\infty} \left( \frac{D_1}{D_1 + \lambda x^{\frac{1}{p}}} \right)^l \mathrm{d}x \qquad \left( i \mapsto \left( \frac{D_1}{D_1 + \lambda i^{1/p}} \right)^l \text{ is positive and decreasing} \right)$$

$$= \int_1^{\infty} \left( \frac{D_1}{D_1 + y^{\frac{1}{p}}} \right)^l \frac{\mathrm{d}y}{\lambda^p} \qquad (y^{1/p} = \lambda x^{1/p})$$

$$\ge \lambda^{-p} \int_1^{\infty} \left( \frac{D_1}{D_1 + 1} \right)^l y^{-\frac{l}{p}} \mathrm{d}y$$

$$= \lambda^{-p} \left( \frac{D_1}{D_1 + 1} \right)^l \frac{p}{l - p}.$$

Therefore, we can take $c_{1,l} = \left( \frac{D_1}{D_1 + 1} \right)^l \frac{p}{l-p}$. $\qquad \square$

**Remark 7.** *We note that Assumption* (EVD+) *with* $p = 1$ *is not compatible with the assumption that* $k$ *is bounded (Assumption 3). Indeed, suppose that* $\mu_i \ge D_1 i^{-1}$, *for all* $i \ge 1$. *Recall that* $\{[e_i]\}_{i \ge 1}$ *forms an orthonormal set in* $L_2(\pi)$. *By Mercer's theorem,*

$$\kappa^2 \ge \int_{\mathcal{X}} k(x,x)\pi(\mathrm{d}x) = \sum_{i \ge 1} \mu_i \int_{\mathcal{X}} e_i(x)^2 \pi(\mathrm{d}x) = \sum_{i \ge 1} \mu_i \ge D_1 \sum_{i \ge 1} i^{-1} = +\infty,$$

*which leads to a contradiction.*

**Lemma 17.** *For any $l \in [1, 2]$, the following equality holds,*

$$\int_{\mathcal{X}} \left\| [C_{X,\lambda}^{-\frac{l}{2}} k(x, \cdot)] \right\|_{2-l}^2 d\pi(x) = \mathcal{N}_l(\lambda).$$

*In particular for $l = 1$,*

$$\int_{\mathcal{X}} \left\| C_{X,\lambda}^{-\frac{1}{2}} k(x, \cdot) \right\|_{\mathcal{H}}^2 d\pi(x) = \mathcal{N}_1(\lambda),$$

*and for $l = 2$,*

$$\int_{\mathcal{X}} \left\| [C_{X,\lambda}^{-1} k(x, \cdot)] \right\|_{L_2(\pi)}^2 d\pi(x) = \mathcal{N}_2(\lambda).$$

*Proof.* Fix $x \in \mathcal{X}$. Since $k(x, \cdot) \in \mathcal{H}$, and $\left\{ \mu_i^{1/2} e_i \right\}_{i \in I}$ is an ONB of $(\ker I_\pi)^\perp$, we have that $\pi-$almost everywhere

$$k(x, \cdot) = \sum_{i \in I} \langle k(x, \cdot), \mu_i^{1/2} e_i \rangle_{\mathcal{H}} \mu_i^{1/2} e_i = \sum_{i \in I} \mu_i e_i(x) e_i.$$

This is Mercer's Theorem [51]. On the other hand, $\pi-$almost everywhere,

$$C_{X,\lambda}^{-l/2} = \sum_{i \in I} (\mu_i + \lambda)^{-l/2} (\sqrt{\mu_i} e_i) \otimes (\sqrt{\mu_i} e_i).$$

Therefore,

$$[C_{X,\lambda}^{-l/2} k(x, \cdot)] = \sum_{i \in I} \frac{\mu_i}{(\mu_i + \lambda)^{l/2}} e_i(x) [e_i],$$

and by Parseval's identity, using that $\{\mu_i^{(2-l)/2} [e_i]\}_{i \in I}$ is an ONB of $[\mathcal{H}]^{2-l}$,

$$\| [C_{X,\lambda}^{-l/2} k(x, \cdot)] \|_{2-l}^2 = \sum_{i \in I} \left( \frac{\mu_i}{\mu_i + \lambda} \right)^l e_i(x)^2$$

Therefore,

$$\int_{\mathcal{X}} \| [C_{X,\lambda}^{-l/2} k(x, \cdot)] \|_{2-l}^2 d\pi(x) = \sum_{i \in I} \left( \frac{\mu_i}{\mu_i + \lambda} \right)^l \int_{\mathcal{X}} e_i(x)^2 d\pi(x) = \mathcal{N}_l(\lambda),$$

where we used that $([e_i])_{i \in I}$ forms an orthonormal set in $L_2(\pi)$. $\qquad\square$

## D.3 Concentration Inequalities

The following Theorem is from [17, Theorem 26].

**Theorem 10** (Bernstein's inequality). *Let $(\Omega, \mathcal{B}, P)$ be a probability space, $H$ be a separable Hilbert space, and $\xi : \Omega \to H$ be a random variable with*

$$\mathbb{E}[\|\xi\|_H^m] \leq \frac{1}{2} m! \sigma^2 L^{m-2}$$

*for all $m \geq 2$. Then, for $\tau \geq 1$ and $n \geq 1$, the following concentration inequality is satisfied*

$$P^n \left( (\omega_1, \ldots, \omega_n) \in \Omega^n : \left\| \frac{1}{n} \sum_{i=1}^n \xi(\omega_i) - \mathbb{E}_P \xi \right\|_H^2 \geq 32 \frac{\tau^2}{n} \left( \sigma^2 + \frac{L^2}{n} \right) \right) \leq 2e^{-\tau}.$$

*In particular, for $\tau \geq 1$ and $n \geq 1$,*

$$P^n \left( (\omega_1, \ldots, \omega_n) \in \Omega^n : \left\| \frac{1}{n} \sum_{i=1}^n \xi(\omega_i) - \mathbb{E}_P \xi \right\|_H \geq 4\sqrt{2} \frac{\tau}{\sqrt{n}} \left( \sigma + \frac{L}{\sqrt{n}} \right) \right) \leq 2e^{-\tau}.$$

**Lemma 18.** *Let $\tau \geq \log(2)$, with probability at least $1 - 2e^{-\tau}$, for $\sqrt{n\lambda} \geq 8\tau\kappa\sqrt{\max\{\mathcal{N}(\lambda), 1\}}$, we have*

$$\left\| \hat{C}_{X,\lambda}^{-1} C_{X,\lambda} \right\|_{\mathcal{H} \to \mathcal{H}} \leq 2.$$

*Proof.* The proof is identical to [5, Proposition 5.4] with the only difference that in their setting $\kappa = 1$. $\qquad\square$

**Proposition 3** (Proposition C.9 29). *Let $\pi$ be a probability measure on $\mathcal{X}$, $f \in L_2(\pi)$ and $\|f\|_\infty \le M$. Suppose we have $x_1, \ldots, x_n$ sampled i.i.d. from $\pi$. Then, for any $\tau \ge \log(2)$, the following holds with probability at least $1 - 2e^{-\tau}$:*

$$\frac{1}{2}\|f\|^2_{L_2(\pi)} - \frac{5\tau M^2}{3n} \le \|f\|^2_{2,n} \le \frac{3}{2}\|f\|^2_{L_2(\pi)} + \frac{5\tau M^2}{3n},$$

*where $\|\cdot\|_{2,n}$ was defined in Definition 7.*

**Lemma 19** (Lemma 12 54). *Let Assumptions* (EMB)*,* (SRC) *and* (MOM) *be satisfied. For $\tau \ge 1$, if $\lambda$ and $n$ satisfy that*

$$n \ge 8A^2 \tau \lambda^{-\alpha} \log\left(2e\mathcal{N}(\lambda)\frac{\|C_X\|_{\mathcal{H}\to\mathcal{H}} + \lambda}{\|C_X\|_{\mathcal{H}\to\mathcal{H}}}\right) \tag{40}$$

*then the following operator norm bounds are satisfied with probability not less than $1 - 2e^{-\tau}$*

$$\left\|C_{X,\lambda}^{-\frac{1}{2}}\hat{C}_{X,\lambda}^{\frac{1}{2}}\right\|^2_{\mathcal{H}\to\mathcal{H}} \le 2, \tag{41}$$

$$\left\|C_{X,\lambda}^{\frac{1}{2}}\hat{C}_{X,\lambda}^{-\frac{1}{2}}\right\|^2_{\mathcal{H}\to\mathcal{H}} \le 3. \tag{42}$$

### D.4    Miscellaneous results

**Lemma 20** (Cordes inequality [18]). *Let $A, B$ be two positive bounded linear operators on a separable Hilbert space $H$ and $s \in [0, 1]$. Then*

$$\|A^s B^s\|_{H\to H} \le \|A\|^s_{H\to H}\|B\|^s_{H\to H}$$

**Lemma 21** (Lemma 25 [17]). *For $\lambda > 0$ and $s \in [0, 1]$, we have*

$$\sup_{t\ge 0}\frac{t^s}{t+\lambda} \le \lambda^{s-1}$$

We recall the following basic Lemma from [30, Lemma 2].

**Lemma 22.** *For $0 \le \gamma \le 1$ and $F \in \mathcal{G}$, the inequality*

$$\|[F]\|_\gamma \le \left\|CC_X^{\frac{1-\gamma}{2}}\right\|_{S_2(\mathcal{H},\mathcal{Y})}$$

*holds, where $C = \bar\Psi^{-1}(F) \in S_2(\mathcal{H},\mathcal{Y})$. If, in addition, $\gamma < 1$ or $C \perp \mathcal{Y} \otimes \ker I_\pi$ is satisfied, then the result is an equality.*

**Definition 11.** *Let $\mathcal{X} \subseteq \mathbb{R}^d$ be a compact set and $\theta \in (0, 1]$. For a function $f : \mathcal{X} \to \mathbb{R}$, we introduce the Hölder semi-norm*

$$[f]_{\theta,\mathcal{X}} := \sup_{x,y\in\mathcal{X},x\ne y}\frac{|f(x) - f(y)|}{\|x-y\|^\theta},$$

*where $\|\cdot\|$ represents the usual Euclidean norm. Then, we define the Hölder space*

$$C^\theta(\mathcal{X}) := \{f : \mathcal{X} \to \mathbb{R} \mid [f]_{\theta,\mathcal{X}} < +\infty\},$$

*which is equipped with the norm*

$$\|f\|_{C^\theta(\mathcal{X})} := \sup_{x\in\mathcal{X}}|f(x)| + [f]_{\theta,\mathcal{X}}.$$

The next lemma is used to prove Lemma 24 below. It appears in [29, Lemma A.3], albeit the use of an erroneous equality in their proof: $\|k(x,\cdot) - k(y,\cdot)\|^2_{\mathcal{H}} = k(x,x)k(y,y) - k(x,y)^2$. We therefore provide our own proof of this result.

**Lemma 23.** *Assume that $\mathcal{H}$ is an RKHS over a compact set $\mathcal{X} \subseteq \mathbb{R}^d$ associated with a kernel $k \in C^\theta(\mathcal{X} \times \mathcal{X})$ for $\theta \in (0, 1]$. Then, we have $\mathcal{H} \subseteq C^{\frac{\theta}{2}}(\mathcal{X})$ and*

$$[f]_{\frac{\theta}{2},\mathcal{X}} \le \sqrt{2[k]_{\theta,\mathcal{X}\times\mathcal{X}}}\|f\|_{\mathcal{H}}$$

*Proof.* For all $(x, y) \in \mathcal{X}$ and $f \in \mathcal{H}$, by the reproducing property and Cauchy–Schwarz inequality,

$$|f(x) - f(y)| = |\langle k(x, \cdot) - k(y, \cdot), f \rangle_{\mathcal{H}}| \leq \|f\|_{\mathcal{H}} \|k(x, \cdot) - k(y, \cdot)\|_{\mathcal{H}}.$$

Then, using $k \in C^{\theta}(\mathcal{X} \times \mathcal{X})$, we obtain

$$\|k(x, \cdot) - k(y, \cdot)\|_{\mathcal{H}}^2 = k(x, x) + k(y, y) - 2k(x, y) \leq 2[k]_{\theta, \mathcal{X} \times \mathcal{X}} \|x - y\|^{\theta},$$

which concludes the proof. $\square$

We derive as a corollary a quantitative upper bound on the $\epsilon$-covering number of the the set of (spectral) regularized kernel basis function with respect to the $\|\cdot\|_{\infty}$ norm.

**Lemma 24** (Lemma C.10 by 29). *Assume that $\mathcal{H}$ is an RKHS over a compact set $\mathcal{X} \subseteq \mathbb{R}^d$ associated with a kernel $k \in C^{\theta}(\mathcal{X} \times \mathcal{X})$ for $\theta \in (0, 1]$. Assume that $k(x, x) \leq \kappa^2$ for all $x \in \mathcal{X}$. Then, we have that for all $\epsilon > 0$,*

$$\mathcal{N}(\mathcal{K}_{\lambda}, \|\cdot\|_{\infty}, \epsilon) \leq c(\lambda\epsilon)^{-\frac{2d}{\theta}}$$

*where $\mathcal{K}_{\lambda} := \left\{ C_{X, \lambda}^{-1} k(x, \cdot) \right\}_{x \in \mathcal{X}}$, and $c$ is a positive constant which does not depend on $\lambda, \epsilon$ and only depends on $\kappa$ and $[k]_{\theta, \mathcal{X} \times \mathcal{X}}$. $\mathcal{N}(\mathcal{K}_{\lambda}, \|\cdot\|_{\infty}, \epsilon)$ denotes the $\epsilon$-covering number of the set $\mathcal{K}_{\lambda}$ in the norm $\|\cdot\|_{\infty}$ (see [49, Definition 6.19] for the definition of covering numbers).*

