# OpenReview forum: "Optimal Rates for Vector-Valued Spectral Regularization Learning Algorithms"
_NeurIPS.cc/2024/Conference — NeurIPS 2024 poster_

### Official Review · Reviewer_48mf · 2024-07-08

**Soundness:** 3
**Presentation:** 3
**Contribution:** 3
**Rating:** 7
**Confidence:** 3

**Summary:**

The paper considers the vector-valued regression problem. Given $n $ iid samples $\{(x\_i,y\_i)\}\_{i=1}^n$ from a distribution $\mathcal{D}$ on $\mathcal{X} \times \mathcal{Y}$, the goal is to output the estimator $\hat{f}$ such that $\mathbb{E}[||\hat{f}(x)-f^{\star}(x)||\_{\mathcal{Y}}^2]$ is small. Here, $f^{\star}$ is the optimal regressor, that is $\mathbb{E}[y \mid x]$. In this work, the authors consider the case where the estimator $\hat{f} $ is obtained through spectral regularization methods such as Tikhonov, hard-thresholding, and iteration filters.

**Strengths:**

- In the RKHS framework, the ridge estimator obtained by Tikhonov regularization is a canonical estimator. However, in Theorem 3, the authors show that the ridge estimator cannot exploit the higher-order smoothness of the optimal regressor  $f^{\star}$ .

- In Theorem 4, the authors show that other spectrally regularized estimators (for example, one based on hard-thresholding) can exploit the higher-order smoothness of the functions.

- The paper also points out some inaccuracies in earlier work. I did not verify the validity of these claims, but if true, this is also an important contribution to the literature.

- Overall, the paper is well-written and is easy to follow.

**Weaknesses:**

The proof of Theorem 4 is fully deferred to the Appendix. Given that the proof is highly technical, it would be helpful for the reader to include a high-level proof sketch in the main text of the paper. For example, some discussion of key challenges in generalizing the proof of scalar-valued to general $\mathcal{Y}$ would be useful.

**Questions:**

If the proof in [1] were correct or can be readily fixed, can't Theorem 3 be inferred immediately through the result in the scalar-valued case? Here is a sketch of the argument:

 Let $\\{d\_j\\}\_{j \in \mathbb{N}}$ be the ONB of $\mathcal{Y}$, and $\mathcal{Y}\_1 = \\{ \langle y, d_1 \rangle \quad |\quad  y \in \mathcal{Y} \\}$ to be the subspace along the direction $d\_1$.  Consider the probability distribution such that $y \mid x$ is only supported over $\mathcal{Y}\_1$.
Suppose $f^{\star}$ is the optimal regressor and $\hat{f}\_{\lambda}$ is the KRR estimator. Then, we have

$$\mathbb{E}[||\hat{f}\_{\lambda}-f^{\star}||\_{\mathcal{Y}}^2] = \int ||\hat{f}\_{\lambda}(x)-f^{\star}(x) ||\_{\mathcal{Y}}^2   p(x, dy)  \pi(dx) \geq \int | \langle \hat{f}\_{\lambda}(x)-f^{\star}(x) ,d_1\rangle |^2  p(x, dy)  \pi(dx) . $$

Now, this is effectively lower bounding the risk for scalar-valued regression. I believe that this argument can be formalized by defining a one-to-one mapping between $\mathcal{Y}\_1 $ and $\mathbb{R}$. I might be missing something here, so please correct me if I am wrong.



[1] Y. Li, H. Zhang, and Q. Lin. On the saturation effect of kernel ridge regression. In The Eleventh 390 International Conference on Learning Representations, 2023.

---

> ### Author Rebuttal · Authors · 2024-08-05
>
> We thank the reviewer for their insightful and helpful feedback.
>
> **Weaknesses**
>
> The key challenge in generalising the proof from the scalar-valued case to the general vector-valued case lies in finding the right way to harmonize the technical definition of vector-valued interpolation space, and using Fourier series computations (see section C.1 in the Appendix) in order to control terms in the vector-valued setting by corresponding terms in the scalar-valued setting. Furthermore, it is essential to find the right decomposition of the learning risk. We appreciate the reviewer's suggestion to include a high-level overview of the proof. For the camera ready version, we will use the extra page allowed to include a proof sketch.
>
> **Questions**
>
> Assuming the proof of [1] was true, we agree with the reviewer that this would be a particularly elegant way to obtain our results through the result in the scalar-valued case. Below, we confirm this by providing the full argument, and discuss a subtle point regarding the noise assumption that we must consider.
>
> Given the vector-valued problem with random variables $(X,Y) \in \mathcal{X} \times \mathcal{Y}$ and Bayes function $F\_* (X) := \mathbb{E}[Y \mid X]$, we consider a modified setting where the target variable is projected along the direction $d\_j$: $\tilde{Y} := \langle Y, d\_j \rangle\_{\mathcal{Y}} \in \mathbb{R}$. We readily see that the Bayes function associated to this setting is $f\_* (X) := \mathbb{E}[\tilde{Y} \mid X] = \langle F\_{\ast},d\_j\rangle\_{\mathcal{Y}}.$ Given our vector-valued estimator $\hat F\_{\lambda}$ (Eq. (7) in the submission), we can verify that $\hat{f}\_{\lambda}(\cdot) := \langle \hat{F}\_{\lambda}(\cdot), d\_j\rangle\_{\mathcal{Y}}$ is the scalar-valued ridge estimator associated to the dataset $\{(x\_i, \tilde{y}\_i)\}\_{i=1}^n \in (\mathcal{X} \times \mathbb{R})^n$ with $\tilde{y}\_i := \langle y\_i, d\_j \rangle\_{\mathcal{Y}}.$ To see this, using that $\hat{F}\_{\lambda}(\cdot) = \hat{C}\_{\lambda}\phi(\cdot)$, with $\hat{C}\_{\lambda}\in S\_2(\mathcal{H},\mathcal{Y})$, we obtain,
> \begin{equation*}
>     \hat{f}\_{\lambda}(\cdot) = \langle \hat{F}\_{\lambda}(\cdot), d\_j\rangle\_{\mathcal{Y}} = \langle \phi(\cdot), \hat{C}\_{\lambda}^{\ast}d\_j\rangle\_{\mathcal{Y}}.
> \end{equation*}
> By Eq. (11) in the current submission, $\hat{C}\_{\lambda} = \frac{1}{n}\sum\_{i=1}^{n}y\_i \otimes \phi(x\_i)\left(\hat{C}\_{XX}+\lambda\right)^{-1}$, therefore,
> \begin{equation*}
>     \hat{C}\_{\lambda}^{\ast}d\_j = \left(\hat{C}\_{XX}+\lambda\right)^{-1}\frac{1}{n}\sum\_{i=1}^{n}\phi(x\_i)\langle y\_i,d\_j\rangle = \left(\hat{C}\_{XX}+\lambda\right)^{-1}\frac{1}{n}\sum\_{i=1}^{n}\phi(x\_i)\tilde{y}\_i,
> \end{equation*}
> which shows exactly that $\langle \hat{f}\_{\lambda}(\cdot),d\_j\rangle$ is the desired scalar-valued ridge estimator. A slightly subtle point is that in order to apply the results of [1] to control the following term
> \begin{equation*}
>     \int\left|\left\langle\hat{F}\_\lambda(x)-F\_{\star}(x), d\_j\right\rangle\right|^2 \pi(dx),
> \end{equation*}
> we must impose the assumption that for $\pi$-almost all $x\in \mathcal{X}$,
> \begin{equation*}
>     \mathbb{E}\_{X,Y}\left[\langle Y - F\_{\ast}(X), d\_j\rangle^2\mid X=x\right] \geq \overline{\sigma}^2,
> \end{equation*}
> for some $\overline{\sigma}>0$. We refer to it as assumption (1). In contrast, the assumption we currently adopt in our paper states that for $\pi$-almost all $x\in \mathcal{X}$,
> \begin{equation*}
>     \mathbb{E}\_{X,Y}\left[\\| Y - F\_{\ast}(X)\\|\_{\mathcal{Y}}^2\mid X=x\right] \geq \overline{\sigma}^2,
> \end{equation*}
> We refer to it as assumption (2). It is clear that (1) implies (2). We now provide an example to show that (2) does not imply (1).
>
> *Example.* Let $\mathcal{Y} = \mathbb{R}^2$ and $\mathcal{X} = \\{0,1\\}$. Let $\pi$ denote the uniform distribution over $\mathcal{X}$.  We define the joint distribution $p(x,y)$ as
> \begin{equation*}
>     p(x,y) = p(y\mid x=0)\frac{1}{2} + p(y\mid x=1)\frac{1}{2}
> \end{equation*}
> where
> \begin{equation*}
>     p(y = (y_1, y_2) \mid x=0) = \frac{1}{2}\delta\_0(y\_1)1[y\_2\in \\{-1,1\\}]
> \end{equation*}
> and
> \begin{equation*}
>     p(y = (y_1, y_2) \mid x=1) = \frac{1}{2}\delta\_0(y\_2)1[y\_1\in \\{-1,1\\}]
> \end{equation*}
> We note that $\mathbb{E}[Y \mid X=0] = \mathbb{E}[Y \mid X=1] = (0, 0)^{T}$. Thus, $F\_{\ast}(x) = \mathbb{E}[Y\mid X=x] =(0, 0)^{T}$ for $x \\in \\{0,1\\}$. On the other hand, writing $Y = (Y\_1,Y\_2)^{T}$, we have
> \begin{align*}
>     &\mathbb{E}[Y\_1^2\mid X = 0] = 0,\quad \mathbb{E}[Y\_2^2\mid X = 0] = \frac{1}{3}\\
>     &\mathbb{E}[Y\_1^2\mid X = 0] = \frac{1}{3},\quad \mathbb{E}[Y\_2^2\mid X = 0] = 0\\
> \end{align*}
> while for all $x\in \\{0,1\\}$, we have
> \begin{equation*}
>     \mathbb{E}[\\|Y \\|^2\mid X = x] = \frac{1}{3}
> \end{equation*}
> This provides the desired example where (2) holds but (1) does not.

---

> > ### Comment · Reviewer_48mf · 2024-08-08
> >
> > Thank you for formalizing my sketch. I will retain my current score.

---

### Official Review · Reviewer_bQuY · 2024-07-09

**Soundness:** 3
**Presentation:** 3
**Contribution:** 2
**Rating:** 6
**Confidence:** 4

**Summary:**

This manuscript presents the excess risk upper bound for spectral regularized algorithms whose output might belong to potential infinite-dimensional Hilbert space. Additionally, the saturation effect for a special case of vector-valued spectral algorithms, KRR, is rigorously confirmed.

**Strengths:**

1. The manuscript is easier to follow as it provides enough detail for moving from scalar-valued RKHS to vector-valued RKHS.
2. The manuscript established a series of well-studied properties/results for spectral algorithms and saturation effect to vector-valued output scenarios, which, to the best of my knowledge, is less concerned and studied in the community.
3. Identify some issues for proving the lower bound in previous work [1] and provide new techniques to handle the bias and variance.

**Weaknesses:**

1. I find that the saturation effect, or more exactly, the lower bounds for real-valued spectral algorithms, is proved in [2], seemingly a following work of [1]. Is there any specific obstacle the authors are facing to prove the lower bounds for the vector-valued spectral algorithms?
2. The (EVD+) condition looks weird to me. The authors state that the lower bound for the eigenvalue depends not only on $p$ but also on the running index $i$, and this is needed for the lower-bound proof. Can authors elaborate more on this? I especially note that [1], who considered the real-valued case, does not have such a requirement of dependence on $i$ in the lower bound. Is this a unique challenge present by the vector-valued setting? Or is this due to your correction for the proof issues in [1]?

[1] Y. Li, H. Zhang, and Q. Lin. On the saturation effect of kernel ridge regression. In The Eleventh International Conference on Learning Representations, 2023.

[2] Li, Yicheng, et al. "Generalization Error Curves for Analytic Spectral Algorithms under Power-law Decay." _arXiv preprint arXiv:2401.01599_ (2024).

**Questions:**

1. Just of independent interest. Consider spectral algorithms with finite qualification context (like KRR). Since there is a gap between the upper and lower bound when the interpolation index $\beta$ is greater than $2\rho$, we can consider the following approach to avoid this gap. Based on your misspecification results, one might consider imposing a kernel whose induced space RKHS is much 'smoother' than the RKHS to which the true function belongs. However, without the prior of the true RKHS, it is hard to pick the imposed kernel to make it 'smooth' enough.

   So, I'm wondering if there are some general approaches to avoid the saturation effect. In practical applications, when one needs to use a specific general algorithm with finite $\rho$, this seems to be an important issue. (I understand this is a theoretical paper, but this just popped up in my head when I read the theorems).

2. Is the $Id_{\mathcal{Y}}$ in Equation (2) the identical element in $\mathcal{Y}$? If so, I think you should define it for completeness.

**Limitations:**

While the authors claim they discussed the limitation in terms of assumptions in the checklist. I don't clearly see them. But, to my knowledge, these assumptions are almost standard in the literature, except the one I raise in W2.

---

> ### Author Rebuttal · Authors · 2024-08-05
>
> **W1.** We thank the reviewer for their encouraging feedback and for providing the reference [Li et al. (2024)]. We were not aware of this work. We think that it may be possible to generalise the techniques in [Li et al. (2024)] to the more challenging vector-valued setting. However, the assumptions and techniques in [Li et al. (2024)] are quite different from our work. In particular, [Li et al. (2024)] Assumption 2 introduced the notion of a "regular RKHS", which is not a standard assumption in the literature. The examples they provide where this assumption holds are restricted to simple compact covariate spaces such as the d-dimensional torus or the d-dimensional unit ball. Hence, we would need to carefully check whether the results in [Li et al. (2024)] can be generalised to work under our weaker assumptions. If it turns out that we can generalise the results of [Li et al. (2024)] to our setting with some simple modifications, we will implement the changes in the camera ready version. If it turns out that it requires substantial technical analysis to harmonise [Li et al. (2024)] and our manuscript, we will include a citation to the approach of [Li et al. (2024)] in our work, and defer the analysis of the saturation effect of spectral algorithms in the vector-valued setting to future work.
>
> **W2.** We thank the reviewer for carefully proofreading our manuscript and bringing this to our attention. The (EVD+) condition as stated in the manuscript contained a typo. It should read instead: For $D_1,D_2>0$ and $p\in (0,1)$ and for all $i\in I$,
>
> $D_1 i^{-\frac{1}{p}} \leq \mu_i \leq D_2i^{-\frac{1}{p}},$
>
> and this is the version we use in the proofs in the appendix. We will make sure to proofread the manuscript and correct the typos in the camera ready version. After correcting the typo, (EVD+) is a standard assumption used in the literature to obtain lower bounds (see for example [6]  or [18]).
>
> **Q1.** We thank the reviewer for their insightful question. In the following discussion we work under strong assumptions that allow us to precisely define what we mean by 'smoothness'. We stress that these assumptions do not likely hold in practice and therefore one needs to be careful when using the concept of 'smoothness'. We consider the setting of [30] (Corollary 2) where `smoothness' takes a formal meaning in terms of weak derivatives: $\mathcal{X}$ is a compact subset of $\mathbb{R}^d$, and $X$ follows a distribution equivalent to the Lebesgue measure on $\mathcal{X}$. Let $W^{s,2}(\mathcal{X};\mathcal{Y})$ denote the vector-valued Sobolev space (see [30] (Definition 3)). If $F \in W^{s,2}(\mathcal{X};\mathcal{Y})$, then all weak derivatives of $F$ of orders up to $r := (r_1,\ldots, r_d) \in \mathbb{N}^d$ where $\|r\|_1\leq s$ exist and are square-integrable. Thus, the larger $s$, the smoother the function $F$.
>
> We emphasize that the smoothness of the function $F_{\ast}$ as measured by the interpolation index $\beta$ depends on the vRKHS $\mathcal{G}$, whereas the smoothness of $F_{\ast}$ as measured by the vSobolev space to which it belongs simply depends on the smoothness index $s$. It is shown in [30] that both notions of smoothness are linked. Let the vRKHS be a Sobolev space: $\mathcal{G} = W^{m,2}(\mathcal{X};\mathcal{Y})$ (this is a vRKHS if $m > d/2$); and let $F_* \in W^{s,2}(\mathcal{X};\mathcal{Y})$ for $s \geq 0$. Then $F_* \in [\mathcal{G}]^{\beta}$ with $\beta = s/m$. Furthermore, it is also shown in [30] that the parameters of (EVD) and (EMB) are $\alpha = p = d/(2m)$. Finally, since $\beta = s/m$, given a qualification $\rho$, the saturation level is $s = 2m\rho$.
> To simplify the discussion, we focus on the $L_2-$error rate ($\gamma=0$). Plugging the values of $\beta$ and $p$ in our rate in Theorem 4, we obtain
> $$
> \\|\hat F_{\lambda} - F_*\\|_{L_2}^2 = O_P\left(n^{-\frac{\min \\{\beta, 2\rho\\}}{\min\\{\beta, 2\rho\\} + p}}\right) = O_P\left(n^{-\frac{\min\\{s, 2\rho m\\}}{\min\\{s, 2\rho m\\} + d/2}}\right) ,
> $$
> This confirms the comment raised by the reviewer: we should impose a kernel whose induced vRKHS is as smooth as possible, as measured by $m$. We note, however, that this reasoning would only be implementable in practice if we could actually control the smoothness of the vRKHS through the kernel. As illustrated above, this is possible if $\mathcal{X}$ is a compact subset of $\mathbb{R}^d$, the input data distribution is equivalent to the Lebesgue measure on $\mathcal{X}$ and for kernels whose induced vRKHS is a Sobolev space, such as the Matérn kernel. Outside this restrictive setting, it is unclear how to interpret 'smoothness' and how we can precisely control the 'smoothness' of the vRKHS through the kernel.
>
> An alternative approach to avoid the saturation effect when learning with a finite qualification spectral algorithm is to increase the qualification $\rho$. This can be achieved by using iterated ridge regression [20] (Section 5.4). The qualification of iterated ridge learner is exactly the number of iterations. As a special case, we recover the fact that vanilla ridge regression has qualification $1$. This approach is popular in the econometrics literature and inverse problem literature. For example, see:
> - Section 3 of S. Darolles, Y. Fan, et al.. Nonparametric Instrumental Regression, Econometrica
> - Section 9 of Z. Li, H. Lan, et al. Regularized DeepIV with Model Selection, arXiv
>
> **Q2.** We thank the reviewer for their question. $\mathrm{Id}_{\mathcal{Y}}$ is indeed the identity map on $\mathcal{Y}$. We will define all relevant notations in the camera ready version.
>
> **Limitations.** We agree with the reviewer that the assumptions we made are standard assumptions in non-parametric kernel regression. As mentioned previously, the one raised in W2 is a typo and the corrected version is standard.
>
> We have done our best to respond to the reviewer's questions. If we have done so, we would be grateful if the reviewer might consider increasing their score.

---

> > ### Comment · Reviewer_bQuY · 2024-08-12
> >
> > I thank the authors for their detailed responses.
> >
> > W1. Thank you for explaining the differences in assumptions between the current manuscript and those in Li et al. (2024). As I am closely following this field, my intention is to understand the technical challenges involved in extending the current results to spectral algorithms, which could be a potential direction for further research.
> >
> > Q1. After reviewing this, I noticed a recent work [1] that controls the smoothness of the RKHS through the Gaussian kernel, i.e., motivated by the fact that the Gaussian kernel is the limit of the Matern kernel, which seems aligned with the need for "controls the smoothness". I think it would be interesting to check if vRKHS with Gaussian kernels can be applied to the setting in the current manuscript.
> >
> > I will retain my score as I'm happier to see a complete story in a paper, i.e., proving the saturation effect for the vector-value spectral algorithm. That being said, the paper is still in good shape, and I think it has made a sufficient contribution to the field. I will also support this paper during the discussion phase with the other reviewers.
> >
> > [1] H. Lin, and M. Reimherr. "Smoothness Adaptive Hypothesis Transfer Learning." Forty-first International Conference on Machine Learning.

---

### Official Review · Reviewer_R42A · 2024-07-12

**Soundness:** 3
**Presentation:** 3
**Contribution:** 2
**Rating:** 6
**Confidence:** 3

**Summary:**

This paper considers the regression task of learning a mapping where both the input space and the output space can potentially be infinite dimensional. The authors formulate the problem setting by proposing a number of assumptions that can be thought of as the vector-valued counterparts of the standard assumptions in (real-valued) kernel regression. In the well-specified regime, the authors show that a saturation effect exists i.e. the Tikhonov regularized regression estimator is provably suboptimal. The same phenomenon is known to exist in the real-valued setting, but extension to vector-valued output is a novel contribution. Finally, the authors show that for estimators based on a class of filter functions, one can establish error rates that match the best-known upper bounds even in the real-valued setting.

**Strengths:**

1. The paper is well written. Though the authors consider an extension of previous works where the output space is R, all relevant notions and assumptions in the current setting are rigorously defined.

2. The main result of this paper addresses the optimality (at least for some set of smoothness parameters) of a class of regression estimators based on filter functions, going beyond Tikhonov regularization that most often appears in existing literature.

**Weaknesses:**

1. While the mathematical parts of this paper look sound, and the results appear to be novel, both the error rates and the proof techniques seem to have no difference with the finite-dimensional output setting. I haven't gone through all the proof details and it might be the case that additional challenges arise in the vector-valued setting. If this is the case, then it would be great if the authors can point to the most challenging parts of the proof with several sentences in the main part of the paper.

2. The main part of the paper ends a little abruptly -- it would be better to write a final section summarizing the contributions of the paper and discussing potential future directions. The problem setting and necessary assumptions may be introduced in a more concise way.

**Questions:**

Does the lower bound for real-valued output ($n^{-\frac{\max\{\alpha,\beta\}-\gamma}{\max\{\alpha,\beta\}+p}}$) directly imply a lower bound in the vector-valued setting? If this is the case, it would be nice to state it as a theorem after Theorem 4.

---

> ### Author Rebuttal · Authors · 2024-08-05
>
> We thank the reviewer for their encouraging review of our work. We have done our best to respond to the reviewer's questions. If we have done so, we would be grateful if the reviewer might consider increasing their score.
>
> **Weaknesses**
> 1. The additional difficulty with respect to the scalar-valued setting is to handling the vector-valued interpolation spaces. Under the right decomposition, the vector-valued interpolation spaces allow us to reduce some terms to the the scalar-valued case while other terms require new analysis. We will add a proof sketch in the appendix to highlight the additional difficulties.
> 2. We fully agree with the reviewer that a conclusion should be added. We will use the extra page given for the camera ready version to add a concluding discussion and perspective on future works.
>
> **Questions**
>
> We believe the lower bound $n^{-\frac{\max\\{\alpha,\beta\\}-\gamma}{\max\\{\alpha,\beta\\}+p}}$ mentioned by the reviewer comes from [18]. We would first like to mention that this lower bound is obtained under the assumption that the target function is **bounded**. It was shown in [55] that this assumption is not needed. The correct information-theoretic lower bound is therefore the following. Given assumption (EVD+) with parameter $p \in (0,1]$, assumption (MOM), assumption (SRC) with parameter $\beta > 0$, and any $\gamma \in [0,\beta)$, any learning algorithm $\hat {F}$, will satisfy
>
> $\\|\hat F - F_*\\|_{\gamma}^2 = \Omega_P \left(n^{-\frac{\beta- \gamma}{\beta + p}}\right).$
>
> This result can be found in Remark 3 of [18] in the scalar-valued setting and in Theorem 5 of [30] in the vector-valued setting. The  insight of the reviewer is correct: the vector-valued lower bound can be obtained using a reduction to the scalar-valued setting. We will make sure to include this remark in the camera ready version.

---

> > ### Comment · Reviewer_R42A · 2024-08-13
> >
> > Thank you for addressing my questions and concerns. I will maintain my score and stay on the positive size.

---

### Official Review · Reviewer_U1CH · 2024-07-14

**Soundness:** 3
**Presentation:** 3
**Contribution:** 2
**Rating:** 6
**Confidence:** 5

**Summary:**

The  submission explores a class of spectral learning algorithms for regression within the context of supervised learning using random design. The focus is on high-dimensional and potentially infinite-dimensional output spaces. The problem is framed as minimizing the risk associated with the least squares loss, over vector-valued reproducing kernel Hilbert spaces (RKHS).

The authors make several contributions:

1. Saturation Effect of Ridge Regression: The paper rigorously confirms the saturation effect in ridge regression for general Hilbert output spaces.

2. Convergence Rates for Spectral Algorithms: The paper provides upper bounds on the rates of convergence for a broad range of spectral algorithms even in the misspecified learning case where the target function might not be contained within the RKHS. The smoothness of the target function is characterized using interpolation spaces.

**Strengths:**

1. The paper extends non-parametric regression results involving spectral regularization to a broader setting that includes potentially infinite-dimensional output spaces.

2. The authors mostly clearly explain their contributions, making the advancements in the field accessible.

2.They rigorously introduce the mathematical framework of vector-valued reproducing kernel Hilbert spaces (RKHSs) and regression, ensuring a solid theoretical foundation.

3. One particularly commendable aspect of the approach is the expression of the smoothness of the target function in terms of vector-valued interpolation spaces, which appears to be a natural and effective strategy.

4. Rates of convergence are presented that are shown to be optimal in the well-specified case, effectively closing a gap in the literature.

5. Additionally, the paper demonstrates tight lower bounds for Tikhonov regularization with a Hölder continuous kernel, proving that saturation is an unavoidable phenomenon in this context. The authors extend the results from [28] from real-valued output spaces to infinite dimensional output spaces using the same bias-variance decomposition, however with a simpler approach.

Overall, the research provides valuable insights into spectral learning algorithms and their application in high-dimensional settings.

**Weaknesses:**

1. While the paper does close a gap in the literature, it follows the usual lines in non-parametric regression over RKHSs. The results are somewhat expected, and the approach is not actually new.

2. The setting seems rather restrictive: The maps that are learned with classical regularization approaches are basically linear operators (Thm. 1) . Perhaps, the authors can state this more clearly to better distinguish between other current streams of operator learning research (in particular non-linear operator learning).

3. The difference between this work and previous work [38], particularly in the well-specified case, could be explained in more detail to highlight the novel contributions.

**Questions:**

Q1: Is the machinery with introducing vvRKHSs really necessary when functions of Hilbert-Schmidt operators are learned?

Q2: Theorem 2: Can "with high probability" be made more precise?

minor:
* some brackets for references are missing, e.g. p. 1, l.20, 33 and through out the manuscript
* the references [33] and [34] are the same, also [6] and [7]

**Limitations:**

yes

---

> ### Author Rebuttal · Authors · 2024-08-05
>
> We thank the reviewer for their encouraging feedback and their helpful comments. We have done our best to address all the questions. If we have done so, we would be grateful if the reviewer might consider increasing their score.
>
> **Weaknesses:**
>
> 1. We agree, we base our investigation on the typical integral operator approach [6], making use of a variety of arguments based on available literature. In that sense, the structure of the rates is not surprising. We argue, however, that *our contribution is precisely the insight that known arguments can be transferred to vector-valued learning in a dimension-free manner* by refining a tensor product trick first used by [38,29]. This was not clear in the initial line of work going back to [6], as a trace condition ruled out infinite-dimensional product kernels. In a practical context, based on the representer theorem in Proposition 1, our work provides the first theoretical foundation for learning the conditional mean embedding with general spectral algorithms. Furthermore, it unifies the theory for kernel-based functional regression [25], which is novel in that degree of generality to the best of our knowledge.
>
> 2. Thank you for pointing this out. We agree that in light of the recent interest in operator learning, we should comment on our results in this context. We note that our work can be directly interpreted as regularised (nonlinear) nonparametric operator learning: Bochner-universality shows that the vvRKHS is *sufficient* to learn all Bochner square integrable nonlinear operators. We hypothesise that, analogously to how scalar kernel regression is used to understand generalisation
> and early stopping in deep learning (for example via the NTK), our work could be a starting point for a theory of neural operator learning; although there is much theory that is still missing in the bigger picture. This seems to be a very interesting direction for future research.
>
> Our work also contains *linear* operator learning (related to [38]): in particular, when $X$ takes values in a Hilbert space $\mathcal{X}$ we choose the scalar kernel $k(x,x') = \langle x, x'\rangle_\mathcal{X}$, we recover precisely the linear setting of [38] via the vector-valued kernel $K = k \\times \\operatorname{Id}\_{\\mathcal{Y}} $. The assumptions in the linear and nonlinear settings require caution, however. The standard assumption that $k$ is bounded guarantees the crucial embedding property (EMB), which only holds for almost surely bounded $X$ in the linear case. However, with the kernel $k(x,x') = \\langle x,x'\\rangle_{\\mathcal{X}} $ and when $X$ is bounded, *our rates apply in the linear operator learning setting without modification, and appear to be novel for this field.*
>
> 3. We note that the authors of [38] work directly in the operator regression setting and do not assume boundedness of the covariate $X$, but subgaussianity. In our work, boundedness of the embedded covariates is assumed implicitly  through boundedness of the kernel $k$. In this sense, [38] is more general. On the other hand, in [38], the authors consider exclusively the well-specified setting and show rates up to $O(1/\sqrt{n})$, as no decay of eigenvalues of the integral operator is assumed and rate optimality under these assumptions is not addressed. In contrast, our assumptions allow for optimal rates up to  $O(1/n)$ by exploiting the aforementioned eigenvalue decay, in which sense our results are more general.
>
> We will include these discussions in the camera ready version.
>
> **Questions**
>
> 1. We see the reviewer's point and argue that the vvRKHS framework is fairly general: it allows both linear and nonlinear operator learning on Hilbert spaces
> as highlighted in W2 above, but also contains more general settings where one observes covariates $X$ on a general topological space without linear structure. We agree that generally, details about the machinery of vvRKHSs could be ignored when purely focusing on the operator learning setting. Nonetheless, interpreting HS-operators as vvRKHS functions allows convenient discussions based on the existing vvRKHS literature. One may also argue that our presentation may be practical for readers with an RKHS background and applications such as conditional mean embeddings are formulated in this setting; although this is likely up to personal preference and background.
>
> 2. We agree with the reviewer that Theorem 2 was introduced in an informal way. The formal version of the theorem can be found in [30] (Theorem 5). We will update Theorem 2 in that spirit in the camera ready version to make sure that what is hidden behind "with high probability" is correctly specified. For completeness we will add Theorem 5 from [30] in the appendix of our current submission so that the reader can have the theorem in full details.
>
> Finally, we thank the reviewer for spotting typos in the references. We will carefully proofread and correct all typos in the camera ready version.

---

### Official Review · Reviewer_f39z · 2024-07-15

**Soundness:** 3
**Presentation:** 3
**Contribution:** 3
**Rating:** 6
**Confidence:** 3

**Summary:**

This papers focuses on learning vector-valued functions in reproducing kernel Hilbert spaces (RKHS). The kernel in this case is an operator-valued function instead of a scalar-valued function. The papers considers kernel-based vector-valued regression with spectral regularization, which include ridge regression and kernel PCR. The contribution of the paper is theoretical: learning rates for vector-valued and spectral regularization-based regression are derived. In the case of kernel ridge regression upper and lower rates are provided. In the general case of spectral filter function only upper rates are given.

**Strengths:**

* The paper provides new theoretical results for vector-valued RKHS-based regression.
* The paper is well written.

**Weaknesses:**

* Contributions compared to previous work should be made more clear.

**Questions:**

* Section 3 focuses on kernel ridge regression (KRR). Learning rates and results on the saturation effect of vector-valued KRR have been reported in [1]. Could you make clear what are the contribution compared to [1] here? What are the main technical challenges compared to [1]?

[1] Li, Zhu, et al. "Towards Optimal Sobolev Norm Rates for the Vector-Valued Regularized Least-Squares Algorithm." JMLR (2024).

* Excess risk bounds for vector-valued learning with spectral filtering are provided in [2]. How these bounds can be compared to those obtained in this work?

[2] Baldassarre, Luca, et al. "Multi-output learning via spectral filtering." Machine learning (2012).

* Theorem 4 provides upper rates for vector-valued function learning with general spectral function. How optimality is maintained in this case?

* Could the results be extended to other classes of operator-valued kernels (e.g., non-separable kernels)?

**Limitations:**

A detailed discussion on limitations is missing.

---

> ### Author Rebuttal · Authors · 2024-08-05
>
> First, we thank the reviewer for their encouraging review of our work. We have done our best to respond to the reviewer's questions. If we have done so, we would be grateful if the reviewer might consider increasing their score.
>
> First of all, the reviewer highlighted that a comparison of our results to [1] and [2] should be made more thoroughly.
>
> *Comparison to [1].* [1] only considers Tikhonov regularisation, while the present work handles arbitrary spectral regularisation. The main difficulty is applying the machinery of vector-valued interpolation spaces developed in [1] to arbitrary filter functions. We will add a proof sketch at the beginning of the appendix to highlight the terms in the bound that require new analysis.
>
> *Comparison to [2].* We highlight the differences in the assumptions that are made to obtain excess risk bounds. 1) [2] focuses on vector-valued learning where the output space is $\mathcal{Y} = \mathbb{R}^d$, while we handle the more general case where $\mathcal{Y}$ is a potentially infinite-dimensional Hilbert space. This generalisation is important to study conditional mean embedding learning, a crucial step in nonparametric causal inference. 2) [2] focuses on the well-specified setting where the target function is assumed to belong to the hypothesis space, while we also handle the mis-specified setting. 3) [2] does not consider the effective dimension in their rate which corresponds to setting $p=1$ in assumption (EVD). 4) [2] works with bounded outputs, which is a stronger assumption than our (MOM) assumption.
>
> Overall, their excess risk bounds are obtained under more stringent assumptions than ours, and they focus in most of the paper on applications. In the well-specified setting, when $\mathcal{Y} = \mathbb{R}^d$, with bounded outputs and ignoring the effective dimension ($p=1)$, our rates are identical. Obtaining excess risk bounds under our more general assumptions requires very different proof techniques. We will add this discussion in the camera ready version of the manuscript.
>
> **Question on optimality after Theorem 4:** We realise that the conversation around optimality has been overshadowed by the fact that we did not clearly state what the information-theoretic lower bound is. Given assumption (EVD+) with parameter $p \in (0,1]$, assumption (MOM), assumption (SRC) with parameter $\beta > 0$, and any $\gamma \in [0,\beta)$, any learning algorithm $\hat {F}$, will satisfy (see Theorem 5 [1])
>
> $||\hat{F} -  F_*||_{\gamma}^2 = \Omega_P \left(n^{-\frac{\beta- \gamma}{\beta + p}}\right).$
>
> On the other hand, for an estimator $\hat F_{\lambda}$ with qualification $\rho$ (Eq.~(11) in our manuscript), upper rates given in Theorem 4 of our submission satisfy the following upper bound (with the correct choice of $\lambda$),
>
> $||\hat F_{\lambda} - F_{*}||_{\gamma}^2 = O_P\left(n^{-\frac{\min(\beta, 2\rho)- \gamma}{\min(\beta, 2\rho) + p}}\right),$
>
> if $\beta > \alpha - p$, and
>
> $||\hat F_{\lambda} - F_*||_{\gamma}^2 = O_P\left(n^{-\frac{\beta- \gamma}{\alpha}}\right),$
>
> if $\beta \leq \alpha - p$. Optimality is therefore maintained for the smoothness parameter $\beta$ in the interval $(\alpha - p, 2\rho]$. In the scalar-valued setting, this recovers the state-of-the-art rates of [3]. We hope this answer clarifies the question around optimality in Theorem 4, and we will make sure to include this detailed discussion in the camera ready version.
>
> **Question on more general vector-valued kernels:** While one could consider operator-valued kernels that do not have the multiplicative structure $K = k \operatorname{Id}$ (with $k$ a scalar-valued kernel), it seems to us that it is currently the only relevant kernel in the infinite-dimensional setting to be found in the literature (see references with applications in the introduction). We hypothesise that this has two main reasons. Firstly, such a kernel allows for the numerical computation and evaluation of the finite-sample estimators via a vector-valued representer theorem analogously to the real-valued setting, as highlighted by Proposition 1 in our manuscript. We are not aware of such results for other types of kernels. Secondly, the available technical investigations of vRKHS induced by such kernels show that critical properties like *universality* are already achieved by this type of kernel (this is addressed in Remark 3 in our manuscript), allowing for universal consistency when used in supervised learning. In theory, one may argue that such kernels are in a sense *sufficient* for vector-valued learning problems.
>
> That being said, we believe it is possible to generalise the result to different families of operator-valued kernels. For example, in the context of vector-valued learning with Tikhonov regularisation [4] or with general spectral regularisation [5], the cited works study vector-valued regression with kernel $K:\mathcal{X} \times \mathcal{X} \to \mathcal{L}(\mathcal{Y})$ such that $K_x$ is Hilbert-Schmidt (hence compact). In infinite dimension, this rules out the kernel $K = k \operatorname{Id}$ due to the non compactness of $\operatorname{Id}$. Note that the lower bound they obtain in both works requires $\operatorname{dim}(\mathcal{Y}) < + \infty$ while we do not impose such a restriction.
>
> However, we note that there is a technical caveat to directly transferring our results to more general kernels: we exploit the tensor product trick (vvRKHS is isomorphic to HS-operators) in order to apply real-valued arguments. For different kernels, such an isomorphism does generally not apply and it is not entirely clear if and how real-valued arguments can be used straightforwardly. This may require a new approach and  additional work.
>
> [3] Z. Haobo, et al. On the Optimality of Misspecified Spectral Algorithms
> [4] A. Caponnetto, E. De Vito. Optimal rates for the regularized least-squares algorithm
> [5] Rastogi, A, Sampath, S. Optimal rates for the regularized learning algorithms under general source condition

---

> > ### Comment · Reviewer_f39z · 2024-08-11
> >
> > Thank you for the additional information. However, some issues still need clarification.
> >
> > 1. The authors said that "[1] only considers Tikhonov regularisation, while the present work handles arbitrary spectral regularisation". Theorem 4 considers an estimator based on a general spectral filter. To obtain optimality, the authors would like to use a lower bound provided in [1] (Theorem 5 [1]), as mentioned in the response. But [1] considers only the case of L2-regularization and so the result cannot be used for general spectral regularization.
> >
> > 2. This paper does not provide a lower bound in the case of arbitrary spectral regularization.
> >
> > 3. Regarding the question about the main technical challenges compared to [1], the answer is "The main difficulty is applying the machinery of vector-valued interpolation spaces developed in [1] to arbitrary filter functions". Can you be more specific?

---

> > > ### Author Response · Authors · 2024-08-12
> > > **Answer to points 1 and 2**
> > >
> > > Thank you for your additional effort in pursuing the discussion. Below we address your concerns.
> > >
> > > **Regarding points 1 and 2**, the reviewer mentioned that ``the result (the lower bound from [1]) cannot be used for general spectral regularisation''. We agree that [1] focuses on vector-valued regression with Tikhonov regularisation. Their *upper bound* (Theorem 3 [1]) only applies to this setting. However, their *lower bound* (Theorem 5 [1]) is the information-theoretic lower bound, i.e. it applies to **any estimator**, including any of the spectral regularisation methods. Therefore **we do have a lower bound for arbitrary spectral regularisation**, directly inherited from [1].
> > >
> > > However, using the lower bound from [1] is not entirely satisfying as it shows that the upper bound for arbitrary spectral regularisation is not tight in the high smoothness regime. Indeed, the lower bound of [1] (Theorem 5) for the squared $\gamma-$norm is in $\Omega(n^{-\frac{\beta- \gamma}{\beta + p}})$ and the upper bound in our current submission (Theorem 4) is in $O(n^{-\frac{\min\\{\beta,2\rho\\}- \gamma}{\min\\{\beta,2\rho\\} + p}})$ (when $\beta + p > \alpha$). It shows that spectral regularization methods do not achieve the optimal rate when $\beta \geq 2 \rho$. In our current submission, we show that this is unavoidable when we employ *Tikhonov regularisation*. Indeed, Theorem 3 provides a lower bound that *applies specifically to Tikhonov regularisation* (for which $\rho = 1$) in $\Omega(n^{-\frac{\min\\{\beta,2\\} - \gamma}{\min\\{\beta,2\\} + p}})$. This demonstrates that the saturation effect for Tikhonov regularisation is unavoidable. What remains to be shown is the following: given a spectral algorithm with qualification $\rho$, can we obtain a lower bound specific to this spectral algorithm in $\Omega(n^{-\frac{\min\\{\beta,2\rho\\} - \gamma}{\min\\{\beta,2\rho\\} + p}})$? This would show that saturation is unavoidable for arbitrary spectral algorithms with qualification $\rho$. This is a challenging topic for future work.

---

> > > > ### Author Response · Authors · 2024-08-12
> > > > **Answer to point 3**
> > > >
> > > > **Regarding point 3**, we provide below a proof sketch of our main proof for the upper bound. We hope that it highlights the technical challenges compared to [1]. We will add this technical discussion in the camera ready version. The key technical challenge in extending the results of [1] to spectral filter functions lies in the analysis of the estimation error. Concretely speaking, the estimation error in $\gamma-$norm is bounded as (see line 1142 on page 38 in our appendix),
> > > >
> > > > $ \\|\hat{C}\_{\lambda} - C\_{\lambda}\\|\_{S\_2([\mathcal{H}]^{\gamma},\mathcal{Y})} \leq 3\lambda^{-\frac{\gamma}{2}}\\|(\hat{C}\_{\lambda} - C\_{\lambda})\hat{C}\_{X,\lambda}^{\frac{1}{2}}\\|_{S_2(\mathcal{H},\mathcal{Y})}.$
> > > >
> > > > With Tikhonov regularization, the authors of [1] exploited the fact that
> > > >
> > > > $Id\_{\mathcal{H}} = \hat{C}\_{X,\lambda}\hat{C}\_{X,\lambda}^{-1}$
> > > >
> > > > to obtain the following decomposition
> > > >
> > > > $(\hat{C}\_{\lambda} - C\_{\lambda})\hat{C}\_{X,\lambda}^{\frac{1}{2}} = (\hat{C}\_{YX}\hat{C}\_{X,\lambda}^{-1} - C\_{YX}C\_{X,\lambda}^{-1})\hat{C}\_{X,\lambda}^{\frac{1}{2}}= (\hat{C}\_{YX} - C\_{YX}C\_{X,\lambda}^{-1}\hat{C}\_{X,\lambda})\hat{C}\_{X,\lambda}^{-\frac{1}{2}}= (\hat{C}\_{YX} - C\_{YX}C\_{X,\lambda}^{-1}\hat{C}\_{X,\lambda})C\_{X,\lambda}^{\frac{1}{2}}(C\_{X,\lambda}^{-\frac{1}{2}}\hat{C}\_{X,\lambda}^{-\frac{1}{2}}).$
> > > >
> > > > Since with high probability we have $\\|C\_{X,\lambda}^{-\frac{1}{2}}\hat{C}\_{X,\lambda}^{-\frac{1}{2}}\\|\_{\mathcal{H}\to\mathcal{H}}\lesssim 1$, the following bound holds
> > > >
> > > > $\\|(\hat{C}\_{\lambda} - C\_{\lambda})\hat{C}\_{X,\lambda}^{\frac{1}{2}}\\|\_{S_2(\mathcal{H},\mathcal{Y})} \lesssim \\|(\hat{C}\_{YX} - C\_{YX}C\_{X,\lambda}^{-1}\hat{C}\_{X,\lambda})C\_{X,\lambda}^{\frac{1}{2}}\\|\_{S_2(\mathcal{H},\mathcal{Y})}. $
> > > >
> > > > In the proof of [1] (Lemma 6), the authors made the observation that
> > > >
> > > > $(\hat{C}\_{YX} - C\_{YX}C\_{X,\lambda}^{-1}\hat{C}\_{X,\lambda})C\_{X,\lambda}^{\frac{1}{2}} = \hat{\mathbb{E}}[(Y- F\_{\lambda}(X))\otimes (C\_{X,\lambda}^{\frac{1}{2}}\phi(X))] - \mathbb{E}[(Y - F\_{\lambda}(X))\otimes (C\_{X,\lambda}^{\frac{1}{2}}\phi(X))] $
> > > >
> > > > whose $S_2(\mathcal{H},\mathcal{Y})$ norm is controlled by concentration using a Bernstein's inequality. On the other hand, in our spectral method setting, we have to rely on the fact that (see Definition 2 for the definition of $g_{\lambda}$ and $r_{\lambda}$),
> > > >
> > > > $Id\_{\mathcal{H}} = \hat{C}\_{X}g\_{\lambda}(\hat{C}\_{X}) + r\_{\lambda}(\hat{C}_{X}),$
> > > >
> > > > to obtain the alternative decomposition
> > > >
> > > > $\hat{C}\_{\lambda} - C\_{\lambda} = \hat{C}\_{\lambda} - C\_{\lambda}(\hat{C}\_{X}g\_{\lambda}(\hat{C}\_{X}) + r\_{\lambda}(\hat{C}\_{X}))= (\hat{C}\_{YX} - C\_{\lambda}\hat{C}\_{X})g\_{\lambda}(\hat{C}\_{X}) - C\_{\lambda}r\_{\lambda}(\hat{C}\_{X}),$
> > > >
> > > > which yields two terms to be controlled,
> > > >
> > > > $\\|(\hat{C}\_{\lambda} - C\_{\lambda})\hat{C}\_{X,\lambda}^{\frac{1}{2}}\\|\_{S\_2(\mathcal{H},\mathcal{Y})} \leq  \underbrace{\\|(\hat{C}\_{YX} - C\_{\lambda}\hat{C}\_{X})g\_{\lambda}(\hat{C}\_{X})\hat{C}\_{X,\lambda}^{\frac{1}{2}}\\|\_{S\_2(\mathcal{H},\mathcal{Y})}}\_{(I)} + \underbrace{\\|C\_{\lambda}r\_{\lambda}(\hat{C}\_{X})\hat{C}\_{X,\lambda}^{\frac{1}{2}}\|\_{S\_2(\mathcal{H},\mathcal{Y})}}\_{(II)}.$
> > > >
> > > > To control term (I), we use the definition of the filter function $g_{\lambda}$ (Eq. (8)) to obtain that
> > > >
> > > > $\\|\hat{C}\_{X,\lambda}g\_{\lambda}(\hat{C}\_{X})\\|\_{\mathcal{H}\to\mathcal{H}} \lesssim 1.$
> > > >
> > > > Thus it suffices to control the term
> > > >
> > > > $\\|(\hat{C}\_{YX} - C\_{\lambda}\hat{C}\_{X})C\_{X,\lambda}^{-\frac{1}{2}}\\|\_{S\_2(\mathcal{H},\mathcal{Y})} = \\|\frac{1}{n}\sum_{i=1}^{n}\xi(x_i,y_i)\\|_{S_2(\mathcal{H},\mathcal{Y})}$
> > > >
> > > > where $ \xi(x,y) = \left((y-C_{\lambda}\phi(x))\otimes \phi(x)\right)C_{X,\lambda}^{-\frac{1}{2}}$. Since the random variable $\xi(X,Y)$ is not centered, we proceed by bounding $\mathbb{E}[\\|\xi(x,y)\\|\_{S_2(\mathcal{H},\mathcal{Y})}^m]$ for $m \geq 1$, and then use Bernstein's inequality to derive the upper bound on $\\|(\hat{C}\_{YX} - C\_{\lambda}\hat{C}\_{X})C\_{X,\lambda}^{-\frac{1}{2}}\\|\_{S_2(\mathcal{H},\mathcal{Y})} $. The technical details on how Bernstein's inequality is applied can be found in Theorem 8 and Lemma 8.
> > > >
> > > > Finally, to control term (II), Lemma 9 in our manuscript shows that
> > > >
> > > > $(II) \lesssim \\|\hat{C}\_{X,\lambda}^{\frac{1}{2}}r\_{\lambda}(\hat{C}\_{X})g\_{\lambda}(C_X)C_{X}^{\frac{\beta+1}{2}}\\|\_{\mathcal{H}\to\mathcal{H}}$
> > > > This term is analyzed in prior work on scalar-valued spectral method, see Theorem 16
> > > >
> > > > [3] Z. Haobo, et al. On the Optimality of Misspecified Spectral Algorithms.

---

### Author Rebuttal · Authors · 2024-08-05

We would like to thank the reviewers for their encouraging and positive feedback. They sparked very interesting discussions that we will use to improve our manuscript in the camera ready version.

As a summary here are the main points that were brought up by the reviewers:

1. As the proof for the upper rates is quite lengthy, it was mentioned that it is not straightforward to understand the technical novelties with respect to previous works in the proof techniques. To address this, we will incorporate a proof sketch at the beginning of the appendix in the camera ready version.

2. Two reviewers wondered if our results could be extended to more general vector-valued kernels. We refer to our answer to reviewers f39z and U1CH.

3. One reviewer noticed that a conclusion should be added. We will use the extra page in the camera ready version to add it and address the points raised by the reviewers.

---

### Decision · Program_Chairs · 2024-09-25

**Decision:**

Accept (poster)

**Comment:**

A theory paper on optimal learning rates for vector-valued regression using spectral regularization methods. All reviews are positive, underlining the fact that the paper is well-written, and the contribution clear, including upper and lower bounds, the treatment of the ill-specified setting and possible saturation effects. Some concern was raised as to the technical novelty with respect to the significant body of previous work in the real-valued case, but the reviewers were satisfied by the author's rebuttal pointing out in particular new technical insights in vector-valued function approximation, these points will be highlighted more prominently in the camera-ready version.